## Registered report

psychology

depression, cognitive biases, optimism, belief updating

**Author for correspondence:**
Catherine Hobbs
e-mail: c.hobbs@bath.ac.uk

# Is depression associated with reduced optimistic belief updating?

Catherine Hobbs, Petra Vozarova, Aarushi Sabharwal, Punit Shah and Katherine Button

Department of Psychology, University of Bath, 10 West, Bath BA2 7AY

CH, 0000-0002-2324-5882

When asked to evaluate their probability of experiencing a negative life event, healthy individuals update their beliefs more following good news than bad. This is referred to as optimistic belief updating. By contrast, individuals with depression update their beliefs by a similar amount, showing reduced optimism. We conducted the first independent replication of this effect and extended this work to examine whether reduced optimistic belief updating in depression also occurs for positive life events. Replicating previous research, healthy and depression groups differed in belief updating for negative events ($\beta = 0.71$, 95% CI: 0.24, 1.18). Whereas healthy participants updated their beliefs more following good news than bad, individuals experiencing depression lacked this bias. However, our findings for positive events were inconclusive. While we did not find statistical evidence that patterns of belief updating between groups varied by valence ($\beta = -0.51$, 95% CI: −1.16, 0.15), mean update scores suggested that both groups showed largely similar updating for positive life events. Our results add confidence to previous findings that depression is characterized by negative future expectations maintained by reduced updating in response to good news. However, further research is required to understand the specificity of this to negative events, and into refining methods for quantifying belief updating in clinical and non-clinical research.

## 1. Introduction

Cognitive neuropsychological models of depression emphasize the role of maladaptive negative beliefs in contributing to the development and maintenance of depression [1]. Individuals experiencing depression are believed to hold pessimistic views of themselves, their future and the world around them. Information is processed in a manner consistent with these views, resulting in negative cognitive biases [2]. Symptom improvement can be

observed when dysfunctional schema and biased cognition are addressed in therapy, suggesting a potential causal role [3].

However, while depressed individuals' self-beliefs and world views may be more pessimistic relative to healthy controls, evidence is emerging that this may be because healthy controls hold overly optimistic beliefs about themselves and their world views. Thus, rather than being pessimistic *per se*, depression may be better characterized as a loss of the optimistic outlook seen in healthy individuals. For example, people in the general population are overly optimistic in their judgements of self-performance. In comparison, depression is associated with less optimistic, but more accurate estimates [4]. Positive processing biases may be protective for mental health in increasing self-esteem, confidence and life satisfaction [5,6]. A loss of these positive biases sometimes referred to as 'depressive realism', may perpetuate depression symptoms through reinforcing negative views of the self.

Recent work has extended this theory to beliefs about the probability of experiencing future life events. Individuals experiencing high levels of depressive symptoms showed a reduced optimism bias when estimating the probability of experiencing a life event. Participants with high levels of depression were less likely to predict experiencing a positive event and more likely to predict experiencing a negative event [7]. However, this does not explain how these biases are maintained when individuals are presented with information that challenges existing perceptions. To examine this a belief updating task has been developed. In this task, participants are asked to estimate their chances of experiencing a negative life event and are then presented with the average probability for someone in a similar sociocultural environment. When asked to re-estimate their chances, healthy individuals have been found to preferentially update their initial beliefs after receiving desirable information (i.e. the probability is lower than their initial estimate), compared with undesirable information (i.e. the probability is greater than their initial estimate). This has been deemed as evidence of optimistic belief updating in healthy individuals [8–10] and posited as a mechanism that serve to maintain positive self-esteem.

By contrast, individuals with depression have not been found to display this asymmetry in belief updating. Korn *et al.* [11] reported that individuals with major depressive disorder showed no differences in the probability of changing their beliefs following desirable or undesirable information. By contrast, healthy controls continued to preferentially update their beliefs following desirable information. These results were later replicated within the same research group, using functional MRI to examine the brain activity underlying belief updating. Whereas healthy individuals displayed diminished neural coding of undesirable information, individuals with depression showed strong neural coding in response to undesirable information in the right inferior parietal lobule and right inferior frontal gyrus [12]. However, despite these findings being upheld as evidence of reduced optimistic belief updating in depression and being widely cited, they have not yet been independently replicated. Replication will increase our certainty regarding the role of belief updating biases in depression.

Additionally, the validity of findings of optimistic belief updating has been questioned [13–18]. At the time of writing, studies investigating loss of optimistic belief updating in depression have used negative life events only. In this case, optimistic updating would be apparent where beliefs are updated more when the event is *less likely* than initially estimated than when the event is *more likely* than initially estimated. However, for positive life events, optimistic updating would be apparent where beliefs are updated more when the event is *more likely* than initially estimated than when the event is *less likely* than initially estimated. If optimistic belief updating biases are indeed present they should be greater in response to desirable information for both positive and negative life events for healthy individuals, although this would result in updates in numerically opposite directions [16].

Although updating of beliefs for positive outcomes has not been investigated in depressed samples, they have been investigated in healthy samples with mixed results. In one study, while participants continued to show biased updates toward desirable information for negative life events, the opposite effect occurred for positive life events. Participants were more likely to update their beliefs following undesirable information for positive life events, therefore displaying a pessimism bias [16]. However, this work has been criticized on the basis of including life events with a high probability of occurring (e.g. 'Going to your favourite restaurant') and using simulated probabilities. Participants may therefore have been aware that the presented average probabilities were inaccurate, influencing their responses [19]. When using these events with actual probabilities of occurring within the next month, participants were found to preferentially update beliefs in response to desirable information for both positive and negative events [19]. However, another study observed optimistic biased updating only for negative events, no bias was observed for positive life events [20]. It is, therefore, unclear whether optimistic belief updating is present in healthy individuals for both positive and negative life events.

Optimistic belief updating for positive events has not yet been explored in relation to depression symptoms. It has been argued that depression is characterized by a blunting of emotional response to both positive and negative stimuli [21]. However, others have suggested information processing is relatively intact for negative information and only aberrant for positive information [22–24]. Research examining future belief updating to date suggests an emotional blunting in depression, where participants were not influenced by the desirability of estimates (i.e. whether it was good or bad news for them). However, it is possible that this is specific to negative life events, and effects may differ according to desirability for positive life events. Incorporating positive life events into this task would allow us to examine whether this is a general bias apparent across events of different valences, or whether this may be dependent on the information presented. This would improve our understanding of the mechanisms of reduced optimistic belief updating in depression and may have implications for clinical strategies to remediate reduced optimism.

In this study, we aim to replicate findings of Korn *et al.* [11] indicating reduced optimistic belief updating in individuals with depression. Specifically, we aim to replicate their finding of preferential updates in beliefs of experiencing negative life events following desirable versus undesirable information in healthy controls, but no differences in individuals experiencing depression. This would be the first independent replication of this effect. Furthermore, we aim to build upon previous results by incorporating positive life events into this paradigm. This will provide us with a greater understanding of the precise mechanisms underlying reduced optimistically biased belief updating previously observed in depression.

# 2. Hypotheses

Hypothesis 1: In line with Korn *et al.* [11] our primary hypothesis was that healthy controls would show optimistically biased belief updating for negative life events, as indicated by a greater change in beliefs following desirable versus undesirable information for negative life events. By contrast, we hypothesized that individuals with depression would show a reduced optimistic belief updating for negative life events, as indicated by smaller differences in change in beliefs following desirable versus undesirable information for negative life events (figure 1).

Hypothesis 2: We hypothesized that this effect would be consistent across both positive and negative life events. That is, healthy controls would show greater changes in beliefs following desirable versus undesirable information irrespective of the valence of the life event. Individuals with depression would show smaller differences in change in beliefs following desirable versus undesirable information for both positive and negative life events (figure 1). However, given the previous mixed findings regarding biased belief updating for positive life events, and no current literature for individuals with depression, we acknowledged that our belief in this hypothesis is weak.

Hypothesis 3: In line with Korn *et al.* [11] we predicted that depressed individuals would on average initially rate their chances of experiencing a negative life event as higher compared with healthy controls.

Hypothesis 4: We extended this hypothesis to initial estimates of positive life events, such that depressed individuals would on average initially rate their chances of experiencing a positive life event as lower compared with healthy controls.

# 3. Study design

A mixed-model design with group as a between-subject factor (depressed versus healthy controls), and desirability of life events (undesirable versus desirable) and valence (positive versus negative) as within-subject effects, was used.

# 4. Sampling plan

## 4.1. Recruitment

Due to the COVID-19 pandemic, recruitment took place online using the participant recruitment platform 'Prolific' [25]. Screening surveys were completed online using 'Qualtrics' [26]. Two groups of participants were screened and recruited on the basis of Patient Health Questionnaire (PHQ-9) scores according to recommended clinical guidelines [27]. A moderately severe to severely depressed sample

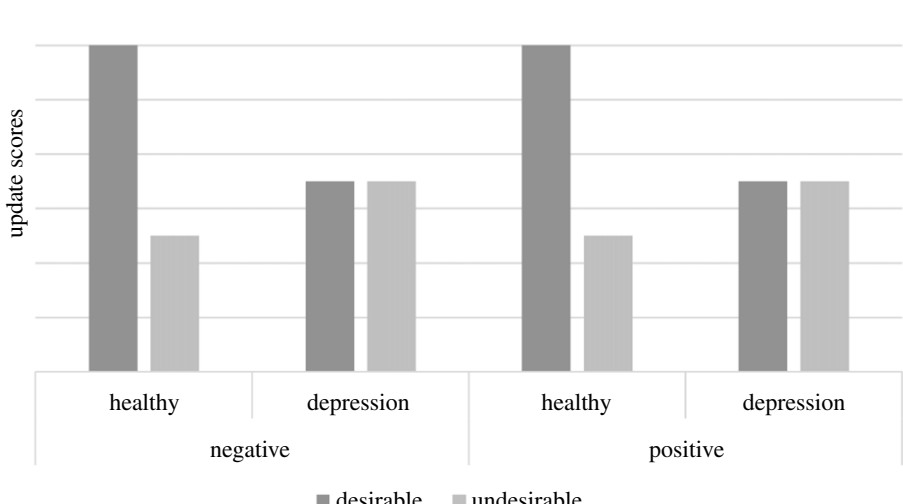

**Figure 1.** Hypothesized differences between absolute mean updates in the likelihood of life events according to desirability and valence of life event, for healthy controls and participants with moderate to severe levels of depression.

was recruited with PHQ-9 scores greater than or equal to 15. A healthy control sample was recruited with PHQ-9 scores less than or equal to 4. Participants were only invited to take part in the test phase of the study if their PHQ-9 scores were within these ranges. To optimize recruitment of participants with high levels of depression we restricted advertisement of this initial survey to participants that have reported experiencing a current diagnosed mental health condition, using custom pre-screening on Prolific. A separate survey with the same procedure was used to recruit healthy controls without this restriction.

To reduce participant burden during online data collection, we did not conduct diagnostic interviews. However, based on previous research within this group [28], we expect 93% of individuals with PHQ-9 scores greater than or equal to 15 to meet diagnostic criteria for a major depressive episode. In combination with targeted recruitment of patients reporting a current mental health condition on Prolific, we anticipated that our sample would have similar clinical profiles to those in previous research within this field [11,12].

## 4.2. Inclusion criteria

Participants in both groups were aged 18 and over, fluent in written and spoken English, with normal or corrected to normal vision, and were current residents of the United Kingdom. Participants were included irrespective of any comorbid psychiatric disorders or current treatment for depression. To ensure high quality of data, custom pre-screening was used on Prolific to identify participants that had previously completed greater than or equal to five studies with a 98% acceptance rate.

## 4.3. Exclusion criteria

As outlined above, only participants with PHQ-9 scores less than or equal to 4 or greater than or equal to 15 were invited to participate. In keeping with Korn *et al.* [11], participants who reported a history of, or current substance abuse, were excluded.

## 4.4. Sample size and rationale

Korn *et al.* [11] reported an interaction between desirability (desirable/undesirable) and group (major depressive disorder (MDD)/healthy) of $F_{1,35} = 6.9$, $p = 0.013$, $\eta_p^2 = 0.17$. To account for potential publication bias we reduced this effect size by one-third, $\eta_p^2 = 0.113$. As the correlation between within-subject effects has not been reported in previous research we took a conservative approach of assuming no correlation. Sample size calculation performed in G*Power [29] for a within-between interaction using ANOVA indicated that 108 participants would be required to detect an effect of $\eta_p^2 = 0.113$ with 95% power at an alpha level of 0.005. We, therefore, aimed to recruit two groups of 54 participants.

## 4.5. Blinding

Blinding was not applied as testing took take place online, eliminating the possibility of experimenter effects.

# 5. Materials

Testing took place online, self-report measures were completed using 'Qualtrics' [26] and the belief updating task was completed using 'Inquisit' [30].

## 5.1. Self-report measures

Prior to completing the task participants completed a number of self-report measures as outlined below.

### 5.1.1. Depression

The Beck Depression Inventory II (BDI-II) [2] was used to measure depression symptoms. The BDI-II is a 21-item self-report questionnaire of depression symptoms. Possible responses for each item range from (0), indicating low levels, to (3) indicating high levels. Higher scores indicate higher levels of depression symptoms.

The PHQ-9 [27] was used as an additional measure of depression symptoms, as this is widely used in UK clinical practice and is a more appropriate depression screening measure due to its brevity [31]. The PHQ-9 is a nine-item self-report questionnaire of depression symptoms. Possible responses range from (0) 'Not at all' to (3) 'Nearly every day'. Again, higher scores indicate higher levels of depression symptoms. The PHQ-9 and BDI-II demonstrate strong correlation in general and clinical populations [32,33].

### 5.1.2. Anxiety

General anxiety was measured using the Generalised Anxiety Disorder Questionnaire (GAD-7) [34]. The GAD-7 is a seven-item questionnaire of anxiety symptoms, with responses ranging from (0) 'Not at all' to (3) 'Nearly every day'. Higher scores indicate higher levels of anxiety symptoms.

Social anxiety was measured using the Brief Fear of Negative Evaluation Scale (BFNE) [35]. This is a 12-item self-report measure of fear of negative evaluation by others.

### 5.1.3. Trait optimism

Trait levels of optimism and pessimism were measured using the Life Orientation Test–Revised (LOT-R) [36]. Participants were asked to indicate their agreement with 10 statements, ranging from (1) 'I agree a lot', to (5) 'I disagree a lot'.

### 5.1.4. Positive and negative mood

The Positive and Negative Affect Scale [37] (PANAS) was used to measure state mood before and after completion of the task. Participants were asked to indicate to what extent they currently experienced 10 positive and 10 negative emotions, ranging from (1) 'very slightly or not at all' to (5) 'extremely'.

### 5.1.5. Demographics and clinical characteristics

Information was collected on age, gender, ethnicity, relationship status, employment status, highest qualification and living situation. Participants were asked if they have previously experienced depression, whether they have received any treatment for depression (both past and current), and whether a family member has experienced depression.

## 5.2. Belief updating task

Following completion of the self-report measures, participants were asked to complete a computerized task measuring beliefs of experiencing a number of life events. A task was created on Inquisit [30], replicating the methods outlined in Korn *et al.* [11] but with the addition of positive life events. This task has been made publicly available on the Open Science Framework (https://osf.io/aqsrb/).

(*a*) first estimate and presentation of actual average probability

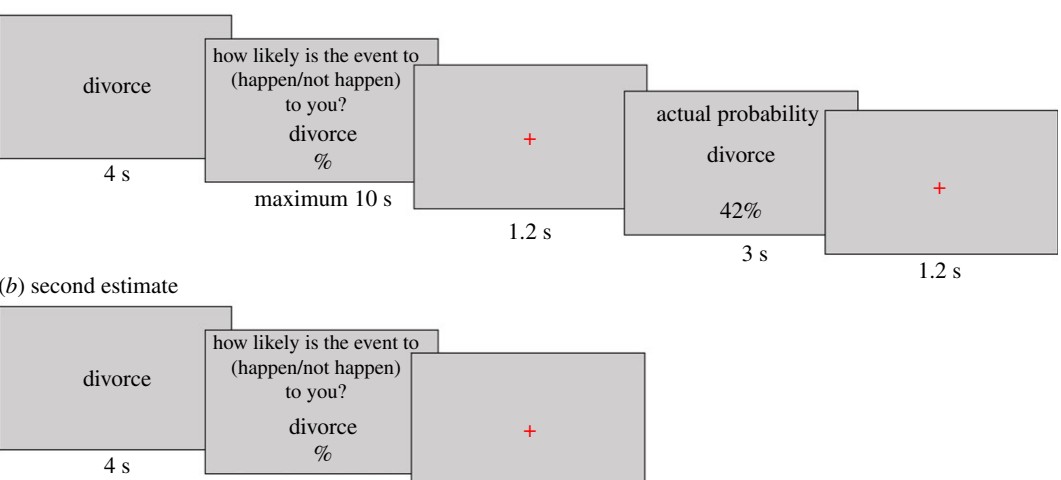

(*b*) second estimate

**Figure 2.** Main task procedure.

### 5.2.1. Stimuli

Task stimuli consisted of short descriptions of 40 negative and 40 positive life events and the average probability of someone in the same environment as the participant experiencing these events in their lifetime. Participants completed two training trials for positive and negative life events, resulting in 42 negative and 42 positive life events overall. In keeping with previous research [11] very common (greater than 70%) and very rare (less than 10%) events were not included. This allowed for change between initial and re-estimates. To allow for variability in ratings for high and low probability life events, participants were told that event probabilities fall within 3% and 77%. Of the 72 negative life events used by Korn *et al.* [11], we used 42 of the same negative life events, although we calculated new estimates to account for possible changes in available statistics for probabilities following the publication of Korn *et al.* [11] (electronic supplementary material, table S1). We were unable to use 30 of the life events used previously, as the event probabilities were very rare (less than 10%), very common (greater than 70%), or data for lifetime probability was not available (electronic supplementary material, table S2). In addition, we identified estimates for 42 positive life events (electronic supplementary material, table S3). Estimates were derived from reputable resources including peer-reviewed academic journal articles, governmental resources (e.g. the Office for National Statistics), non-governmental organization resources, and marketing and consumer data companies.

### 5.2.2. Task procedure

For each life event, participants were presented with a short description of the life event for 4 s and asked to imagine this event happening/not happening to them. Following this, participants were asked to estimate the probability of this event happening/not happening to them in their lifetime. A response threshold of 10 s was imposed. A fixation cross was then displayed for 1.2 s, before participants were presented with the average probability of this event happening/not happening to someone like them for 3 s. A fixation point was again displayed for 1.2 s before this sequence was repeated for the remaining life events (figure 2*a*).

Participants were then asked to re-estimate the probability of experiencing/not experiencing the events that were presented previously. A short description of the event was displayed for 4 s, before participants were again asked to estimate their probability of experiencing/not experiencing the event in their lifetime. Again, a response threshold of 10 s was imposed. A fixation cross was then displayed for 1.2 s, before this sequence was repeated for the remaining life events (figure 2*b*). Participants also completed two training trials to ensure that they understand the task instructions.

Participant's memory of the average probability of life events was tested following the completion of probability estimations. Participants were asked to recall the average probability of each event happening for someone like them as presented previously (i.e. the actual probability). Subjective ratings for each life

event were then collected. Participants were asked to rate each event on seven scales; vividness, familiarity, prior experience, emotional arousal, negativity, positivity and controllability. Each scale was rated from 1, indicating low levels, to 6, indicating high levels.

Separate blocks were completed for negative and positive life events, with order of completion counterbalanced (i.e. participants completed estimations, recall and ratings for all events of one valence, before repeating each procedure for all events of the remaining valence). This ensures that a pure replication of the procedure used by Korn *et al.* [11] was available for the subgroup of participants who completed the negative block first. Within each of the positive and negative life events estimation blocks, half of the events were framed as 'happening' whereas the other half were framed as 'not happening'. More specifically, in half of the blocks participants were asked to estimate the probability of the event happening to them in their life time and were presented with the average probability of the event occurring. In the other half of the blocks, participants were asked to estimate the probability of the event not happening to them in their lifetime and were presented with the average probability of the event not occurring. Life events were, therefore, split into four lists of 20 events each (two lists of negative events, and two list of positive events). Average probabilities did not significantly differ between lists. The order of happening versus not happening, and list completion order, was counterbalanced between participants (electronic supplementary material, tables S4 and S5).

## 5.3. Debriefing

Although presented probabilities originate from validated sources, participants' own perceptions may differ, contributing to scepticism regarding the credibility of probabilities. To account for this, participant's perceptions of the credibility of the presented probabilities for life events were measured through self-report following completion of the belief updating task. Participants were asked to indicate how strongly they agree with the item 'The average probabilities presented in the task were accurate', on a five-point scale ranging from 'Strongly Agree' to 'Strongly Disagree'.

### 5.3.1. Task outcomes

To allow us to combine estimates between 'not happening' and 'happening', participants' estimates in the 'not happening' blocks were subtracted from 100 to calculate the equivalent probability of the event happening.

Estimation errors were calculated by subtracting the actual probability of events from the participant's initial estimate of the events (i.e. estimation error difference = initial estimate − actual probability). Overestimations are, therefore, represented by positive values, whereas underestimations are represented by negative values.

Participant trials for each life event were categorized according to whether the actual probability presented is desirable or undesirable. For negative life events, trials were deemed as desirable if the actual probability was lower than the participant's initial estimate (i.e. the participant initially overestimated their probability of experiencing the life event, a positive estimation error). For negative life events, trials were deemed as undesirable if the actual probability was greater than the participant's initial estimate (i.e. the participant underestimated their probability of experiencing the life event, a negative estimation error) (figure 3).

For positive life events, trials were deemed as desirable if the actual probability was greater than the participant's initial estimate (i.e. the participant initially underestimated their probability, a negative estimation error). Trials were deemed as undesirable if the actual probability was lower than the participant's initial estimate (i.e. the participant initially overestimated their probability, a positive estimation error) (figure 3).

The primary outcome of this task is the extent to which participants change their initial estimates, following information regarding the average probability of experiencing the life event (continuous variable). Optimistic biased updating is indicated by greater changes toward desirable information relative to undesirable information. To calculate update scores, re-estimates (i.e. estimates made *after* presentation of the actual probability) were subtracted from initial estimates (i.e. update = initial estimate − re-estimate). These values were then coded as positive when the update (the re-estimate) moved toward the actual probability, and negative when the update moved away from the actual probability. Positive scores, therefore, indicate movement toward the actual probability, whereas negative scores indicate movement away from the actual probability (irrespective of valence and desirability).

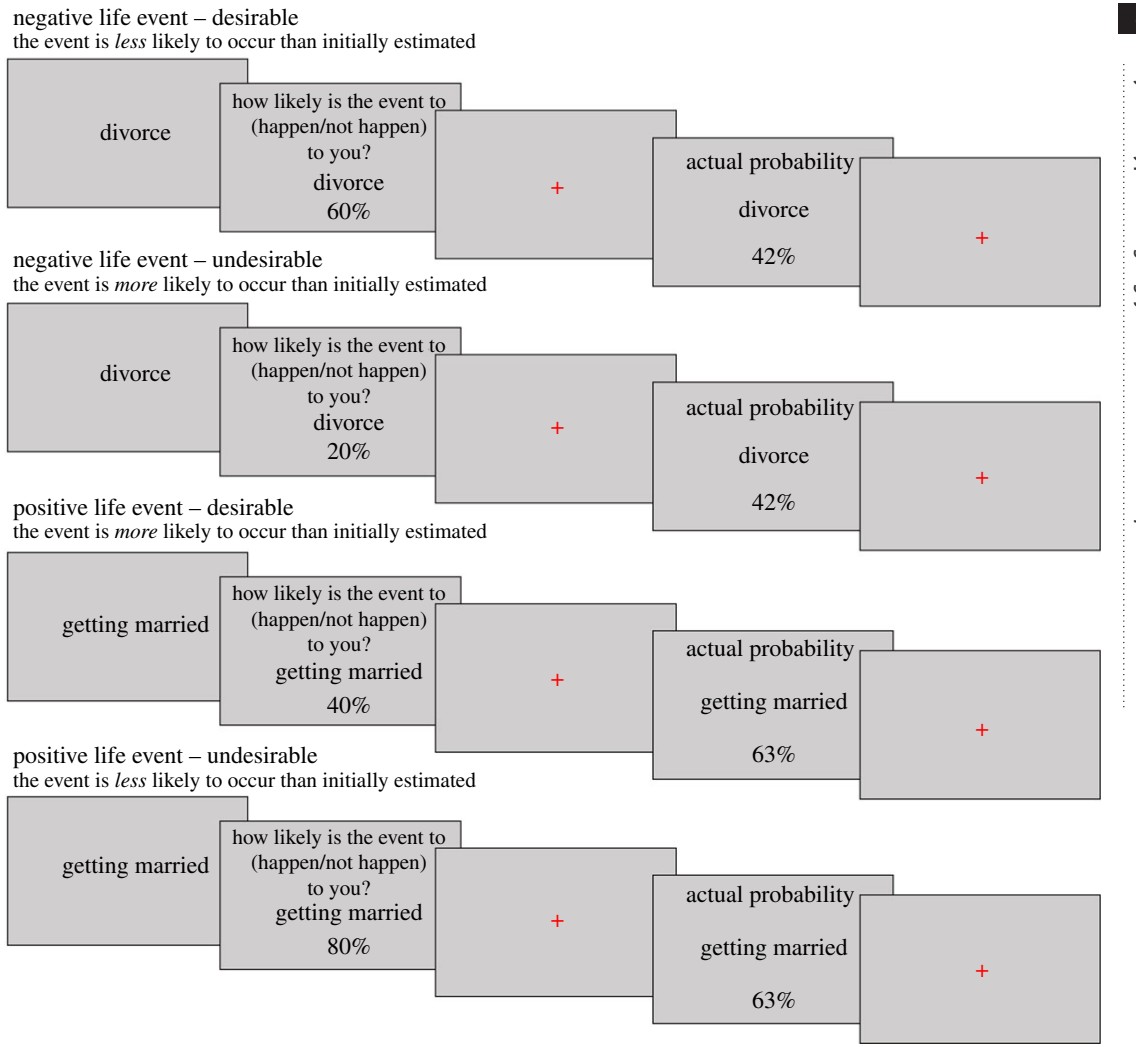

**Figure 3.** Example of desirable versus undesirable categorizations in positive and negative life events. For negative life events initial overestimations are desirable, whereas initial underestimations are undesirable. For positive life events, initial underestimations are desirable, whereas initial overestimations are undesirable.

Memory errors were calculated by subtracting the participant's recalled average probability from the actual probability of the event (i.e. memory error = actual probability − participant's recollection of probability).

### 5.3.2. Quality check and exclusion criteria

Participants were not exposed to experimental manipulation within this study and were presented with the average probability of life events based on validated statistics. A positive control does, therefore, not apply to this study. However, to ensure data quality, following the procedure of Korn *et al.* [11] trials were excluded if participants do not respond within the maximum response limit of 10 s. Responses were restricted to within 3% and 77% for the probability of the event happening, and within 23% and 97% for the event not happening, to reflect the range of probabilities specified to participants at the beginning of the task. Trials where the estimation error is 0 (i.e. the initial estimate is the same as the actual probability) were excluded from analyses including desirability, as this cannot be categorized. However, these trials were retained in analyses where calculation of desirability was not required.

The inclusion of 'happening' and 'not happening' trials controlled for any directional effects of any participant not appropriately engaged in the task. For example, selecting responses at random or selecting the same response will add noise (making our analyses more conservative), but is unlikely to introduce directional bias into the estimates. Therefore, all data from participants completing the task were included in the analyses.

healthy group (PHQ-9 < 5)                depression group (PHQ-9 >= 15)

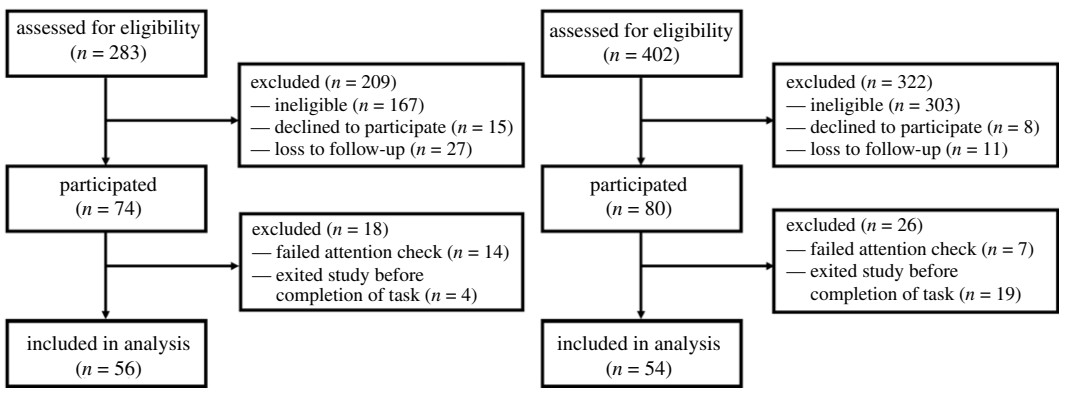

**Figure 4.** Consort flow diagram for the healthy and depression groups.

As outlined in the inclusion criteria, to ensure data quality participants were restricted to those that have previously completed at least five studies with a 98% acceptance rate on Prolific. Attention checks were also placed throughout the self-report measures and the belief updating task, in the form of eight items directing participants to provide a specified response (e.g. 'Please select "Disagree"'). Participants that failed any of these attention checks were considered non-compliant and excluded (figure 4).

# 6. Analysis plan

The data and code that support the findings of this study are openly available in the University of Bath Research Data Archive (https://doi.org/10.15125/BATH-01078) [38].

## 6.1. Inference criteria

A frequentist approach was taken using a stringent alpha level of 0.005 [39].

## 6.2. Primary analyses

Hypothesis 1: Healthy controls will show a greater change in beliefs following desirable versus undesirable information for negative life events, whereas individuals with depression will show a smaller difference in change in beliefs following desirable versus undesirable information for negative life events.

We used a mixed-effects linear regression model using only trials for negative life events. Absolute mean update scores were entered as the outcome, and group (healthy versus depressed), desirability (desirable versus undesirable), and an interaction term between group and desirability entered as predictors. Subject was entered as a random effect to account for the repeated measures design.

ANOVA and linear regression are equivalent statistical models, both following general linear modelling [40]. Our mixed-effects linear regression model, therefore, corresponds to the mixed-design ANOVA used by Korn *et al.* [11], but with the additional interpretational advantages provided by the regression coefficients.

Hypothesis 2: The effect of optimism bias will be consistent across both positive and negative life events. Healthy controls will show a greater change in beliefs following desirable versus undesirable information irrespective of the valence of the life event. Individuals with depression will show smaller differences in change in beliefs following desirable versus undesirable information for both positive and negative life events.

We conducted a mixed-effects linear regression model using trials for both positive and negative life events. Absolute mean update scores were entered as the outcome, and group (healthy versus depressed), desirability (desirable versus undesirable) and valence (positive versus negative) entered as predictors. Additionally, interaction terms between group and desirability, group and valence, desirability and valence, and group, desirability and valence were entered as additional predictors. Subject was entered as a random effect to account for the repeated measures design.

Hypothesis 3: Initial estimates of negative life events will be associated with depression, such that depressed individuals will on average initially rate their chances of experiencing a negative life event as higher compared with healthy controls.

This hypothesis was tested using linear regression using only trials for negative life events. Initial scores were entered as the outcome, and group (healthy versus depressed) was the predictor.

Hypothesis 4: Initial estimates of positive life events will be associated with depression, such that depressed individuals will on average initially rate their chances of experiencing a positive life event as lower compared with healthy controls.

This hypothesis was tested using a linear regression model using only trials for positive life events. Initial scores were entered as the outcome, and group (healthy versus depressed) was the predictor.

## 6.3. Additional analyses

Following the procedure of Korn *et al.* [11], we investigated whether any observed differences in update scores between depressed individuals and healthy controls are attributable to estimation errors (e.g. the difference between initial estimates and presented average probability). Scaled absolute mean update scores will be calculated by dividing mean update scores by absolute mean estimation errors. A mixed-effects linear regression model was created using only trials for negative life events. Scaled absolute mean update scores were entered as the outcome, with group (healthy versus depressed), desirability (desirable versus undesirable) and an interaction term between group and desirability entered as predictors. Subject was entered as a random effect to account for the repeated measures design.

This model was also repeated using both trials for positive and negative events, with valence, and an interaction term between valence and desirability, valence and group, and valence, desirability and group added as additional predictors.

To examine whether there was a dose–response relationship between depression symptoms and optimism bias, the linear regression models outlined in hypotheses 1 and 2 were repeated with continuous BDI-II scores entered as a predictor individually and in interaction with desirability. Group was entered as an additional predictor as a design variable. These models were repeated for PHQ-9 scores.

To assess whether any observed differences in updates according to group are attributable to trait levels of optimism the regression models outlined for hypotheses 1 and 2 were repeated with LOT-R scores as an additional predictor.

Differences in absolute mean memory errors were examined using a mixed-effect linear regression model. Absolute mean memory errors were entered as the outcome, with group (healthy versus depressed), desirability (desirable versus undesirable) and valence (positive versus negative) entered as predictors. Additionally, interaction terms between group and desirability, group and valence, desirability and valence, and group, desirability and valence were entered as additional predictors. Subject was entered as a random effect. The linear regression models outlined in hypotheses 1 and 2 were repeated with memory estimates as a predictor.

To assess whether observed results in the primary analyses were influenced by the framing of life events (happening versus not happening), these models were repeated with framing entered as an additional predictor. Likewise, to assess whether the observed results were influenced by order effects of the valence of life events, order of completion for positive and negative blocks was entered as an additional predictor to these models.

To examine whether the observed results in the primary analyses were influenced by participants' ratings of vividness, familiarity, prior experience, emotional arousal, negativity, positivity and controllability, the models specified in the primary analyses section were repeated with ratings added as additional predictors in separate models. If the results of the primary analyses are similar following adjustment for participants' ratings, this suggests that variations in perceptions of life events between participants did not significantly influence our findings regarding belief updating. We also conducted sensitivity analyses, repeating the primary analyses but with positive and negative events classified according to each participant's negativity and positivity ratings, to see how robust our findings were to any discrepancies in classification of positive/negative events. To classify life events according to participants' ratings, we aggregated positivity and negativity ratings by calculating the mean of negativity ratings and reverse coded positivity ratings. We then categorized the life event as being perceived as positive if the value was in the lower half of the scale (less than 3.5), neutral if the event

was at the midpoint of the scale (3.5), and negative if the value was in the upper half of the scale (greater than 3.5).

Dependent on evidence from our regression models that the order in which participants complete negative and positive events blocks has a strong influence on the results, we planned to repeat hypotheses 1 and 3 in the half of participants who completed the negative events first, to provide an uncontaminated replication of Korn *et al.*'s [11] finding. This sample ($n = 27$ per group) would have provided 86% power to detect an effect of $\eta_{\mathrm{p}}^2 = 0.113$ at an alpha level of 0.05. However, as we found no evidence of order effects, we did not conduct this analysis.

To account for potential variations in perceptions of the credibility of probabilities presented for life events, we conducted a sensitivity analysis replicating the primary analyses with data for participants who indicated scepticism excluded.

## 6.4. Exploratory analyses

As participants' age varied by group, we conducted exploratory analyses to examine whether our findings were influenced by age by including it as an additional predictor in the models for our primary analyses outlined above.

# 7. Results

## 7.1. Sample

The study sample consisted of 56 participants in the healthy group and 54 participants in the depression group. Full recruitment information is available in figure 4, sample demographics are outlined in table 1, and clinical characteristics are provided in table 2. There was some variation in demographics according to the group. Participants experiencing depression were on average younger, a greater proportion was female, they were less likely to be employed or a homeowner and were more likely to not be in a relationship.

**Hypothesis 1: Healthy controls will show a greater change in beliefs following desirable versus undesirable information for negative life events, whereas individuals with depression will show a smaller difference in change in beliefs following desirable versus undesirable information for negative life events.**

In support of our hypothesis, we found evidence that the relationship between update scores and group varied according to the desirability of presented information ($\beta = 0.71$, $\beta$ 95% CI: 0.24, 1.18, $p = 0.004$). Participants in the healthy comparison group displayed optimistic belief updating, changing their beliefs to a greater extent following desirable (M 12.49, s.d. 7.76) versus undesirable information (M 7.98, s.d. 6.72; $t_{108} = 3.54$, $p < 0.001$). In comparison, individuals experiencing depression on average updated their scores to a largely similar extent following desirable or undesirable information (Desirable: M 10.50, s.d. 7.30, Undesirable: M 11.34, s.d. 7.80, $t_{108} = -0.65$, $p = 0.518$, figure 5).

**Hypothesis 2: The effect of optimism bias will be consistent across both positive and negative life events. Healthy controls will show a greater change in beliefs following desirable versus undesirable information irrespective of the valence of the life event. Individuals with depression will show smaller differences in change in beliefs following desirable versus undesirable information for both positive and negative life events.**

When analysing both positive and negative life events, we again found evidence of optimistic belief updating in healthy individuals (Group × Desirability: $\beta = 0.71$, 95% CI = 0.24, 1.17, $p = 0.003$). Participants in the healthy group updated their beliefs more toward desirable (M 10.16, s.d. 7.09) versus undesirable information (M 7.90, s.d. 7.20; $t_{324} = 2.55$, $p = 0.011$), whereas no difference was observed in the depression group (Desirable: M 9.62, s.d. 8.41, Undesirable: M 10.80, s.d. 7.28; $t_{324} = -1.31$, $p = 0.192$). We did not find statistical evidence that this pattern differed between positive and negative life events (Group × Desirability × Valence: $\beta = -0.51$, $\beta$ 95% CI = −1.16, 0.15, $p = 0.130$). However, while the depression group showed largely similar updating behaviour for positive (Desirable: M 8.75, s.d. 9.38, Undesirable: M 10.26, s.d. 6.76; $t_{324} = -1.19$, $p = 0.235$) and negative (Desirable: M 10.50, s.d. 7.30, Undesirable: M 11.34, s.d. 7.80; $t_{324} = -0.66$, $p = 0.510$) life events, a differential pattern was observed in the healthy group. Participants in the healthy group showed optimistic belief updating for negative life events (Desirable: M 12.49, s.d. 7.76, Undesirable: M 7.98,

**Table 1.** Sample demographics by group. Note: percentages represent the proportion of participants within each group.

|  | healthy | depression |
|---|---|---|
| *N* | 56 | 54 |
| age, *M* (s.d.) | 37.36 (14.56) | 32.56 (12.17) |
| gender, *N* (%) |  |  |
| male | 19 (33.9) | 8 (14.8) |
| female | 36 (64.3) | 43 (79.6) |
| non-binary | 1 (1.8) | 3 (5.6) |
| ethnicity, *N* (col%) |  |  |
| White | 47 (83.9) | 46 (85.2) |
| Black | 0 (0.0) | 2 (3.7) |
| Asian | 7 (12.5) | 2 (3.7) |
| mixed | 2 (3.6) | 4 (7.4) |
| occupation, *N* (%) |  |  |
| employed | 36 (64.3) | 22 (40.7) |
| student | 8 (14.3) | 10 (18.5) |
| unemployed | 6 (10.7) | 14 (25.9) |
| other | 6 (10.7) | 8 (14.8) |
| educational attainment, *N* (%) |  |  |
| primary education | 0 (0.0) | 3 (5.6) |
| secondary education | 18 (32.1) | 24 (44.4) |
| degree | 18 (32.1) | 18 (33.3) |
| higher degree | 10 (35.7) | 9 (16.7) |
| relationship, *N* (%) |  |  |
| single | 13 (23.2) | 24 (44.4) |
| in a relationship | 42 (75.0) | 29 (53.7) |
| other | 1 (1.8) | 1 (1.9) |
| living situation, *N* (%) |  |  |
| homeowner | 30 (53.6) | 13 (24.1) |
| renting | 20 (35.7) | 23 (42.6) |
| living with a relative/friend | 6 (10.7) | 18 (33.3) |

s.d. 6.72, $t_{324} = 3.60$, $p < 0.001$), but for positive life events showed similar updating toward desirable (M 7.82, s.d. 5.50) and undesirable (M 7.82, s.d. 7.70) information ($t_{324} = 0.00$, $p = 0.998$). In combination with the wide confidence intervals, we cannot rule out the presence of a Group × Valence × Desirability interaction smaller than we have statistical power to detect.

Full descriptive statistics are provided in table 3 and analytical results are available in table 4.

**Hypothesis 3: Initial estimates of negative life events will be associated with depression, such that depressed individuals will on average initially rate their chances of experiencing a negative life event as higher compared with healthy controls.**

As hypothesized, individuals with depression initially estimated their chance of experiencing a negative life event as higher compared with healthy individuals ($\beta = 1.06$, 95% CI: 0.74, 1.38, $p < 0.001$; figure 6).

**Hypothesis 4: Initial estimates of positive life events will be associated with depression, such that depressed individuals will on average initially rate their chances of experiencing a positive life event as lower compared with healthy controls.**

In support of our hypothesis, individuals in the depression group initially estimated their chance of experiencing a positive life event as lower compared with the healthy group ($\beta = -0.71$, $\beta$ 95% CI: −1.06, −0.35, $p < 0.001$; figure 6).

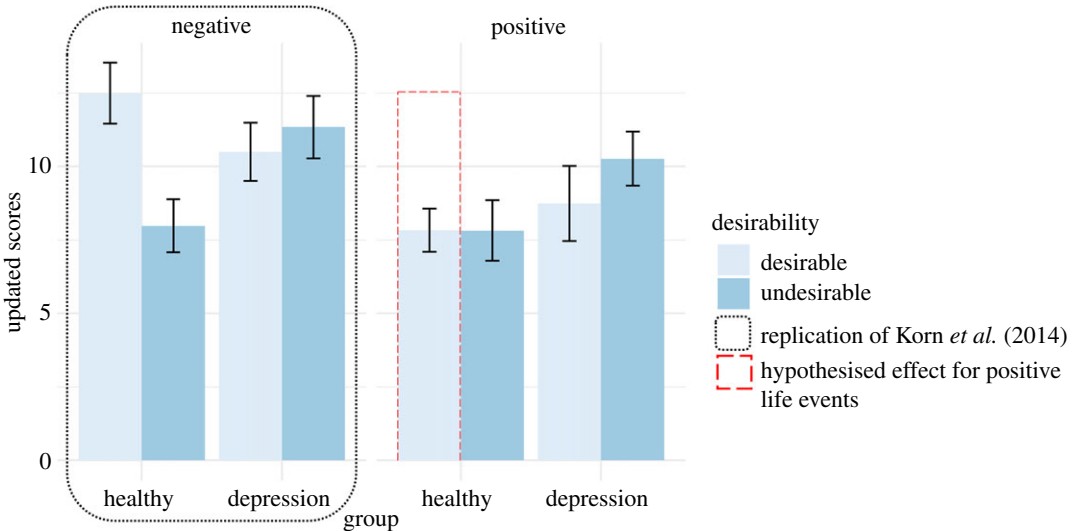

**Figure 5.** Mean update scores by group, desirability and valence. Error bars represent standard errors. For negative life events (left panel) we replicated Korn *et al.*'s [11] findings of reduced optimistic updating in individuals experiencing depression; whereas participants in the healthy group showed greater updates for desirable versus undesirable information, participants in the depression group showed similar levels of updating for desirable and undesirable information. For positive life events, we hypothesized that the effect of optimism bias would be consistent with those observed for negative life events. As illustrated by the red dashed bar, mean update scores did not support our hypothesis; while participants in the healthy group showed greater updating for desirable versus undesirable information for negative life events, they showed a similar change in beliefs for desirable and undesirable information for positive life events, that is, no optimistic belief updating for positive events.

**Table 2.** Sample psychiatric characteristics. PHQ-9 = Patient Health Questionnaire, BDI-II = Beck Depression Inventory II, GAD-7 = Generalised Anxiety Disorder Questionnaire, BFNE = Brief Fear of Negative Events Scale, LOT-R = Life Orientation Test–Revised, PANAS = Positive and Negative Affect Scale. Note: percentages represent the proportion of participants within each group. PANAS change scores represent change in scores before and after completion of the belief updating task.

|  | healthy | depression |
| --- | --- | --- |
| *N* | 56 | 54 |
| PHQ-9, M (s.d.) |  |  |
| screening | 2.27 (1.34) | 18.69 (2.87) |
| testing | 2.48 (2.13) | 17.41 (3.76) |
| BDI-II, M (s.d.) | 6.36 (5.65) | 38.11 (9.21) |
| GAD-7, M (s.d.) | 1.84 (2.19) | 13.35 (4.08) |
| BFNE, M (s.d.) | 33.70 (9.88) | 50.02 (9.25) |
| LOT-R, M (s.d.) | 16.09 (5.40) | 4.56 (4.22) |
| PANAS change, M (s.d.) |  |  |
| positive | −1.78 (5.40) | −0.96 (3.71) |
| negative | −1.64 (2.68) | −1.94 (4.18) |
| self-reported depression, *N* (%) |  |  |
| current | 0 (0%) | 46 (85.2%) |
| previous | 21 (38%) | 8 (14.8%) |
| current treatment, *N* (%) |  |  |
| psychological therapy | 0 (0%) | 12 (22.2%) |
| antidepressants | 3 (5.4%) | 22 (40.7%) |
| family history of depression, *N* (%) | 30 (53.6%) | 41 (75.9%) |

**Table 3.** Mean (standard deviation) belief updating task outcomes by group, valence of life events and desirability.

| | negative | | | | positive | | | |
| --- | --- | --- | --- | --- | --- | --- | --- | --- |
| | healthy (n = 56) | | depression (n = 54) | | healthy (n = 56) | | depression (n = 54) | |
| | desirable | undesirable | desirable | undesirable | desirable | undesirable | desirable | undesirable |
| total eligible trials | 22.12 (4.30) | 16.52 (4.39) | 26.20 (3.72) | 12.06 (3.57) | 12.29 (5.38) | 26.02 (5.49) | 17.43 (6.10) | 20.78 (6.07) |
| ineligible trials | | | | | | | | |
| estimation error of zero | 1.44 (0.69) | | 1.47 (0.70) | | 1.47 (0.73) | | 1.76 (0.91) | |
| missing estimate | 1.50 (0.97) | | 2.14 (4.00) | | 1.53 (1.17) | | 1.20 (0.41) | |
| outside specified range | 1.29 (0.76) | | 1.00 (0.00) | | 1.57 (1.40) | | 1.20 (0.42) | |
| initial estimates | 35.84 (6.68) | | 44.63 (7.44) | | 40.50 (9.30) | | 33.42 (9.45) | |
| estimation errors | 24.13 (5.00) | −22.42 (3.88) | 29.37 (5.51) | −20.64 (4.71) | −13.78 (3.61) | 30.74 (7.93) | −15.31 (3.04) | 29.50 (7.47) |
| absolute estimation errors | 24.13 (5.00) | 22.42 (3.88) | 29.37 (5.51) | 20.64 (4.71) | 13.78 (3.61) | 30.74 (7.93) | 15.31 (3.04) | 29.50 (7.47) |
| update scores | 12.49 (7.76) | 7.98 (6.72) | 10.50 (7.30) | 11.34 (7.80) | 7.82 (5.50) | 7.82 (7.70) | 8.75 (9.38) | 10.26 (6.76) |
| scaled update scores | 0.52 (0.30) | 0.36 (0.30) | 0.38 (0.28) | 0.54 (0.31) | 0.58 (0.42) | 0.30 (0.31) | 0.60 (0.60) | 0.38 (0.26) |
| absolute memory errors | 14.91 (6.44) | 14.96 (5.33) | 18.41 (6.62) | 16.52 (6.05) | 13.90 (5.96) | 16.92 (8.04) | 14.48 (5.79) | 17.44 (8.98) |
| ratings | | | | | | | | |
| controllability | 2.49 (0.64) | 2.58 (0.73) | 2.35 (0.53) | 2.33 (0.71) | 3.53 (0.90) | 4.11 (0.74) | 3.11 (0.82) | 3.69 (0.76) |
| emotional arousal | 3.06 (1.09) | 3.06 (1.08) | 3.64 (0.92) | 3.37 (1.05) | 3.18 (0.94) | 3.78 (1.04) | 3.34 (1.03) | 3.78 (0.89) |
| familiarity | 2.90 (0.85) | 2.60 (0.93) | 3.24 (0.82) | 2.72 (1.00) | 2.69 (0.94) | 3.78 (0.77) | 2.41 (0.96) | 3.44 (0.97) |
| negativity | 4.97 (0.82) | 4.98 (0.83) | 4.89 (0.65) | 4.87 (0.90) | 1.94 (0.69) | 1.51 (0.41) | 2.19 (0.94) | 1.80 (0.57) |
| positivity | 1.29 (0.30) | 1.24 (0.38) | 1.37 (0.45) | 1.33 (0.55) | 4.25 (0.91) | 4.91 (0.64) | 3.82 (1.04) | 4.57 (0.70) |
| prior experience | 2.02 (0.61) | 1.77 (0.65) | 2.50 (0.73) | 2.08 (0.84) | 2.06 (0.86) | 3.26 (0.86) | 1.67 (0.74) | 2.84 (0.95) |
| vividness | 3.20 (0.93) | 2.95 (1.07) | 3.77 (0.88) | 3.51 (0.85) | 3.19 (1.00) | 4.14 (0.78) | 3.03 (0.92) | 3.96 (0.85) |

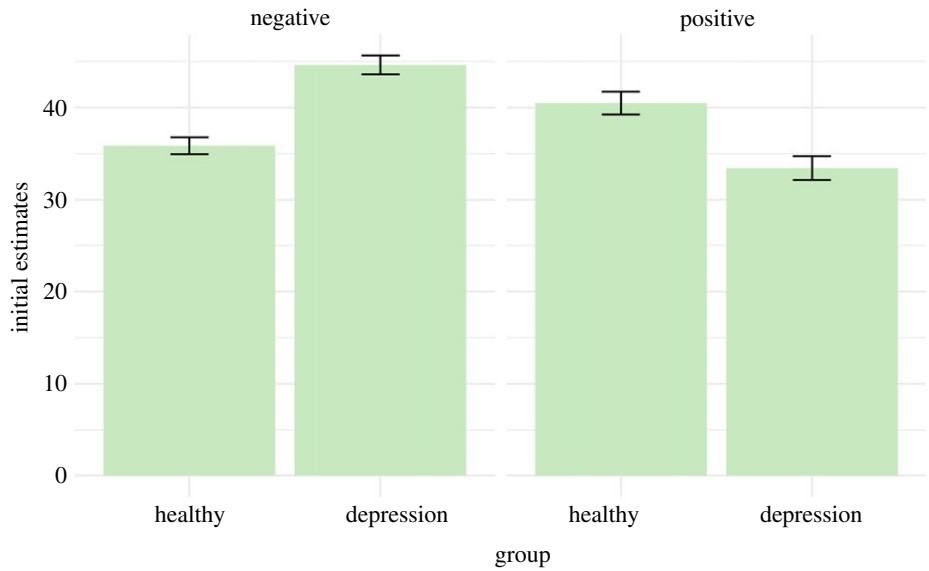

**Figure 6.** Mean initial estimates for negative and positive life events by group. Error bars represent standard errors. Participants in the depression group initially estimated their chances of experience a negative life event as greater, and chances of experiencing a positive life event as lower, than participants in the healthy group.

**Table 4.** Results from mixed-effects linear regression models for hypotheses one and two examining changes in belief updating. Reference groups: group – healthy, valence – negative, desirability – desirable.

| | $\beta$ | 95% CI | $p$ |
|---|---|---|---|
| *hypothesis 1 – negative life events* | | | |
| intercept | 0.25 | 0.00, 0.51 | <0.001 |
| group | −0.26 | −0.63, 0.10 | 0.160 |
| desirability | −0.60 | −0.93, −0.27 | 0.001 |
| group × desirability | 0.71 | 0.24, 1.18 | 0.004 |
| *hypothesis 2 – positive and negative life events* | | | |
| intercept | 0.38 | 0.12, 0.64 | <0.001 |
| group | −0.26 | −0.63, 0.10 | 0.161 |
| valence | −0.62 | −0.94, −0.29 | <0.001 |
| desirability | −0.60 | −0.92, −0.27 | <0.001 |
| group × valence | 0.39 | −0.08, 0.85 | 0.104 |
| group × desirability | 0.71 | 0.24, 1.17 | 0.003 |
| valence × desirability | 0.60 | 0.14, 1.06 | 0.011 |
| group × desirability × valence | −0.51 | −1.16, 0.15 | 0.130 |

# 8. Additional analyses

## 8.1. Scaled update scores

To evaluate whether differences in update scores between groups may be driven by differences in estimation errors, we calculated scaled update scores by dividing update scores by absolute estimation errors. Greater scaled update values indicate greater change in re-estimates relative to the magnitude of estimation errors. When analysing only negative life events we again found evidence of an interaction between group and desirability ($\beta = 1.04$, 95% CI: 0.59, 1.48, $p < 0.001$). Consistent with hypothesis 1, participants in the healthy group displayed optimistic belief updating ($t_{108} = 3.34$, $p = 0.001$), whereas participants in the depression group showed the opposite pattern ($t_{108} = −3.11$, $p = $

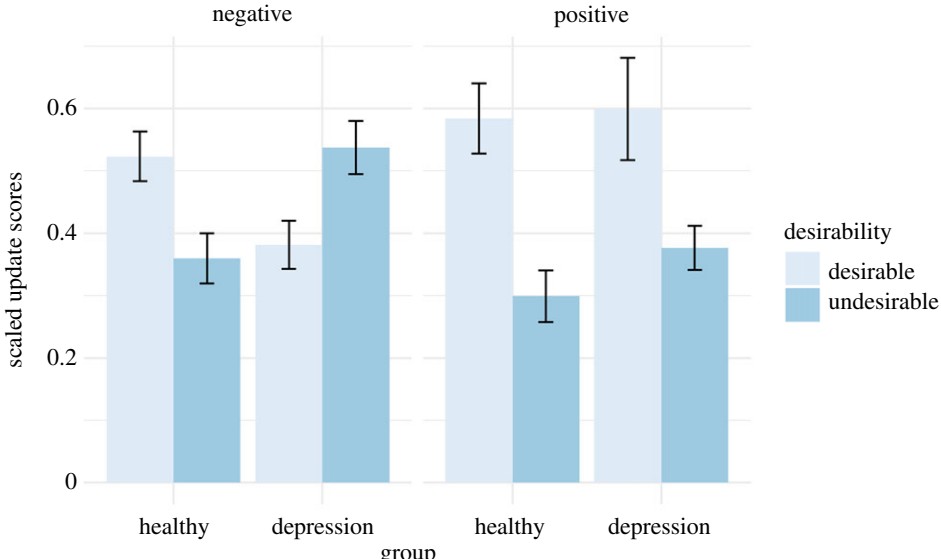

**Figure 7.** Mean scaled update scores (updates divided by estimation errors) by group, valence and desirability. Error bars represent standard errors. For negative life events (left panel), whereas participants in the healthy group showed greater updates for desirable versus undesirable information, participants in the depression group showed greater updates for undesirable versus desirable information. For positive life events (right panel), both the healthy and depression groups showed greater updates for desirable versus undesirable information.

**Table 5.** Results from mixed effects linear regression models replicating our primary analyses for update scores scaled for estimation errors. Reference groups: group – healthy, valence – negative, desirability – desirable.

|  | $\beta$ | $\beta$ 95% CI | $p$ |
|---|---|---|---|
| *hypothesis 1 – negative life events* |  |  |  |
| intercept | 0.24 | −0.02, 0.49 | <0.001 |
| group | −0.46 | −0.82, −0.10 | 0.014 |
| desirability | −0.53 | −0.84, −0.22 | 0.001 |
| group × desirability | 1.04 | 0.59, 1.48 | <0.001 |
| *hypothesis 2 – positive and negative life events* |  |  |  |
| intercept | 0.18 | −0.08, 0.43 | <0.001 |
| group | −0.38 | −0.74, −0.02 | 0.041 |
| valence | 0.16 | −0.16, 0.48 | 0.327 |
| desirability | −0.43 | −0.76, −0.11 | 0.008 |
| group × valence | 0.42 | −0.04, 0.88 | 0.075 |
| group × desirability | 0.85 | 0.39, 1.31 | <0.001 |
| valence × desirability | −0.32 | −0.78, 0.13 | 0.166 |
| group × desirability × valence | −0.68 | −1.33, −0.04 | 0.039 |

0.002; figure 7). Hypothesis 2 was not supported; when analysing both positive and negative life events, we found weak evidence that this pattern differed by valence (Group × Desirability × Valence: $\beta = -0.68$, 95% CI: −1.33, −0.04, $p = 0.039$). By contrast to hypothesis 2, both the healthy ($t_{324} = 4.62$, $p < 0.001$) and depression group ($t_{324} = 3.54$, $p = 0.001$) showed greater scaled updating toward desirable versus undesirable information for positive life events (figure 7). Full results are reported in table 5.

## 8.2. Dose–response relationship

When measuring depression symptoms using the BDI-II we found little evidence of a relationship with update scores for either positive or negative life events. When measuring depression symptoms using the

PHQ-9 we found only weak evidence of a dose–response relationship for negative life events ($\beta = 0.25$, 95% CI: 0.01, 0.49, $p = 0.046$). As depression severity increased there was an increasing overlap in confidence intervals between desirable and undesirable trials, suggesting that greater depression was associated with reduced differences in updating behaviour following desirable or undesirable information (electronic supplementary material, figure S1). We found no evidence that this effect differed by valence when analysing both positive and negative life events ($\beta = -0.12$, 95% CI: −0.45, 0.21, $p = 0.480$). Full results are reported in electronic supplementary material, table S6.

## 8.3. Memory errors

While there was some evidence that participants in the depression group had worse memory overall (Group: $\beta = 0.51$, 95% CI: 0.14, 0.88, $p = 0.007$), there was only weak evidence that this interacted with valence ($\beta = -0.43$, 95% CI: −0.85, −0.01, $p = 0.047$) and no evidence of an interaction with desirability ($\beta = -0.28$, 95% CI: −0.70, 0.14, $p = 0.187$). There was no evidence of an interaction between group, valence and desirability ($\beta = 0.28$, 95% CI: −0.32, 0.87, $p = 0.365$). When adjusting for memory errors in our primary analyses, results relating to our hypotheses were unchanged (electronic supplementary material, tables S7 and S8).

## 8.4. Trait optimism and order effects

The effects outlined above relating to our hypotheses were unchanged when adjusting for trait optimism, framing of life events, and valence order (electronic supplementary material, tables S9–S11).

## 8.5. Life event ratings

Mean ratings of life events (table 3, electronic supplementary material, figure S2) suggested some differences in perceptions of events between groups. Most notably, healthy participants tended to rate both positive and negative life events as less controllable than the depression group. Additionally, participants in the healthy group rated negative life events, on average, as less emotionally arousing, familiar and vivid, and reported less prior experience of these events, compared with the depression group. There were relatively small differences in mean ratings of negativity and positivity. Both groups rated negative events as more negative and less positive, and vice versa for positive events. However, healthy participants tended to rate positive events as slightly less negative and slightly more positive compared with the depression group. Although, when we adjusted for life event ratings our findings within our primary analyses were unchanged (electronic supplementary material, table S12).

To evaluate whether differences in our categorization of events as positive or negative versus participants' perceptions of valence may have influenced our findings, we conducted sensitivity analyses repeating our primary analyses with events classified according to each participant's negativity and positivity ratings. Results relating to our hypotheses were again unchanged suggesting that this did not account for our findings (electronic supplementary material, table S13).

We have provided the mean ratings for each life event to aid the selection of life events in future research in electronic supplementary material, table S14.

## 8.6. Adjusting for age

We did not find evidence that age influenced our findings (electronic supplementary material, table S15).

## 8.7. Scepticism over validity of life event estimates

Most participants in the healthy ($n = 32$, 58.18%) and depression groups ($n = 30$, 57.69%) disagreed with the statement 'The average probabilities in the task were accurate'. When we repeated our primary analyses excluding these participants who indicated scepticism, our results for hypotheses 1, 3 and 4 were consistent with our previous findings but effects were strengthened (electronic supplementary material, figure S3). For hypothesis 2, we did not find statistical evidence of an interaction between valence, group and desirability ($\beta = -0.69$, 95% CI: −1.57, 0.20, $p = 0.129$), but visual inspection of data indicated differences in a largely similar pattern to those previously observed. For negative life events, healthy individuals updated more toward desirable versus undesirable information whereas individuals with depression showed the opposite pattern (electronic supplementary material, figure

S3A). However, for positive life events both groups showed approximately similar levels of updating for undesirable and desirable information. Full results for all models are available in electronic supplementary material, table S16.

# 9. Discussion

When estimating the likelihood of experiencing negative life events in the future, healthy individuals are more likely to update their beliefs following desirable information (good news) versus undesirable information (bad news) [8,19,41,42] (although see [14–16,43] for critiques of this evidence and [19,44,45] for rebuttals). By contrast, individuals experiencing depression have been found to lack this optimistic belief updating bias [11,12]. We conducted the first independent replication of the effect of depression on optimistic belief updating to negative life events. Additionally, we extended previous research to examine the effect of depression on belief updating for positive life events.

## 9.1. Negative life events

This was the first attempt to independently replicate findings of reduced optimistic belief updating for negative life events in depression. In support of hypothesis 1, and replicating previously reported effects [11], we observed a pattern of belief updating in the healthy comparison group consistent with optimism bias; healthy participants changed their beliefs more following desirable versus undesirable information. By contrast, individuals experiencing depression showed a lack of biased updating, changing their beliefs to a similar extent following desirable or undesirable information. When we accounted for estimation errors, the effects of depression were even greater; healthy individuals continued to show optimistic updating whereas individuals with depression showed a pessimistic belief updating bias, changing their beliefs to a greater extent following undesirable versus desirable information. Individuals experiencing depression were also more likely to believe that they would experience a negative life event in the future. As in previous research [11], depression was, therefore, characterized by both initial pessimistic beliefs of experiencing a negative life event, as well as a reduced ability to update these beliefs following disconfirmatory positive evidence. However, we did not replicate a dose–response association between BDI-II scores and optimistic belief updating. Although we did find weak evidence using another measure of depression severity, the PHQ-9. Rather than a dose–response association, our findings instead suggest that reduced optimistic belief updating may only be apparent at higher severities of depression.

## 9.2. Positive life events

To our knowledge, this is the first study to examine the effect of depression on belief updating for positive life events. Previous research regarding optimistic belief updating for positive life events has been restricted to healthy samples, producing mixed results [16,19,20]. Additionally, research was limited by the use of simulated probabilities and shorter timeframes of reference than typically used. We attempted to address these methodological limitations by using validated probabilities for the likelihood of experiencing positive life events across one's lifetime.

We hypothesized that we would observe similar patterns of optimistic belief updating in healthy controls versus individuals experiencing depression across both positive and negative life events. However, evidence for this hypothesis was ambiguous. In support of hypothesis 2, we did not find evidence within linear mixed effect models that differences in patterns of belief updating between groups varied according to the valence of events (i.e. there was little support for the Group × Desirability × Valence interaction). However, contrary to hypothesis 2, mean update scores suggested that while the depression group showed similar updating, the healthy group displayed optimistic updating for negative events but did not show optimistic belief updating for positive events. Furthermore, when we accounted for estimation errors we did find evidence of the Group × Desirability × Valence interaction; the healthy and depression groups showed different patterns of belief updating for negative events (consistent with hypothesis 1), but both groups displayed similar optimistic updating for positive life events (contrary to hypothesis 2). Despite the mixed patterns of belief updating according to desirability, group and valence across our analyses, it is clear that the patterns of belief updating by group were not the same for positive and negative life events. On balance, therefore, we find little support for hypothesis 2. Although further research is required, our findings suggest that

reduced optimistic belief updating related to depression predominantly occurred in response to negative life events.

We also hypothesized that depressed individuals would initially rate their chances of experiencing a positive event as lower compared with healthy controls. In keeping with our hypothesis, the depression group showed reduced optimism for positive life events; providing lower estimates of experiencing positive events in their lifetime compared with the healthy group. Individuals experiencing depression, therefore, have initial reduced optimistic beliefs regarding their likelihood of experiencing positive events in the future. However, it is unclear at present from our findings how these beliefs are changed following disconfirmatory evidence.

Our results are in line with previous research in healthy volunteers, where optimistic belief updating was more clear-cut for negative and not positive life events [20]. Additionally, our findings of reduced optimistic belief updating in depression for negative but not positive life events are broadly in keeping with previous research. In a study examining interpretation biases, while individuals with greater levels of depression showed a reduced ability to change initially negative interpretations, no differences were observed for revision of initially positive interpretations [46]. Similarly, in another study that induced positive expectations of task performance, while individuals with depression showed lower initial positive expectations they did not differ from healthy controls in subsequent updating of their beliefs [47]. Differences in belief updating associated with depression, therefore, appear to be specific to initial increased negative expectations. While depression appears to be associated with cognitive inflexibility when processing emotional stimuli [48], our findings suggest that, when anticipating future life events, this is heightened for negative events. It is possible that this is driven by differences in the perceptions of positive and negative life events beyond those measured in this study. Alternatively, as processing of negative information elicits distinct brain regions to positive, rewarding information, differences associated with depression may be specific to particular neural pathways [49–51]. Future research is required to disentangle the potential mechanisms underlying the specificity of belief updating effects in depression to negative life events.

## 9.3. Clinical implications

According to cognitive theories, depression is characterized by core negative beliefs about the self, the environment, and the future [2]. In keeping with this theory, we found that individuals experiencing depression were more likely to believe that negative life events would happen and were less likely to believe that positive life events would happen. When examining how these expectations were updated following novel information, in keeping with theories of depressive realism [4], we found that the healthy controls, rather than individuals experiencing depression, displayed biased processing. Whereas healthy individuals changed their beliefs more after receiving good versus bad news about negative life events, individuals experiencing depression showed approximately equal change irrespective of the desirability of new information. Within the context of models of belief updating, our findings suggest that depressive core beliefs are maintained by reduced accommodation of positive information, preventing updating of negative beliefs even when presented with good news [52–54]. Treatments for depression could target not only core negative beliefs about the future but also how these beliefs are updated following novel information. In particular, attempting to increase accommodation of positive information into existing beliefs may be beneficial. However, our findings in combination with other research in this field suggest that reduced optimistic belief updating is currently only reliably observed for negative life events. Treatments should currently focus on maintenance of negative beliefs about the future.

Additionally, there is research suggesting that change in optimistic belief updating may be a potential predictor of change in mood disorder symptoms. In a study examining bipolar patients over a 5-year period, reduced optimistic belief updating predicted earlier relapse [55]. This has yet to be examined in patients experiencing unipolar depression but raises the possibility that measuring belief updating may allow us to identify individuals at risk of developing depression or patients at risk of relapse. Additionally, belief updating has been found to predict change in depression symptoms during treatment. In an observational study, treatment-resistant depression patients were initially found to update their beliefs following desirable or undesirable information to a similar extent. However, following a single infusion of ketamine, patients displayed an optimistic belief updating bias. Notably, this change mediated an improvement in depressive symptoms after one week of treatment [56]. Change in belief updating may, therefore, be a potential biomarker of early treatment response for

depression. Further research examining this possibility within conventional antidepressant treatments for depression may be beneficial in improving our understanding of treatment mechanisms.

## 9.4. Future research

While our study adds to evidence that depression is associated with reduced optimistic belief updating for negative life events, the cognitive mechanisms underlying this effect remain unclear. Our findings were not explained by trait optimism, recall of presented probabilities, or perceptions of the life events such as controllability, positivity, or negativity. One possibility is that initial levels of optimism influenced participants' attention toward desirable information. In a recent study, inducing optimistic expectancies regarding task performance increased attention toward rewards versus punishments [57]. Another possibility is that depression may be associated with differences in the use of cognitive immunization strategies. Cognitive immunization refers to strategies used to challenge evidence contradictory to existing beliefs, such as questioning the credibility of the source. In line with previous research [53], it is possible that individuals experiencing depression used cognitive immunization strategies to a greater extent when presented with desirable news that contradicted negative expectations. Finally, it is possible that transdiagnostic symptoms underlying a range of mental health disorders, rather than depression symptoms *per se*, may be at least partially responsible for reduced optimistic belief updating. One possible candidate may be stress. Both induced acute stress and naturalistic stress (firefighters on call) have been associated with an absence of optimistic belief updating [58]. Further exploration of potential cognitive mechanisms underlying reduced optimistic belief updating in depression would be useful in identifying sensitive targets for therapeutic intervention.

## 9.5. Limitations

Due to the COVID-19 pandemic and accompanying social restrictions we collected all data online. Data quality may, therefore, have been lower than in-person research previously conducted in this field. However, we included attention checks and had a strict exclusion criterion, removing any participant that failed a single attention check. Research has also indicated no evidence of performance differences on cognitive tasks requiring high levels of concentration between participants completing the task online versus laboratory conditions [59].

Additionally, it is possible that our recruitment methods may have led to differences in sample characteristics to those of previous studies, potentially influencing our findings. Whereas previous research recruited depression groups from clinical settings, we recruited our sample online. Psychiatric characteristics may, therefore, be expected to differ. However, levels of depression and reported treatment in our depression group were largely similar to that of previous research, limiting the possibility that this impacted our findings.

Despite our life event estimates being derived from reputable sources, a large proportion of our participants expressed scepticism as to their credibility. Participants' willingness to update their beliefs may have been influenced by perceptions of credibility. Indeed, we found our effects were strengthened when only participants that did not report scepticism were included. However, it is difficult to tell how much this may have contributed to differences in our findings to that of previous research, as to our knowledge we are the first to report data for a measure of scepticism for this task. Additionally, as described above, attempts to discount the credibility of evidence may be a potential strategy within belief updating. While participant scepticism over presented stimuli may traditionally be viewed as a limitation within cognitive studies, within this context it may be a potential mechanism associated with observed effects.

Finally, in replicating Korn *et al.* [11], we employed a widely used version of the belief updating paradigm. However, this task has recently come under increasing scrutiny. In the classic version of the task that we employed, participants are only asked to provide an estimate of their personal likelihood of experiencing life events. It has been suggested that a more appropriate way of administering the task necessitates asking participants about both their personal as well as the average person's likelihood of experiencing life events, that is, 'the base rate' [16]. This task has also been criticized for issues relating to scale attention and base rate regression [13,14] (although see [19,44] for rebuttals).

That we did not find evidence of optimistic belief updating for positive events in either people with depression or healthy controls, could arguably be taken as evidence against the existence of a universal optimistic updating bias. Similarly, our pattern of results could be taken as another sign of an underlying problem with the validity of the optimistic updating paradigm, which is currently being debated [13–16,43]. As such, reduced optimistic belief updating effects associated with depression may require

further validation after a more universally accepted measure of optimistic updating is developed. Further research examining whether effects relating to depression are replicated in future adaptations of this task (e.g. using neutral life events [43]) may help us to understand the role of belief updating in depression. Given the contentious debate over this paradigm, we would encourage researchers within this field to publish materials and data as open access to facilitate further constructive discussion.

## 9.6. Summary

In this study, we independently replicated findings of reduced optimistic belief updating for negative life events in individuals experiencing depression. This study was conducted within the format of a registered report and recruited a substantially larger sample than that of previous research. Our results, therefore, add confidence to previous findings that depression is characterized by negative future expectations maintained by a reduced ability to update these expectations in response to good news. Treatments targeting belief updating by increasing the accommodation of positive information may be beneficial in remediating depressive symptoms. However, our findings for positive life events suggest that optimistic belief updating effects are not a universal feature of human cognition. While individuals with depression were less optimistic in their belief of experiencing positive life events in the future, there was little difference in updating of these beliefs compared with healthy controls. Further research understanding the cognitive mechanisms underlying the specificity of reduced optimistic belief updating associated with depression to negative life events is required.

Ethics. The authors assert that all procedures contributing to this work comply with the ethical standards of the relevant national and institutional committees on human experimentation and with the Helsinki Declaration of 1975, as revised in 2008. All procedures involving human subjects were approved by the University of Bath Department of Psychology Ethics Committee (19-100). Written informed consent was obtained from all participants.

Data accessibility. The stage 1 protocol (https://osf.io/f2t9p), and study materials (https://osf.io/aqsrb/) are openly available on the Open Science Framework. The data and code that support the findings of this study are openly available in the University of Bath Research Data Archive: https://doi.org/10.15125/BATH-01078.

    The data are provided in electronic supplementary material [60].

Authors' contributions. C.H.: conceptualization, data curation, formal analysis, funding acquisition, investigation, methodology, project administration, resources, visualization, writing—original draft, writing—review and editing; P.V.: resources; A.S.: resources; P.S.: conceptualization, writing—review and editing; K.B.: conceptualization, funding acquisition, supervision, writing—review and editing.

    All authors gave final approval for publication and agreed to be held accountable for the work performed therein.

Competing interests. We declare we have no competing interests.

Funding. This study was funded by a GW4 BioMed MRC Doctoral Training Partnership award to Catherine Hobbs.

Acknowledgements. We would like to thank Dr Korn for their assistance in clarifying the procedure and data analysis for their research examining optimistic belief updating in depression, and Dr Janina Hoffmann for their guidance on statistical analyses. We would also like to thank the participants who took part in this study.

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
