## [Peer Review File · Royal Society Open Science]

Review History

RSOS-190814.R0 (Original submission)

Review form: Reviewer 1

Do you have any ethical concerns with this paper?

No

Have you any concerns about statistical analyses in this paper?

No

Recommendation?

Major revision

Comments to the Author(s)

The authors plan to test whether the 'update bias' for negative life events is abolished in depression, replicating Korn et al. They further plan to extend the task to study update bias for positive life events.

(i) My main concern is the extension plan to positive life events. There are serious limitations in the design stemming from the problematic choice of “positive” life events.

1. “Positive events” are not positive. While all the negative life events are clearly negative many of the ‘positive life events’ are not in fact positive. Some events may be viewed as positive by some individuals but quite terrible by others. For example, I consider the following ‘positive events’ very negative: Have 3 or more children, Owning a cat, Owning a dog, Owning a pet, Retire early (before 65), Having a baby before the age of 34, Go to an Opera, Having musical instrument lessons, Go on a caravan trip or camping trip. I will pay good money to avoid all (especially the first and last). In fact, I found most of the “positive events” mildly negative or of no interest to me. I would not like to “play a musical instrument”, “go to a national park”, “rent out 2 or more properties”, neither am I interested in “owning 2 cars” and on and on. Now, while I like skiing and running I know many people that dislike these activities. Most of the positive stimuli will raise objection.

2. No relevant base rates for positive events. It seems unlikely that the authors would have base rates for the positive events. How do the authors know what the base rate is of “going on a spontaneous holiday” or “taking part in a running race”? The authors should include a table with the base rates of all events and the source where they got the base rate. Moreover, the probability will surely alter for each subject according to their preference. That is – I like the ballet so I know I am 100% likely to go in the future. My friends hates the ballet so they know they are 0% likely to go. The base rates the authors will provide will be irrelevant to specific subjects. This is partially because of point 3 below.

3. Positive events under subject’s control, not so negative events. Almost all the positive events are very much under the subjects’ control. A person can mostly decide if they want to go visit a national park or take a holiday. Many of the negative events are much less under the subject’s control. It has recently been shown that control is critical to asymmetric learning from positive and negative outcomes (Dorfman, H.M., Bhui, R., Hughes, B.L., & Gershman, S.J- Psychological Science 2019). The authors may well be contrasting events that are under one’s control with events that are not, rather than contrasting valence. This is a serious confound.

There is a reason why the “update task” for life events has not been extended to positive events despite the original task being replicated numerous times by labs around the world. It just does not lend itself to such extension – there are not enough positive events that are positive to all and that have objective statistics that are associated with them. All the above critique has already been articulated before (18).

I would recommend the authors consider using the design of (Lefebvre et al. 2017- Nature Human Behaviour). They look at valence-dependent asymmetric learning using a basic reinforcement learning task which allows them to look at updating in response to both positive and negative outcomes about rewards (positive events) and losses (negative events). They also relate the results to optimism and have replicated the results a few times. There are other well designed tasks that allow looking at update about positive and negative stimuli (e.g., Eil & Rao, 2011) why not use one of those rather than a design (15) that has twice failed to replicate (18,19)?

(ii) I have some comments regarding the replication portion.

1. This study should be considered a “conceptual replication” rather than a pure replication as there are many differences between the proposed study and that of Korn et al. First, almost half the stimuli are different than those used originally. The authors add 35 new stimuli to the 40 Korn et al used. Half of the new stimuli are about the likelihood of dying and use the word

“death” or “mortality”. There is a huge literature about how priming people with death alters cognition and this new manipulation could have unknown effects. I think it is fine to have new stimuli, but under these conditions it can simply not be considered a pure replication and this should be clear in the abstract, intro and discussion. Moreover, the population used by Korn et al., are different to that used here. Korn et al., had a large number of hospitalized patients, many on medication. It seems the authors plan to use a non-hospitalized convenient sample. Also take note that Korn et al., tested German subjects in German. All these differences should be noted from the beginning as they may influence the results. This is not a “pure replication” as the authors suggest in page 11 line 32 (also see pg 12 line 45).

2. The authors should provide all base rates of the stimuli in the supplementary Table. A major critique of the work by Shah et al., (15) is that the base rates used induced a statistical artefact (18). Also the authors say they calculated new base rates, which are different from Korn et al. “to account for possible changes over time” - please clarify what is meant by this.

3. Power. The authors calculate that 56 subjects are needed in each group. That seems very reasonable. However, they then say that only half (23 in each group) can be considered as a replication of Korn et al because the other half will be given positive event blocks before negative event blocks and thus cannot be considered in the analysis of the replication data. This is problematic as it means the authors would be underpowered. My suggestion is to not include the positive events, which are problematic in any case. Moreover, the authors say they will re-do the analysis on subjects that meet the strict criteria used in Korn et al (pg 17), but they do not indicate which N they will use. Could it be that as a result only about 11 subjects will be left in each group? If this is a replication attempt than only subjects that meet the criteria used by Korn et al (with regards to depression or its absence) should be recruited to begin with, and the N should be 56 of those per group. (side note - Can the authors include a diagram of all blocks?).

4. Update should be calculated as movement towards the information given rather than absolute (pg 13 line 54). That is, if the initial estimate is 20 and the information is 10 and the second estimate is 30 then update is -10 not 10, because the subject is updating away from the direction of the information.

5. Different analysis plans. The authors plan to use a different analysis to that used by Korn et al., (pg 15) – linear regression rather than ANOVA. I believe that a replication should follow the exact analysis. The authors can add additional analysis if they wish. Why are there no plans to look at, and control for, the additional ratings (familiarity, arousal etc)?

General comments

The authors confuse “optimism bias” with “update bias”. The distinction between the two is important. Optimism bias is the tendency to overestimate the likelihood of positive events and underestimate the likelihood of negative events. This has already been shown to be absent in depression with both positive and negative life events (Strunk et al., 2006 – please cite). The update bias is the tendency to update beliefs more in response to information that is better than expected than information that is worse. The latter is theorized to be a mechanism generating the former.

I have noted a few sentences that need to be amended to correct for the above confusion, but I suspect there are other incidents I missed. Please correct third sentence of abstract; pg 4 line 44; pg 5 line 22 – the authors cite papers putting in question the optimism bias and the update bias but this study is about the update bias. In theory one bias may exist while the other does not – the authors should clarify which is of interest in this paper and which reference is related to which phenomena; pg 7 line 27.

Review form: Reviewer 2

Do you have any ethical concerns with this paper?

No

Have you any concerns about statistical analyses in this paper?

Yes

Recommendation?

Major revision

Comments to the Author(s)

Hobbs et al. suggest a study to thoroughly and independently replicate an article published in 2014 that assessed biased updating of beliefs about negative future events in healthy and depressed individuals. The authors plan to extend this replication in several ways, most importantly by including positive future events. A PhD student in the lab, who helped me write this review, and I agree that this extended replication would considerably advance the literatures on belief updating, optimism, and depression. Therefore, we strongly favor this registered report. Below, we provide several points for improvements on the current plan. We realize that a few of these points would equally apply to the to-be-replicated study; nevertheless we think the authors should try as much as possible to improve on this earlier study.

Major points:

- 1) Stimuli: The authors now include positive stimuli and replaced some of the negative stimuli from the earlier study (because some original stimuli did not comply with their criteria, which is fine). However, in our opinion there were several issues with the stimuli; some conceptual and some related to insufficient information. We realize that it is very difficult to compile a sufficient number of good stimuli but the points below should ideally be addressed – or at least the caveats should be mentioned.
 - o Many of the stimuli are overlapping, both for negative and positive events. That is, these events are statistically not independent. To name but a few:
 - “Being a victim of fraud, computer virus or being hacked” & “Fraud when buying something on the internet” & “Victim of computer fraud leading to loss of money”
 - “Getting married before the age of 40” & “Getting married in the summer” & “Getting married or having a civil partner”
 - “Living in a house that you own” & “Owning a house” & “Buying a home with help of friends”
 - “Earn more than £30,000 or more a year (before tax)” & “Earning £570 or more per week” are almost exact equivalents.
 - o Many of the positive events are likely to happen at least once (and some are even likely to happen multiple times). Such events should clearly be avoided since this was a major issue in earlier attempts to test biased belief updating in healthy individuals as the authors noted themselves in their introduction. Alternatively, the time horizon could be limited as has been done previously.

Events were for example:

 - “Attend a ballet performance or a dance event
 - “Attend a music concert”
 - “Go on a beach holiday”
- o Relatedly, items falling in the previous category are likely to have happened before; plus some others; e.g., items related to vacation. This could affect participants’ probability estimates.
- o Some stimuli are quite vague, e.g.,
 - “Death due to reasons considered avoidable”

o Many positive events problematic because of what we would like to dub the “inverse Anna-Karenina principle,” following Tolstoi’s statement that all happy families are alike but that unhappy families are unhappy in different ways. Here, negative events are alike for everybody but positive events are differently positive for many people. That is, almost all people want to avoid negative events equally (at least most of them; e.g., divorce may be a positive event in a deleterious relationship). But the positivity of many positive events depends on whether people actually want to experience them. Again a few examples:

- “Being admitted into Oxford University ...”
- All items related to marriage, engagement, wedding, honeymoon, children, etc.
- All items related to owning a house, a pet, etc.
- Many items related to vacations

We are not sure how to exactly deal with this issue; maybe an individual selection of events for each person (a priori or a posteriori) based on preference ratings.

o Some positive items do not seem to be positive – or only for altruistic people. Pre-ratings could help here. E.g.,

- Lend money to a friend or family member
- Doing some unpaid volunteer work

o Some positive items seem to imply a bad starting point, such that the item would only be considered good if you started off in a bad position. E.g.:

- “Earn more than £13 an hour” is not a high salary (~50% more than national minimum wage), so this would only be good if you’re current salary is below that
- The same applies to “Earn more than £30,000 or more a year (before tax).” We are not sure what the average wage is but why not choose a salary that is obviously much better than most people’s, e.g., £90,000?

o The probabilities to be shown should be included in the Table.

o Ideally the sources of the used probabilities should be included as well. The authors write that these sources were of “sufficient quality.” This should be made a bit more explicit.

o Related to the previous point, participants may hold (second-order) beliefs about the reliability or credibility with which the presented probabilities can be acquired from statistical data. This is likely to influence belief updating. Such differences in estimated reliability may also lead to important difference between negative and positive events since statistical bureaus and medical and criminal databases tend to track negative events more reliably (probably because of the “inverse Anna-Karenina principle”). One way to address this would be to ask participants or some independent group to rate perceived reliability. Alternatively, the authors could rank the reliability of their sources and the precision of the provided probabilities.

o A few items are a bit problematic because they are contingent on other events, which are also included, e.g., divorce is contingent on being married

2) Group assignment:

o PHQ-9 was not in the to-be-replicated study. This should be made clear. Can the authors provide literature on the correlation between PHQ-9 and BDI, which was used in the previous study?

o Relatedly, the to-be-replicated study recruited partly in clinical settings and assessed diagnosis in a personal interview. This is different here. Again, this should be clarified and motivated.

o Relatedly, the authors plan to conduct “sensitivity analyses” by only including participants with clinical depression in some follow up analyses. How many individuals exceeding the threshold on the PHQ-9 are expected to also fulfill the diagnostic criteria? That is, what would be the power of these analyses?

3) Power calculations:

o Why was the alpha-level set at 0.005? This is of course desirable. But it is also different to the earlier study.

- o As we understood it, the authors calculate the required power, and for the strict replication of the earlier study they use a subset of all participants – doesn't that to some degree defeat the purpose of the power analysis?
- 4) Overall length of experiment: This seems to be quite long (at least double the time of the earlier study).
 - o Especially the depressed group may suffer from concentration difficulties, fatigue, etc. Could the number of trials or ratings be reduced (which may mitigate some of the concerns mentioned above)? Could the experiment be split into two sessions on different days? Alternatively, would there be breaks?
 - o Relatedly, the authors plan to assess only participants who performed the blocks with the negative events first (to be more similar to the earlier study). What would be the power of these analyses (see before)?
 - o The repetition of blocks with 1st and 2nd estimates might alter the updating bias because participants learn over the course of the experiment that they will have to re-estimate the event probabilities. The authors could assess this.
- 5) Analyses: Overall, it is good that the authors plan to conduct LMEs and ANOVAs.
 - o We are not completely sure why comparing updates between first estimates with desirable or undesirable feedback would be circular. This approach is definitely different from selecting non-independent ROIs in neuroimaging. The calculations are just giving an updating metric scaled by the estimation error.
 - o We are not quite sure about the presented LME: Shouldn't there be a binary regressor for desirable and undesirable information (and also an interaction term for information and group)? Alternatively, a regressor with (signed) estimation errors could be included. Or two regressors: One with desirable estimation errors and zeros otherwise and another with undesirable estimation errors and zeros otherwise. This would roughly correspond to an analysis in the earlier study, in which correlations between estimation errors and updates were calculated separately for desirable and undesirable events on an individual participant level.
 - o Are there any exclusion criteria of participants with insufficient trial numbers? That is, what is the minimum number of trials per bin?
 - o What happens if updates "overshoot" (i.e., updates exceed the feedback) or go into the wrong direction? This could happen for single items or on average for some participants? In some events, this could be a result of simple typing mistakes, which could unduly influence the updating measure (or the estimates of the LMEs), especially if single trial updates are divided by single trial estimation errors.
 - o Do the authors also conduct ANOVAs with scaled absolute mean update scores?
 - o Did the authors consider including random slopes in the LMEs?
 - o Are random effects for items included in the LMEs? This could mitigate some of the concerns above.
 - o The authors could also check interactions of negative and positive events for initial estimates.

Minor points:

- 6) How will the data be made publicly available?
- 7) The terms 'probability' and 'likelihood' are often used synonymously in everyday language, but in a research paper that deals with probabilities it might be useful to use the correct statistical term. So all cases of 'likelihood' should be changed to 'probability' (at least in the technical write-up; for the text given to participants this does not necessarily apply, given that the participants probably aren't aware of the distinction)"
- 8) In the hypotheses section the authors expect a "reduced optimism bias" in depressed sample, which is completely fine and corresponds to the earlier study. But in the introduction they write mostly about an "absent optimism bias." This could be adjusted in the introduction.

- 9) The authors could also briefly introduce the hypotheses about the correlations between the initial estimates and depressive symptoms. Furthermore, the wording could be a bit briefer and more precise (e.g., correlation instead of association).
- 10) Post-ratings: How would they be used? Are they all necessary?
- 11) What is the targeted age range? This should be roughly matched to the used event stimuli.
- 12) Rewriting the section 'task procedure'. This section seems is somewhat confusing and could profit from a restructuring.
- o It might be useful to take the last sentence of that section ("This procedure is a replication...") and put it at the beginning of the section. The authors already do this in the next section ("Task outcomes" starts with "The subsequent task outcomes follow the procedure outlined by Korn et al (2014)"), so it would make the paper's writing style more consistent.
 - o It might also help to create a figure outlining the block order. First the authors talk about blocks (1,3,6,8), then about blocks (2,4,7,9), then again about (1,3,6,8) and then about the missing blocks (5,10). Visualizing this might clarify everything and not have the reader trying to piece together what happened in which order
- 13) Add labels to figure 3. Adding 'negative-desirable', 'negative-undesirable', 'positive-desirable', and 'positive-undesirable' should be a small change that makes the figure a little more obvious

Review form: Reviewer 3

Do you have any ethical concerns with this paper?

No

Have you any concerns about statistical analyses in this paper?

No

Recommendation?

Accept with minor revision

Comments to the Author(s)

I think this is a strong piece of research with a solid theoretical basis and important implications. I think the description of methodology would benefit from some clarification around the trials and the definition of desirability. In both cases, I found I had to draw some diagrams and flow charts to be able to follow the process. I didn't find Fig 3 particularly useful in terms of explaining what is described as desirability. If I have understood correctly, "desirability" refers to the extent to which the provided estimates are perceived as desirable or undesirable to the participants? If so, this isn't immediately clear and it would be useful to have this explicitly stated. I also wonder whether a diagram with the blocks might be useful - there are so many dichotomies (happen/not happen; desirable/not desirable; positive/negative) that with the counterbalancing it gets very hard to conceptualise. While I appreciate and applaud the rigorous approach, I think it would be helpful to the reader to present some of this in diagrammatic form.

I have two queries around the analysis. (1) I may have misunderstood but you state that trials are removed where the estimation error is 0 and yet the description of analysis on pg 15 (line 22-26) seems to contradict this? (2) I am unclear what the statement about removing those trials where estimates are <3% or >77% means or where in the procedure this was specified to participants (line 22, pg 14).

Otherwise, I think this is an interesting and robust piece of research.

Decision letter (RSOS-190814.R0)

24-Jun-2019

Dear Ms Hobbs,

The Editors assigned to your Stage 1 Replication submission ("Is depression associated with reduced optimism bias?") have now received comments from reviewers. We would like you to revise your paper in accordance with the referee and editors suggestions which can be found below (not including confidential reports to the Editor). Please note this decision does not guarantee eventual acceptance.

Please submit a copy of your revised paper within three weeks (i.e. by the 16-Jul-2019). If deemed necessary by the Editors, your manuscript will be sent back to one or more of the original reviewers for assessment. If the original reviewers are not available we may invite new reviewers.

When submitting your revised manuscript, you must respond to the comments made by the referees and upload a file "Response to Referees" in the "File Upload" step. Please use this to document how you have responded to the comments, and the adjustments you have made. In order to expedite the processing of the revised manuscript, please be as specific as possible in your response.

Once again, thank you for submitting your manuscript to Royal Society Open Science and I look forward to receiving your revision. If you have any questions at all, please do not hesitate to get in touch. Full author guidelines may be found at <http://rsos.royalsocietypublishing.org/page/replication-studies#AuthorsGuidance>.

Kind regards,
Professor Chris Chambers
Registered Reports Editor
Royal Society Open Science
openscience@royalsociety.org

Editor Comments to Author (Professor Chris Chambers):

Three expert reviewers have now assessed the manuscript. Two of the reviews (Rev 2 and 3) are mostly positive, noting that one or both of the Stage 1 primary criteria are already met, while one review is more negative, noting that they neither of the primary criteria are met. Reviewer 1 offers a valuable critical appraisal of the entire project, among other points noting lack of clarity in the conceptual framing of the study, and potential interpretative confusion between optimism bias and update bias. Most crucially for the Replication article type, however, the reviewer also notes areas that lack sufficient detail and which deviate substantially from the original report. These comments in particular suggest that the manuscript may fall short of primary criteria #1 and #2 (see section 3.2 in the author guidelines for details: <https://royalsocietypublishing.org/rsos/replication-studies>). Please note that only the primary criteria need be met to achieve in principle acceptance as a Replication article type.

Reviewer 2 is more positive but also notes areas of deviation. An exact replication of a study is rarely possible, and not all deviations from the original study will necessarily violate primary criterion #2. However, please take care in responding to minimise and fully justify any deviations so that a final editorial judgment can be made. In the event that the deviations are too substantial to qualify for the Replications track, it may be possible for your submission to be diverted as a regular Registered Report while preserving the current review process. As a Registered Report, a broader range of criteria must be met to achieving in principle acceptance (bringing to the fore many of the comments of Reviewer 1 that fall outside the Replications criteria) but there is also greater flexibility to deviate from the original target study. Finally Reviewer 3 offers a mostly positive assessment but also notes areas of the methodology that require additional clarification and justification.

Overall, these assessments prompt a Major Revision recommendation, focusing especially on satisfying the primary criteria for a Replication article type, while also responding as thoroughly as possible to other issues raised by the reviewers. Where recommendations from the reviewers conflict with the primary criteria (e.g. suggestions to change the design away from the target study) note that the primary criteria take precedence, unless as indicated above, the authors decide to pursue the Registered Reports track instead.

Comments to Author:

Reviewer: 1

The authors plan to test whether the 'update bias' for negative life events is abolished in depression, replicating Korn et al. They further plan to extend the task to study update bias for positive life events.

(i) My main concern is the extension plan to positive life events. There are serious limitations in the design stemming from the problematic choice of "positive" life events.

1. "Positive events" are not positive. While all the negative life events are clearly negative many of the 'positive life events' are not in fact positive. Some events may be viewed as positive by some individuals but quite terrible by others. For example, I consider the following 'positive events' very negative: Have 3 or more children, Owning a cat, Owning a dog, Owning a pet, Retire early (before 65), Having a baby before the age of 34, Go to an Opera, Having musical instrument lessons, Go on a caravan trip or camping trip. I will pay good money to avoid all (especially the first and last). In fact, I found most of the "positive events" mildly negative or of no interest to me. I would not like to "play a musical instrument", "go to a national park", "rent out 2 or more properties", neither am I interested in "owning 2 cars" and on and on. Now, while I like skiing and running I know many people that dislike these activities. Most of the positive stimuli will raise objection.
2. No relevant base rates for positive events. It seems unlikely that the authors would have base rates for the positive events. How do the authors know what the base rate is of "going on a spontaneous holiday" or "taking part in a running race"? The authors should include a table with the base rates of all events and the source where they got the base rate. Moreover, the probability will surely alter for each subject according to their preference. That is - I like the ballet so I know I am 100% likely to go in the future. My friends hates the ballet so they know they are 0% likely to go. The base rates the authors will provide will be irrelevant to specific subjects. This is partially because of point 3 below.
3. Positive events under subject's control, not so negative events. Almost all the positive events are very much under the subjects' control. A person can mostly decide if they want to go visit a national park or take a holiday. Many of the negative events are much less under the

subject's control. It has recently been shown that control is critical to asymmetric learning from positive and negative outcomes (Dorfman, H.M., Bhui, R., Hughes, B.L., & Gershman, S.J- Psychological Science 2019). The authors may well be contrasting events that are under one's control with events that are not, rather than contrasting valence. This is a serious confound.

There is a reason why the "update task" for life events has not been extended to positive events despite the original task being replicated numerous times by labs around the world. It just does not lend itself to such extension - there are not enough positive events that are positive to all and that have objective statistics that are associated with them. All the above critique has already been articulated before (18).

I would recommend the authors consider using the design of (Lefebvre et al. 2017- Nature Human Behaviour). They look at valence-dependent asymmetric learning using a basic reinforcement learning task which allows them to look at updating in response to both positive and negative outcomes about rewards (positive events) and losses (negative events). They also relate the results to optimism and have replicated the results a few times. There are other well designed tasks that allow looking at update about positive and negative stimuli (e.g., Eil & Rao, 2011) why not use one of those rather than a design (15) that has twice failed to replicate (18,19)?

(ii) I have some comments regarding the replication portion.

1. This study should be considered a "conceptual replication" rather than a pure replication as there are many differences between the proposed study and that of Korn et al. First, almost half the stimuli are different than those used originally. The authors add 35 new stimuli to the 40 Korn et al used. Half of the new stimuli are about the likelihood of dying and use the word "death" or "mortality". There is a huge literature about how priming people with death alters cognition and this new manipulation could have unknown effects. I think it is fine to have new stimuli, but under these conditions it can simply not be considered a pure replication and this should be clear in the abstract, intro and discussion. Moreover, the population used by Korn et al., are different to that used here. Korn et al., had a large number of hospitalized patients, many on medication. It seems the authors plan to use a non-hospitalized convenient sample. Also take note that Korn et al., tested German subjects in German. All these differences should be noted from the beginning as they may influence the results. This is not a "pure replication" as the authors suggest in page 11 line 32 (also see pg 12 line 45).
2. The authors should provide all base rates of the stimuli in the supplementary Table. A major critique of the work by Shah et al., (15) is that the base rates used induced a statistical artefact (18). Also the authors say they calculated new base rates, which are different from Korn et al. "to account for possible changes over time" - please clarify what is meant by this.
3. Power. The authors calculate that 56 subjects are needed in each group. That seems very reasonable. However, they then say that only half (23 in each group) can be considered as a replication of Korn et al because the other half will be given positive event blocks before negative event blocks and thus cannot be considered in the analysis of the replication data. This is problematic as it means the authors would be underpowered. My suggestion is to not include the positive events, which are problematic in any case. Moreover, the authors say they will re-do the analysis on subjects that meet the strict criteria used in Korn et al (pg 17), but they do not indicate which N they will use. Could it be that as a result only about 11 subjects will be left in each group? If this is a replication attempt than only subjects that meet the criteria used by Korn et al (with regards to depression or its absence) should be recruited to begin with, and the N should be 56 of those per group. (side note - Can the authors include a diagram of all blocks?).
4. Update should be calculated as movement towards the information given rather than absolute (pg 13 line 54). That is, if the initial estimate is 20 and the information is 10 and the second estimate is 30 then update is -10 not 10, because the subject is updating away from the direction of the information.

5. Different analysis plans. The authors plan to use a different analysis to that used by Korn et al., (pg 15) – linear regression rather than ANOVA. I believe that a replication should follow the exact analysis. The authors can add additional analysis if they wish. Why are there no plans to look at, and control for, the additional ratings (familiarity, arousal etc)?

General comments

The authors confuse “optimism bias” with “update bias”. The distinction between the two is important. Optimism bias is the tendency to overestimate the likelihood of positive events and underestimate the likelihood of negative events. This has already been shown to be absent in depression with both positive and negative life events (Strunk et al., 2006 – please cite). The update bias is the tendency to update beliefs more in response to information that is better than expected than information that is worse. The latter is theorized to be a mechanism generating the former.

I have noted a few sentences that need to be amended to correct for the above confusion, but I suspect there are other incidents I missed. Please correct third sentence of abstract; pg 4 line 44; pg 5 line 22 – the authors cite papers putting in question the optimism bias and the update bias but this study is about the update bias. In theory one bias may exist while the other does not – the authors should clarify which is of interest in this paper and which reference is related to which phenomena; pg 7 line 27.

Reviewer: 2

Comments to the Author(s)

Hobbs et al. suggest a study to thoroughly and independently replicate an article published in 2014 that assessed biased updating of beliefs about negative future events in healthy and depressed individuals. The authors plan to extend this replication in several ways, most importantly by including positive future events. A PhD student in the lab, who helped me write this review, and I agree that this extended replication would considerably advance the literatures on belief updating, optimism, and depression. Therefore, we strongly favor this registered report. Below, we provide several points for improvements on the current plan. We realize that a few of these points would equally apply to the to-be-replicated study; nevertheless we think the authors should try as much as possible to improve on this earlier study.

Major points:

1) Stimuli: The authors now include positive stimuli and replaced some of the negative stimuli from the earlier study (because some original stimuli did not comply with their criteria, which is fine). However, in our opinion there were several issues with the stimuli; some conceptual and some related to insufficient information. We realize that it is very difficult to compile a sufficient number of good stimuli but the points below should ideally be addressed – or at least the caveats should be mentioned.

o Many of the stimuli are overlapping, both for negative and positive events. That is, these events are statistically not independent. To name but a few:

□ “Being a victim of fraud, computer virus or being hacked” & “Fraud when buying something on the internet” & “Victim of computer fraud leading to loss of money”

□ “Getting married before the age of 40” & “Getting married in the summer” & “Getting married or having a civil partner”

□ “Living in a house that you own” & “Owning a house” & “Buying a home with help of friends”

□ “Earn more than £30,000 or more a year (before tax)” & “Earning £570 or more per week” are almost exact equivalents.

o Many of the positive events are likely to happen at least once (and some are even likely to happen multiple times). Such events should clearly be avoided since this was a major issue in

earlier attempts to test biased belief updating in healthy individuals as the authors noted themselves in their introduction. Alternatively, the time horizon could be limited as has been done previously.

Events were for example:

- “Attend a ballet performance or a dance event
- “Attend a music concert”
- “Go on a beach holiday”
- o Relatedly, items falling in the previous category are likely to have happened before; plus some others; e.g., items related to vacation. This could affect participants’ probability estimates.
- o Some stimuli are quite vague, e.g.,
- “Death due to reasons considered avoidable”
- o Many positive events problematic because of what we would like to dub the “inverse Anna-Karenina principle,” following Tolstoy’s statement that all happy families are alike but that unhappy families are unhappy in different ways. Here, negative events are alike for everybody but positive events are differently positive for many people. That is, almost all people want to avoid negative events equally (at least most of them; e.g., divorce may be a positive event in a deleterious relationship). But the positivity of many positive events depends on whether people actually want to experience them. Again a few examples:
- “Being admitted into Oxford University ...”
- All items related to marriage, engagement, wedding, honeymoon, children, etc.
- All items related to owning a house, a pet, etc.
- Many items related to vacations

We are not sure how to exactly deal with this issue; maybe an individual selection of events for each person (a priori or a posteriori) based on preference ratings.

- o Some positive items do not seem to be positive – or only for altruistic people. Pre-ratings could help here. E.g.,
 - Lend money to a friend or family member
 - Doing some unpaid volunteer work
 - o Some positive items seem to imply a bad starting point, such that the item would only be considered good if you started off in a bad position. E.g.:
 - “Earn more than £13 an hour” is not a high salary (~50% more than national minimum wage), so this would only be good if you’re current salary is below that
 - The same applies to “Earn more than £30,000 or more a year (before tax).” We are not sure what the average wage is but why not choose a salary that is obviously much better than most people’s, e.g., £90,000?
 - o The probabilities to be shown should be included in the Table.
 - o Ideally the sources of the used probabilities should be included as well. The authors write that these sources were of “sufficient quality.” This should be made a bit more explicit.
 - o Related to the previous point, participants may hold (second-order) beliefs about the reliability or credibility with which the presented probabilities can be acquired from statistical data. This is likely to influence belief updating. Such differences in estimated reliability may also lead to important difference between negative and positive events since statistical bureaus and medical and criminal databases tend to track negative events more reliably (probably because of the “inverse Anna-Karenina principle”). One way to address this would be to ask participants or some independent group to rate perceived reliability. Alternatively, the authors could rank the reliability of their sources and the precision of the provided probabilities.
 - o A few items are a bit problematic because they are contingent on other events, which are also included, e.g., divorce is contingent on being married
- 2) Group assignment:
- o PHQ-9 was not in the to-be-replicated study. This should be made clear. Can the authors provide literature on the correlation between PHQ-9 and BDI, which was used in the previous study?

- o Relatedly, the to-be-replicated study recruited partly in clinical settings and assessed diagnosis in a personal interview. This is different here. Again, this should be clarified and motivated.
- o Relatedly, the authors plan to conduct “sensitivity analyses” by only including participants with clinical depression in some follow up analyses. How many individuals exceeding the threshold on the PHQ-9 are expected to also fulfill the diagnostic criteria? That is, what would be the power of these analyses?
- 3) Power calculations:
 - o Why was the alpha-level set at 0.005? This is of course desirable. But it is also different to the earlier study.
 - o As we understood it, the authors calculate the required power, and for the strict replication of the earlier study they use a subset of all participants – doesn’t that to some degree defeat the purpose of the power analysis?
- 4) Overall length of experiment: This seems to be quite long (at least double the time of the earlier study).
 - o Especially the depressed group may suffer from concentration difficulties, fatigue, etc. Could the number of trials or ratings be reduced (which may mitigate some of the concerns mentioned above)? Could the experiment be split into two sessions on different days? Alternatively, would there be breaks?
 - o Relatedly, the authors plan to assess only participants who performed the blocks with the negative events first (to be more similar to the earlier study). What would be the power of these analyses (see before)?
 - o The repetition of blocks with 1st and 2nd estimates might alter the updating bias because participants learn over the course of the experiment that they will have to re-estimate the event probabilities. The authors could assess this.
- 5) Analyses: Overall, it is good that the authors plan to conduct LMEs and ANOVAs.
 - o We are not completely sure why comparing updates between first estimates with desirable or undesirable feedback would be circular. This approach is definitely different from selecting non-independent ROIs in neuroimaging. The calculations are just giving an updating metric scaled by the estimation error.
 - o We are not quite sure about the presented LME: Shouldn’t there be a binary regressor for desirable and undesirable information (and also an interaction term for information and group)? Alternatively, a regressor with (signed) estimation errors could be included. Or two regressors: One with desirable estimation errors and zeros otherwise and another with undesirable estimation errors and zeros otherwise. This would roughly correspond to an analysis in the earlier study, in which correlations between estimation errors and updates were calculated separately for desirable and undesirable events on an individual participant level.
 - o Are there any exclusion criteria of participants with insufficient trial numbers? That is, what is the minimum number of trials per bin?
 - o What happens if updates “overshoot” (i.e., updates exceed the feedback) or go into the wrong direction? This could happen for single items or on average for some participants? In some events, this could be a result of simple typing mistakes, which could unduly influence the updating measure (or the estimates of the LMEs), especially if single trial updates are divided by single trial estimation errors.
 - o Do the authors also conduct ANOVAs with scaled absolute mean update scores?
 - o Did the authors consider including random slopes in the LMEs?
 - o Are random effects for items included in the LMEs? This could mitigate some of the concerns above.
 - o The authors could also check interactions of negative and positive events for initial estimates.

Minor points:

- 6) How will the data be made publicly available?

- 7) The terms ‘probability’ and likelihood’ are often used synonymously in everyday language, but in a research paper that deals with probabilities it might be useful to use the correct statistical term. So all cases of ‘likelihood’ should be changed to ‘probability’ (at least in the technical write-up; for the text given to participants this does not necessarily apply, given that the participants probably aren’t aware of the distinction)”
- 8) In the hypotheses section the authors expect a “reduced optimism bias” in depressed sample, which is completely fine and corresponds to the earlier study. But in the introduction they write mostly about an “absent optimism bias.” This could be adjusted in the introduction.
- 9) The authors could also briefly introduce the hypotheses about the correlations between the initial estimates and depressive symptoms. Furthermore, the wording could be a bit briefer and more precise (e.g., correlation instead of association).
- 10) Post-ratings: How would they be used? Are they all necessary?
- 11) What is the targeted age range? This should be roughly matched to the used event stimuli.
- 12) Rewriting the section ‘task procedure’. This section seems is somewhat confusing and could profit from a restructuring.
- o It might be useful to take the last sentence of that section (“This procedure is a replication...”) and put it at the beginning of the section. The authors already do this in the next section (“Task outcomes” starts with “The subsequent task outcomes follow the procedure outlined by Korn et al (2014)”), so it would make the paper’s writing style more consistent.
 - o It might also help to create a figure outlining the block order. First the authors talk about blocks (1,3,6,8), then about blocks (2,4,7,9), then again about (1,3,6,8) and then about the missing blocks (5,10). Visualizing this might clarify everything and not have the reader trying to piece together what happened in which order
- 13) Add labels to figure 3. Adding ‘negative-desirable’, ‘negative-undesirable’, ‘positive-desirable’, and ‘positive-undesirable’ should be a small change that makes the figure a little more obvious

Reviewer: 3

Comments to the Author(s)

I think this is a strong piece of research with a solid theoretical basis and important implications. I think the description of methodology would benefit from some clarification around the trials and the definition of desirability. In both cases, I found I had to draw some diagrams and flow charts to be able to follow the process. I didn't find Fig 3 particularly useful in terms of explaining what is described as desirability. If I have understood correctly, "desirability" refers to the extent to which the provided estimates are perceived as desirable or undesirable to the participants? If so, this isn't immediately clear and it would be useful to have this explicitly stated. I also wonder whether a diagram with the blocks might be useful - there are so many dichotomies (happen/not happen; desirable/not desirable; positive/negative) that with the counterbalancing it gets very hard to conceptualise. While I appreciate and applaud the rigorous approach, I think it would be helpful to the reader to present some of this in diagrammatic form.

I have two queries around the analysis. (1) I may have misunderstood but you state that trials are removed where the estimation error is 0 and yet the description of analysis on pg 15 (line 22-26) seems to contradict this? (2) I am unclear what the statement about removing those trials where estimates are <3% or >77% means or where in the procedure this was specified to participants (line 22, pg 14).

Otherwise, I think this is an interesting and robust piece of research.

Author's Response to Decision Letter for (RSOS-190814.R0)

See Appendix A.

RSOS-190814.R1 (Revision)

Review form: Reviewer 1

Do you have any ethical concerns with this paper?

No

Have you any concerns about statistical analyses in this paper?

Yes

Recommendation?

Reject

Comments to the Author(s)

The revised manuscript did not address most of my original concerns. The major concern is that the manuscript does not meet Primary Criterion #2 - The manuscript does not describes a sufficiently valid (i.e. close) and robust (e.g. statistically powerful) replication of the original study methods.

The proposed study is not a replication attempt. It is a proposal for a new study, asking novel questions with new experimental conditions, new stimuli and examining a non-clinical population. It simply cannot be considered a replication attempt. Below I list the main deviations from the original study:

Major Deviations from original study:

1. New Experimental Condition: The original study examined how people with depression update their beliefs about negative life events. The new study suggests testing a new experimental variable - comparing update in response to negative life events with updating in response to positive life events. This additional variable was not a factor in the original study. Including this new variables means that the question asked is different, the design is different, the stimuli are different. The new question is indeed interesting, but it is quit simply not a replication attempt and should not be submitted via the replication path.
2. New Stimuli (new items and new associated stats - which are miscalculated and misestimated): The authors suggest using 40 new positive stimuli which were not used in the original study as well as 40 old negative stimuli which were used in the original study.

However, even for the "old stimuli" the authors plan to present new probabilities to the subjects associated with those stimuli. They claim that the reason for this is that "Unfortunately, it was not possible for us to use the baseline probabilities used in the original paper as the probabilities and their sources were not reported." Yet, in a previous study by one of the current authors (Shah) the authors "thank Tali Sharot and Christoph Korn for providing a list of the negative life events and their associated probabilities" (acknowledgements). This indicates that (i) the current authors should have the materials they need, and (ii) the original authors are willing to provide any necessary materials to the current authors. If the current authors no longer have the materials previously provided to them by the original authors, have they attempted to request these materials again from the original authors?

The journal clearly states the following:

"In some cases, the published methods of the original study may not provide enough detail to permit a close replication (Stage 1 Primary Criterion #1), and the original authors of the study may also be unwilling or unable to provide the necessary missing information. In such cases, estimates of the original procedures will be accepted provided the authors can establish that the

chosen method is a reasonable estimation of the original approach, and provided (where applicable) they can supply documentary evidence to the editor, such as emails, proving any attempts to obtain such information from the original authors."

The authors then provide an additional reason for using new statistics "Furthermore, in their paper Korn et al specify that life event probabilities must fall within 10% to 70% to allow for sufficient movement in re-estimates. This prevented us from including a number of life events used in the original paper, as they were rarer or more common than these specified probabilities." Are the authors suggesting that the original authors falsely claim that they are using events with probabilities between 10-70 when in fact they were not? Or perhaps different sources provide different estimates and the current authors are using different sources than the original authors?

While I strongly suggest the authors simply obtain the original statistics from the original authors, as I explain below that the current authors have miscalculated and misestimated the probabilities associated with the stimuli in Supplementary Table 2 and elsewhere.

The question the subject is asked is "How likely is this event to happen to you in the future". This means that the associated probability needs to reflect the likelihood of a condition (lets say Alzheimer's) happening to a person in their lifetime (i.e. anytime in the next 50 years or so) and not the current % of people suffering from a condition.

Lets take Alzheimer's - the current authors claims that the likelihood of a person suffering from Alzheimer's anytime in the future is 0.68%. Yet "According to the Alzheimer's Association, 10% of all people over the age of 65 have Alzheimer's disease, and as many as 50% of people over 85 have it." <https://www.alz.org/alzheimers-dementia/facts-figures>

The authors employ the same miscalculation for almost every stimulus. Here is another example - they claim the likelihood of ever being fired from your job is 1.5%. However, the source they attach to this number suggests this is the number of currently unemployed individuals. This is not the same as the probability of being fired over your life-time (most subjects have about 40 more years of work years). Moreover, being unemployed is not the same as being fired, as one can be fired but still be employed elsewhere.

Moreover, many of the sources they provide have nothing to do with the associated stimulus they claim. For example, they claim the likelihood of fraud when buying something on the internet is 6% (see Supp Table 2) - as their source they cite articles about Hepatitis B. Online fraud b.t.w is about 70% over your life time according to some sources. For domestic burglary - they cite an epilepsy paper. They claim it is 2.3% when in fact it is 30% in London (where the very first update study was conducted). For epilepsy they cite a heart failure paper. For heart failure they suggest the likelihood is 1.39% when in fact it is 20% according to www.heart.org ("Of all adults 40 and older, one in five Americans will develop heart failure in their lifetime") and as their source they cite a paper about hospital injuries. And so on and so forth.

3. Non-Clinical Population. The original study examines individuals that have been clinically diagnosed with depression "Participants were assessed for psychiatric disorders using a structured clinical interview (SCID-I; Wittchen et al. 1997) by a cognitive neuroscientist (C.W.K.), who had been trained by a psychotherapist in conducting the SCID-I." (Korn et al., 2014). The subjects had sever, clinically diagnosed, depression - an average BDI of 32.6 with an SD 7.69 (for those not familiar with the BDI - these numbers reflect sever depression). The controls had a BDI of 4.3 (SD 3.56). In contrast, the current authors have no plans to conduct a clinical diagnosis. Moreover they suggest also recruiting subjects with only moderate depression (PHQ-9 > 10). This is a critical deviation. The original study examined a clinical population that were carefully

diagnosed as having clinical depression. The “replication” suggests examining a convenient sample that are not clinically diagnosed with depression. Whether the subjects have clinical depression is not a minor deviation in the characteristics of the population - it is the major point of the whole study.

Insufficient Power

1. 57% of number of original trials. The authors suggest using 57% of the number of trails used in the original study (the original study included 70 negative events and the current study only 40). This means power is greatly reduced.
2. Incorrect power calculation in justifying N, resulting in insufficient power. The authors start by claiming their power is 95% with 108 participants, but that in fact only 89 are likely to meet criteria so power will be 89% for 89 subjects. Then they say that in fact only half of 108 will undergo the same procedure as in Korn et al thus 54 subjects (27 in each group) can be considered a replication which will give 86% power. However, they do not take into account that out of those 54 not all will reach criteria. They say in fact that only 75% will reach criteria. That means only 40 subjects (20 in each group) will be considered in the replication. I do not think that will provide enough power. The original study had 19 subjects in one group and 18 in the other. Given the publication bias I assume they will need significantly more than 20 in each group. My suggestion is - first provide the calculation of how many is needed in each group for a set power (lets say 85%). Then double that number (as only half the subjects are planned for replication)- that number will reflect $\frac{3}{4}$ of the number you actually need due to those not meeting criteria.

Other problems

The rest of my concerns are related to the new experimental design and stimuli. This relates to the portion of the study that is not meant to be a replication but meant to be a new design (which as mentioned above makes this submission inappropriate in the first place).

1. The authors claim that inserting a confound of “controllability” will provide new information. The confound will not provide new information as it covaries with valence, thus one will not be able to conclude anything about valence nor about controllability
2. The authors claim they will account for controllability and the fact that “positive events” are not necessarily positive in their analysis. However, no details about the new analysis is provided. They only offer a vague sentence on page 44, with no sufficient details about statistical analysis plan.
3. The authors claim they will include the original statistical approach - using an ANOVA - yet no such plan has been added to the manuscript.

Review form: Reviewer 2

Do you have any ethical concerns with this paper?

No

Have you any concerns about statistical analyses in this paper?

No

Recommendation?

Accept with minor revision

Comments to the Author(s)

We thank Hobbs et al. for responding to our comments, especially for providing additional details on stimuli and analyses. Most of our points have been addressed. Below you will find a list of some remaining issues, which are almost all about the selection of positive events.

- 1) Positive Stimuli (previously major point #1): The sub-selection of 40 (out of 70) items improved the stimuli.

- a. Some items still overlap (or are contingent on each other): E.g., many events relate to marriage or vacation. Could this be reduced further? Alternatively, could these events be clustered in LME analyses?
- b. Positive events that are rated as negative: Ratings are a good way of addressing the issue that positive words may not be regarded as positive by all participants. We are just not sure whether the authors would treat these events as “negative” or whether they would exclude these events altogether. It sounds as if the authors think that including these ratings as a covariate would mitigate all individual differences, which may not be the case if most of the events are rated as negative by some participants. That is, depressed participants might be more likely to rate “positive” events as negative.
- c. Problematic statistics: The following issues might reduce the credibility of the provided statistics. Maybe part of the solution will be to rate the credibility of the statistics (per item or at least overall)
- i. Overlapping events and problematic extrapolation: It seems weird that “celebrating your diamond wedding anniversary” has a probability of 60% and “getting married” has a probability of 66%. Will only 6% get divorced or die? Probably, data for the diamond anniversary are based on people who got married & 60 years ago. These were very different times. So a simple extrapolation is not possible. (Admittedly, one might say that probabilities for many age-related might get lower within the next 20-40 years). Also, it seems weird that 23% will live abroad but only 13% will go on a holiday abroad for & 2 weeks.
- ii. Specific example 1: The authors say that the probability of ‘celebrating your 20th wedding anniversary’ is 60%. We think they calculated it the following way: according to (<https://web.archive.nationalarchives.gov.uk/20160106011951/http://www.ons.gov.uk/ons/rel/vsob1/divorces-in-england-and-wales/2011/sty-what-percentage-of-marriages-end-in-divorce.html>), which we got from the document they cited, the probability of marriage ending because one spouse has died is around 5%, and the probability of marriage ending in divorce is around 35% at the 20-year mark. It seems as if they added those two numbers together and then subtracted that from 100%, meaning that the 60% probability of celebrating one’s 20th wedding anniversary is only true if everyone (or close to everyone) got married. Is that true though? If not, then maybe it should be made clear that this figure of 60% only applies to people who have gotten married in the first place.
- iii. Specific example 2: The probability of celebrating one’s 70th birthday is given as 14%, but the probability of celebrating one’s diamond (60 years) wedding anniversary is given at 16% - so, for the sake of the argument, if we assume that people get married on average at age 10, then at age 70 you’re more likely to be married than alive.)
- d. Problem with context-dependence of positive life events: it’s still not clear what the context is for the positive items, and this can have grave effects on how many items people will rate as positive. For example: ‘Be mortgage free before turning 55’ really depends on the context: if you’re 54 and mortgage free because you’ve bought a house and paid it off, then that’s great; but if you’re 54 and mortgage free because you’ve never been able to afford a house in the first place, then that’s not so great. So the question is: what exactly is the scenario under which I’m rating these items? Similarly, ‘live abroad’ can be great if you’ve moved abroad on your own wish, but might be terrible if e.g., you’re a refugee or you’ve moved abroad because your spouse accepted a job offer abroad and you feel forced to follow (even though you want to live close to your family). Owning 2 cars or more is great if you can afford it, but the (largely) student sample the authors plan to recruit probably can’t afford 2 cars, so they’d have to sell them. So, depending on the context, the same person will rate the same items differently and depending on the context under which participants choose to make their ratings, a lot of the items might be rated as not-positive, meaning that the authors would be left with few data points to use for the positive items (on a personal glance, I’d probably rate most items as neutral, simply because I don’t know the context in which they’ll take place)
- e. Vague items:

- i. 'Be mortgage free before turning 55': anyone who is younger than 55 and doesn't have a mortgage (i.e., most students) is currently mortgage free
- ii. 'Fall in love with someone you met on a dating site': the emphasis of this item seems to be the dating site, rather than falling in love (otherwise, why mention the dating site?). To me, the whereabouts of falling in love are somewhat secondary to the falling in love itself and I wouldn't care much whether I fell in love on or off a dating site, so I'd probably rate this items as neutral
- iii. 'Hold an investment': an investment doesn't have to entail a lot of money, so this item is somewhat empty without specifying 1) that that investment is (at least for the person) large, and 2) not depreciating rapidly
- f. Reference population: E.g., a probability of 25% for getting engaged on Valentine's Day seems rather high. Is this with respect to all people, i.e., also those who never get engaged, or only with respect to those who get engaged. This has to be clear for several events. Copying mistake in Supplementary Table 2: Something is wrong on page 55. Sources do not fit the events (unless e.g., the British Heart Foundation really cares about epilepsy, while Epilepsy Action really cares about domestic burglary).
- 2) Replication versus new data: Maybe it would be helpful to include a Table listing the differences (and communalities) with the previous work by Korn et al (including recruitment, analyses, inference criteria, etc.). This could also help future readers to appreciate the new aspects of the current work and to assess which effects could be replicated and which not.

Decision letter (RSOS-190814.R1)

10-Sep-2019

Dear Ms Hobbs,

The Editors assigned to your Stage 1 Registered Report ("Is depression associated with reduced optimistically biased belief updating?") have now received comments from reviewers. We would like you to revise your paper in accordance with the referee and editors suggestions which can be found below (not including confidential reports to the Editor). Please note this decision does not guarantee eventual acceptance.

When submitting your revised manuscript, you must respond to the comments made by the referees and upload a file "Response to Referees". Please use this to document how you have responded to the comments, and the adjustments you have made. In order to expedite the processing of the revised manuscript, please be as specific as possible in your response.

Kind regards,
 Andrew Dunn
 Senior Publishing Editor
 Royal Society Open Science

on behalf of Professor Chris Chambers (Registered Reports Editor, Royal Society Open Science)
openscience@royalsociety.org

Associate Editor Comments to Author (Professor Chris Chambers):

Associate Editor: 1

Comments to the Author:

The revised manuscript was returned to two of the original reviewers. As you will see the reviews are polarised, with Reviewer 1 recommending rejection and Reviewer 2 recommended acceptance following minor revision. Some of Reviewer 1's concerns arise from deviations from the original study protocol, and as noted in my previous decision letter, transferring the article type to the Registered Reports track (as the authors have done), means that the authors are no longer subject to the Replications criteria. However, the reviewer also raises concerns about many of the extended elements of the design, which are pertinent will need to be addressed through comprehensive revision or rebuttal. Reviewer 2 is more positive and notes some areas requiring further attention, primarily in the selection of positive life events.

This submission has presented a number of challenges, both in terms of polarised reviews and also the transfer between article types. On balance, given the positive appraisals of Reviewer 2 (and Reviewer 3 from the previous round), I am convinced that the study has potential merit, provided the concerns that relate specifically to the study as a Registered Report (as opposed to a direct Replication) can be addressed. Therefore I invite the authors to submit a final revision. To avoid unnecessarily prolonging the review process, I will make a final Stage 1 editorial decision to accept or reject following the next submission.

Comments to Author:

Reviewer: 1

Comments to the Author(s)

The revised manuscript did not address most of my original concerns. The major concern is that the manuscript does not meet Primary Criterion #2 - The manuscript does not describes a sufficiently valid (i.e. close) and robust (e.g. statistically powerful) replication of the original study methods.

The proposed study is not a replication attempt. It is a proposal for a new study, asking novel questions with new experimental conditions, new stimuli and examining a non-clinical population. It simply cannot be considered a replication attempt. Below I list the main deviations from the original study:

Major Deviations from original study:

1. New Experimental Condition: The original study examined how people with depression update their beliefs about negative life events. The new study suggests testing a new experimental variable - comparing update in response to negative life events with updating in response to positive life events. This additional variable was not a factor in the original study. Including this new variables means that the question asked is different, the design is different, the stimuli are different. The new question is indeed interesting, but it is quit simply not a replication attempt and should not be submitted via the replication path.
2. New Stimuli (new items and new associated stats - which are miscalculated and misestimated): The authors suggest using 40 new positive stimuli which were not used in the original study as well as 40 old negative stimuli which were used in the original study.

However, even for the "old stimuli" the authors plan to present new probabilities to the subjects associated with those stimuli. They claim that the reason for this is that "Unfortunately, it was not possible for us to use the baseline probabilities used in the original paper as the probabilities and

their sources were not reported." Yet, in a previous study by one of the current authors (Shah) the authors "thank Tali Sharot and Christoph Korn for providing a list of the negative life events and their associated probabilities" (acknowledgements). This indicates that (i) the current authors should have the materials they need, and (ii) the original authors are willing to provide any necessary materials to the current authors. If the current authors no longer have the materials previously provided to them by the original authors, have they attempted to request these materials again from the original authors?

The journal clearly states the following:

"In some cases, the published methods of the original study may not provide enough detail to permit a close replication (Stage 1 Primary Criterion #1), and the original authors of the study may also be unwilling or unable to provide the necessary missing information. In such cases, estimates of the original procedures will be accepted provided the authors can establish that the chosen method is a reasonable estimation of the original approach, and provided (where applicable) they can supply documentary evidence to the editor, such as emails, proving any attempts to obtain such information from the original authors."

The authors then provide an additional reason for using new statistics "Furthermore, in their paper Korn et al specify that life event probabilities must fall within 10% to 70% to allow for sufficient movement in re-estimates. This prevented us from including a number of life events used in the original paper, as they were rarer or more common than these specified probabilities." Are the authors suggesting that the original authors falsely claim that they are using events with probabilities between 10-70 when in fact they were not? Or perhaps different sources provide different estimates and the current authors are using different sources than the original authors?

While I strongly suggest the authors simply obtain the original statistics from the original authors, as I explain below that the current authors have miscalculated and misestimated the probabilities associated with the stimuli in Supplementary Table 2 and elsewhere.

The question the subject is asked is "How likely is this event to happen to you in the future". This means that the associated probability needs to reflect the likelihood of a condition (lets say Alzheimer's) happening to a person in their lifetime (i.e. anytime in the next 50 years or so) and not the current % of people suffering from a condition.

Lets take Alzheimer's - the current authors claims that the likelihood of a person suffering from Alzheimer's anytime in the future is 0.68%. Yet "According to the Alzheimer's Association, 10% of all people over the age of 65 have Alzheimer's disease, and as many as 50% of people over 85 have it." <https://www.alz.org/alzheimers-dementia/facts-figures>

The authors employ the same miscalculation for almost every stimulus. Here is another example - they claim the likelihood of ever being fired from your job is 1.5%. However, the source they attach to this number suggests this is the number of currently unemployed individuals. This is not the same as the probability of being fired over your life-time (most subjects have about 40 more years of work years). Moreover, being unemployed is not the same as being fired, as one can be fired but still be employed elsewhere.

Moreover, many of the sources they provide have nothing to do with the associated stimulus they claim. For example, they claim the likelihood of fraud when buying something on the internet is 6% (see Supp Table 2) - as their source they cite articles about Hepatitis B. Online fraud b.t.w is about 70% over your life time according to some sources. For domestic burglary - they cite an epilepsy paper. They claim it is 2.3% when in fact it is 30% in London (where the very

first update study was conducted). For epilepsy they cite a heart failure paper. For heart failure they suggest the likelihood is 1.39% when in fact it is 20% according to www.heart.org (“Of all adults 40 and older, one in five Americans will develop heart failure in their lifetime”) and as their source they cite a paper about hospital injuries. And so on and so forth.

3. Non-Clinical Population. The original study examines individuals that have been clinically diagnosed with depression “Participants were assessed for psychiatric disorders using a structured clinical interview (SCID-I; Wittchen et al. 1997) by a cognitive neuroscientist (C.W.K.), who had been trained by a psychotherapist in conducting the SCID-I.” (Korn et al., 2014). The subjects had severe, clinically diagnosed, depression - an average BDI of 32.6 with an SD 7.69 (for those not familiar with the BDI - these numbers reflect severe depression). The controls had a BDI of 4.3 (SD 3.56). In contrast, the current authors have no plans to conduct a clinical diagnosis. Moreover they suggest also recruiting subjects with only moderate depression (PHQ-9 > 10). This is a critical deviation. The original study examined a clinical population that were carefully diagnosed as having clinical depression. The “replication” suggests examining a convenient sample that are not clinically diagnosed with depression. Whether the subjects have clinical depression is not a minor deviation in the characteristics of the population - it is the major point of the whole study.

Insufficient Power

1. 57% of number of original trials. The authors suggest using 57% of the number of trials used in the original study (the original study included 70 negative events and the current study only 40). This means power is greatly reduced.
2. Incorrect power calculation in justifying N, resulting in insufficient power. The authors start by claiming their power is 95% with 108 participants, but that in fact only 89 are likely to meet criteria so power will be 89% for 89 subjects. Then they say that in fact only half of 108 will undergo the same procedure as in Korn et al thus 54 subjects (27 in each group) can be considered a replication which will give 86% power. However, they do not take into account that out of those 54 not all will reach criteria. They say in fact that only 75% will reach criteria. That means only 40 subjects (20 in each group) will be considered in the replication. I do not think that will provide enough power. The original study had 19 subjects in one group and 18 in the other. Given the publication bias I assume they will need significantly more than 20 in each group. My suggestion is - first provide the calculation of how many is needed in each group for a set power (lets say 85%). Then double that number (as only half the subjects are planned for replication)- that number will reflect $\frac{3}{4}$ of the number you actually need due to those not meeting criteria.

Other problems

The rest of my concerns are related to the new experimental design and stimuli. This relates to the portion of the study that is not meant to be a replication but meant to be a new design (which as mentioned above makes this submission inappropriate in the first place).

1. The authors claim that inserting a confound of “controllability” will provide new information. The confound will not provide new information as it covaries with valence, thus one will not be able to conclude anything about valence nor about controllability
2. The authors claim they will account for controllability and the fact that “positive events” are not necessarily positive in their analysis. However, no details about the new analysis is provided. They only offer a vague sentence on page 44, with no sufficient details about statistical analysis plan.
3. The authors claim they will include the original statistical approach - using an ANOVA - yet no such plan has been added to the manuscript.

Reviewer: 2

Comments to the Author(s)

We thank Hobbs et al. for responding to our comments, especially for providing additional details on stimuli and analyses. Most of our points have been addressed. Below you will find a list of some remaining issues, which are almost all about the selection of positive events.

- 1) Positive Stimuli (previously major point #1): The sub-selection of 40 (out of 70) items improved the stimuli.
 - a. Some items still overlap (or are contingent on each other): E.g., many events relate to marriage or vacation. Could this be reduced further? Alternatively, could these events be clustered in LME analyses?
 - b. Positive events that are rated as negative: Ratings are a good way of addressing the issue that positive words may not be regarded as positive by all participants. We are just not sure whether the authors would treat these events as “negative” or whether they would exclude these events altogether. It sounds as if the authors think that including these ratings as a covariate would mitigate all individual differences, which may not be the case if most of the events are rated as negative by some participants. That is, depressed participants might be more likely to rate “positive” events as negative.
 - c. Problematic statistics: The following issues might reduce the credibility of the provided statistics. Maybe part of the solution will be to rate the credibility of the statistics (per item or at least overall)
 - i. Overlapping events and problematic extrapolation: It seems weird that “celebrating your diamond wedding anniversary” has a probability of 60% and “getting married” has a probability of 66%. Will only 6% get divorced or die? Probably, data for the diamond anniversary are based on people who got married > 60 years ago. These were very different times. So a simple extrapolation is not possible. (Admittedly, one might say that probabilities for many age-related might get lower within the next 20-40 years). Also, it seems weird that 23% will live abroad but only 13% will go on a holiday abroad for > 2 weeks.
 - ii. Specific example 1: The authors say that the probability of ‘celebrating your 20th wedding anniversary’ is 60%. We think they calculated it the following way: according to (<https://webarchive.nationalarchives.gov.uk/20160106011951/http://www.ons.gov.uk/ons/rel/vsob1/divorces-in-england-and-wales/2011/sty-what-percentage-of-marriages-end-in-divorce.html>), which we got from the document they cited, the probability of marriage ending because one spouse has died is around 5%, and the probability of marriage ending in divorce is around 35% at the 20-year mark. It seems as if they added those two numbers together and then subtracted that from 100%, meaning that the 60% probability of celebrating one’s 20th wedding anniversary is only true if everyone (or close to everyone) got married. Is that true though? If not, then maybe it should be made clear that this figure of 60% only applies to people who have gotten married in the first place.
 - iii. Specific example 2: The probability of celebrating one’s 70th birthday is given as 14%, but the probability of celebrating one’s diamond (60 years) wedding anniversary is given at 16% - so, for the sake of the argument, if we assume that people get married on average at age 10, then at age 70 you’re more likely to be married than alive.)
 - d. Problem with context-dependence of positive life events: it’s still not clear what the context is for the positive items, and this can have grave effects on how many items people will rate as positive. For example: ‘Be mortgage free before turning 55’ really depends on the context: if you’re 54 and mortgage free because you’ve bought a house and paid it off, then that’s great; but if you’re 54 and mortgage free because you’ve never been able to afford a house in the first place, then that’s not so great. So the question is: what exactly is the scenario under which I’m rating these items? Similarly, ‘live abroad’ can be great if you’ve moved aboard on your own wish, but might be terrible if e.g., you’re a refugee or you’ve moved abroad because your spouse accepted a job offer abroad and you feel forced to follow (even though you want to live close to your family). Owning 2 cars or more is great if you can afford it, but the (largely) student sample the authors plan to recruit probably can’t afford 2 cars, so they’d have to sell them. So, depending on the context, the same person will rate the same items differently and depending on the context under which participants choose to make their ratings, a lot of the items might be rated as not-positive, meaning that the authors would be left with few data points to use for the positive items (on a personal glance, I’d probably rate most items as neutral, simply because I don’t know the context in which they’ll take place)

e. Vague items:

i. 'Be mortgage free before turning 55': anyone who is younger than 55 and doesn't have a mortgage (i.e., most students) is currently mortgage free

ii. 'Fall in love with someone you met on a dating site': the emphasis of this item seems to be the dating site, rather than falling in love (otherwise, why mention the dating site?). To me, the whereabouts of falling in love are somewhat secondary to the falling in love itself and I wouldn't care much whether I fell in love on or off a dating site, so I'd probably rate this items as neutral

iii. 'Hold an investment': an investment doesn't have to entail a lot of money, so this item is somewhat empty without specifying 1) that that investment is (at least for the person) large, and 2) not depreciating rapidly

f. Reference population: E.g., a probability of 25% for getting engaged on Valentine's Day seems rather high. Is this with respect to all people, i.e., also those who never get engaged, or only with respect to those who get engaged. This has to be clear for several events. Copying mistake in Supplementary Table 2: Something is wrong on page 55. Sources do not fit the events (unless e.g., the British Heart Foundation really cares about epilepsy, while Epilepsy Action really cares about domestic burglary).

2) Replication versus new data: Maybe it would be helpful to include a Table listing the differences (and communalities) with the previous work by Korn et al (including recruitment, analyses, inference criteria, etc.). This could also help future readers to appreciate the new aspects of the current work and to assess which effects could be replicated and which not.

Author's Response to Decision Letter for (RSOS-190814.R1)

See Appendix B.

Decision letter (RSOS-190814.R2)

01-Nov-2019

Dear Ms Hobbs,

On behalf of the Editor, I am pleased to inform you that your Stage 1 Registered REport RSOS-190814.R2 entitled "Is depression associated with reduced optimistic belief updating?" has been accepted in principle for publication in Royal Society Open Science.

You may now progress to Stage 2 and complete the study as approved. Before commencing data collection we ask that you:

1) Update the journal office as to the anticipated completion date of your study.

2) Register your approved protocol on the Open Science Framework (<https://osf.io/rr>) or other recognised repository, either publicly or privately under embargo until submission of the Stage 2 manuscript. Please note that a time-stamped, independent registration of the protocol is mandatory under journal policy, and manuscripts that do not conform to this requirement cannot be considered at Stage 2. The protocol should be registered unchanged from its current approved state, with the time-stamp preceding implementation of the approved study design.

Following completion of your study, we invite you to resubmit your paper for peer review as a

Stage 2 Registered Report. Please note that your manuscript can still be rejected for publication at Stage 2 if the Editors consider any of the following conditions to be met:

- The results were unable to test the authors' proposed hypotheses by failing to meet the approved outcome-neutral criteria.
- The authors altered the Introduction, rationale, or hypotheses, as approved in the Stage 1 submission.
- The authors failed to adhere closely to the registered experimental procedures. Please note that any deviations from the approved experimental procedures must be communicated to the editor immediately for approval, and prior to the completion of data collection. Failure to do so can result in revocation of in-principle acceptance and rejection at Stage 2 (see complete guidelines for further information).
- Any post-hoc (unregistered) analyses were either unjustified, insufficiently caveated, or overly dominant in shaping the authors' conclusions.
- The authors' conclusions were not justified given the data obtained.

We encourage you to read the complete guidelines for authors concerning Stage 2 submissions at <https://royalsocietypublishing.org/rsos/registered-reports#ReviewerGuideRegRep>. Please especially note the requirements for data sharing, reporting the URL of the independently registered protocol, and that withdrawing your manuscript will result in publication of a Withdrawn Registration.

Please note that Royal Society Open Science will introduce article processing charges for all new submissions received from 1 January 2018. Registered Reports submitted and accepted after this date will ONLY be subject to a charge if they subsequently progress to and are accepted as Stage 2 Registered Reports. If your manuscript is submitted and accepted for publication after 1 January 2018 (i.e. as a full Stage 2 Registered Report), you will be asked to pay the article processing charge, unless you request a waiver and this is approved by Royal Society Publishing. You can find out more about the charges at <http://rsos.royalsocietypublishing.org/page/charges>. Should you have any queries, please contact openscience@royalsociety.org.

Once again, thank you for submitting your manuscript to Royal Society Open Science and we look forward to receiving your Stage 2 submission. If you have any questions at all, please do not hesitate to get in touch. We look forward to hearing from you shortly with the anticipated submission date for your stage two manuscript.

Kind regards,
Lianne Parkhouse
Royal Society Open Science
openscience@royalsociety.org

on behalf of Professor Chris Chambers (Registered Reports Editor, Royal Society Open Science)
openscience@royalsociety.org

Author's Response to Decision Letter for (RSOS-190814.R2)

See Appendix C.

RSOS-190814.R3

Review form: Reviewer 2

Do you have any ethical concerns with this paper?

No

Recommendation?

Accept with minor revision

Comments to the Author(s)

It is unfortunate that the authors could not collect the data due to COVID-19. The PhD student helping with the review and I think that it is a good approach to adjust the original plan and collect data online. Below are a few comments for improvement. Given potential time constraints, we do not necessarily need to review the authors' replies.

1. The authors could include parts of their cover letter into the manuscript to state up-front why and how they adjust their original plan (e.g., they could mention in the abstract that this is an online study). This would be slightly more transparent. Everybody will understand these changes.
2. The authors apply high quality standards to online recruitment. Two comments here:
 - a. We are not sure if this is easily possible; if they want to the authors could conduct verbal interviews to strengthen and refine the diagnosis of depressive symptomatology.
 - b. Can the authors make sure that participants do not search online for the statistics? Probably, the time limit precludes that on a trial-by-trial base. But participants might do that in-between blocks.
3. Have the authors already pre-registered the study with OSF? If yes, this could be mentioned and amended.
4. Stimuli: We would like to stress that – in our view – some of our previous suggestions regarding the stimuli and the probabilities have not been taken into account properly. See details below. To the very least, we deem it necessary to read about these potential limitations in the final manuscript.
 - a. The positive events (i) overlap considerably (e.g., various items about holidays and marriage), (ii) differ in their positivity (e.g., participants not wanting to get married), and/or (iii) happen with an extremely high probability across the lifetime (e.g., holidays). This makes the comparison between negative and positive events very difficult. If the authors really want to use this list of stimuli, they should make this clear in their descriptions of the methods and results.
 - b. The reputability/applicability of the statistics varies for both negative and positive events. We rather randomly selected three items (obesity, skin burn, create a successful start-up). We did not find numbers for lifetime prevalence in the cited papers. Also, it seems rather odd to select data from Northern Saudi-Arabia for a UK sample for “ski burn.” More importantly, for the positive event “successful start-up for 5 years,” the number in the literature refers to the number of people how have actually launched a start-up – not the number of people overall. So, the number is not what the item in their Registered Report is about. The question is not about how many business survive the 5-year mark; the question is about what is “the average probability that someone in the same environment as the participant experiencing these events in their lifetime.” Now let’s think this through just a minimal amount: how many people even start a business? Probably only few people.
 - c. The above point also motivates us to suggest asking participants for the confidence in the statistics presented (even if this was not included in the to-be-replicated study).

Decision letter (RSOS-190814.R3)

Dear Ms Hobbs,

On behalf of the Editors, I am pleased to inform you that your Manuscript RSOS-190814.R3 entitled "Is depression associated with reduced optimistic belief updating?" has been accepted in principle for publication in Royal Society Open Science subject to minor revision in accordance with the referee and editor suggestions. Please find their comments at the end of this email.

The reviewers and handling editors have recommended publication, but also suggest some minor revisions to your manuscript. Therefore, I invite you to respond to the comments and revise your manuscript.

Please be aware that we need all authors to have active email addresses or addresses that can receive messages from the journal. Currently pv343@bath.ac.uk and as3603@bath.ac.uk are showing as blocked/bouncing. Please can you doublecheck these addresses and, if necessary, send the editorial office amended addresses?

Please you submit the revised version of your manuscript within 7 days (i.e. by the 13-Jan-2021). If you do not think you will be able to meet this date please let me know immediately.

When submitting your revised manuscript, you will be able to respond to the comments made by the referees and you should upload a file "Response to Referees". You can use this to document any changes you make to the original manuscript. In order to expedite the processing of the revised manuscript, please be as specific as possible in your response to the referees.

Full author guidelines can be found here <https://royalsocietypublishing.org/rsos/registered-reports>.

on behalf of Professor Chris Chambers (Subject Editor, Royal Society Open Science)
openscience@royalsociety.org

Associate Editor Comments to Author (Professor Chris Chambers):

Associate Editor: 1

Comments to the Author:

The modified Stage 1 manuscript was returned to one of the previous reviewers for specialist assessment. As you will see, the review is supportive of the proposed revisions to the design and offers a number of constructive suggestions for minor changes and clarifications. Provided these are addressed fully in a revised submission (together with a point by point response to each point), Stage 1 IPA should be re-issued promptly without requiring further in-depth review.

Reviewer comments to Author:

Reviewer: 2

Comments to the Author(s)

It is unfortunate that the authors could not collect the data due to COVID-19. The PhD student helping with the review and I think that it is a good approach to adjust the original plan and collect data online. Below are a few comments for improvement. Given potential time constraints, we do not necessarily need to review the authors' replies.

1. The authors could include parts of their cover letter into the manuscript to state up-front why and how they adjust their original plan (e.g., they could mention in the abstract that this is an online study). This would be slightly more transparent. Everybody will understand these changes.
2. The authors apply high quality standards to online recruitment. Two comments here:
 - a. We are not sure if this is easily possible; if they want to the authors could conduct verbal interviews to strengthen and refine the diagnosis of depressive symptomatology.
 - b. Can the authors make sure that participants do not search online for the statistics? Probably, the time limit precludes that on a trial-by-trial base. But participants might do that in-between blocks.
3. Have the authors already pre-registered the study with OSF? If yes, this could be mentioned and amended.
4. Stimuli: We would like to stress that – in our view – some of our previous suggestions regarding the stimuli and the probabilities have not been taken into account properly. See details below. To the very least, we deem it necessary to read about these potential limitations in the final manuscript.
 - a. The positive events (i) overlap considerably (e.g., various items about holidays and marriage), (ii) differ in their positivity (e.g., participants not wanting to get married), and/or (iii) happen with an extremely high probability across the lifetime (e.g., holidays). This makes the comparison between negative and positive events very difficult. If the authors really want to use this list of stimuli, they should make this clear in their descriptions of the methods and results.
 - b. The reputability/applicability of the statistics varies for both negative and positive events. We rather randomly selected three items (obesity, skin burn, create a successful start-up). We did not find numbers for lifetime prevalence in the cited papers. Also, it seems rather odd to select data from Northern Saudi-Arabia for a UK sample for “ski burn.” More importantly, for the positive event “successful start-up for 5 years,” the number in the literature refers to the number of people how have actually launched a start-up – not the number of people overall. So, the number is not what the item in their Registered Report is about. The question is not about how many business survive the 5-year mark; the question is about what is “the average probability that someone in the same environment as the participant experiencing these events in their lifetime.” Now let’s think this through just a minimal amount: how many people even start a business? Probably only few people.
 - c. The above point also motivates us to suggest asking participants for the confidence in the statistics presented (even if this was not included in the to-be-replicated study).

Author's Response to Decision Letter for (RSOS-190814.R3)

See Appendix D.

Decision letter (RSOS-190814.R4)

Dear Ms Hobbs

On behalf of the Editor, I am pleased to inform you that your revised Stage 1 Registered Report, RSOS-190814.R4 entitled "Is depression associated with reduced optimistic belief updating?" has been accepted in principle for publication in Royal Society Open Science.

You may now progress to Stage 2 and complete the updated study protocol as approved. Before commencing data collection we ask that you:

- 1) Update the journal office as to the anticipated completion date of your study.
- 2) Re-register your approved protocol on the Open Science Framework, either publicly or privately under embargo until submission of the Stage 2 manuscript. Since you had already registered the previously accepted protocol, one option would be to register the updated protocol at <https://osf.io/rr> as a new project and then withdraw the old registration, with the OSF withdrawal note explaining the reasons for the withdrawal and pointing to the updated protocol. Alternatively, you could explore modifying the original protocol. We will leave you to explore these possibilities but whichever path you take, please ensure that the current approved Stage 1 manuscript is registered unchanged from its current approved state, with the time-stamp preceding implementation of the updated study design.

Following completion of your study, we invite you to resubmit your paper for peer review as a Stage 2 Registered Report. Please note that your manuscript can still be rejected for publication at Stage 2 if the Editors consider any of the following conditions to be met:

- The results were unable to test the authors' proposed hypotheses by failing to meet the approved outcome-neutral criteria.
- The authors altered the Introduction, rationale, or hypotheses, as approved in the Stage 1 submission.
- The authors failed to adhere closely to the registered experimental procedures. Please note that any deviations from the approved experimental procedures must be communicated to the editor immediately for approval, and prior to the completion of data collection. Failure to do so can result in revocation of in-principle acceptance and rejection at Stage 2 (see complete guidelines for further information).
- Any post-hoc (unregistered) analyses were either unjustified, insufficiently caveated, or overly dominant in shaping the authors' conclusions.
- The authors' conclusions were not justified given the data obtained.

We encourage you to read the complete guidelines for authors concerning Stage 2 submissions at <https://royalsocietypublishing.org/rsos/registered-reports#ReviewerGuideRegRep>. Please especially note the requirements for data sharing, reporting the URL of the independently registered protocol, and that withdrawing your manuscript will result in publication of a Withdrawn Registration.

Once again, thank you for submitting your manuscript to Royal Society Open Science and we look forward to receiving your Stage 2 submission. If you have any questions at all, please do not hesitate to get in touch. We look forward to hearing from you shortly with the anticipated submission date for your stage two manuscript.

Kind regards,

Royal Society Open Science Editorial Office
Royal Society Open Science
openscience@royalsociety.org

on behalf of Professor Chris Chambers (Registered Reports Editor, Royal Society Open Science)
openscience@royalsociety.org

Author's Response to Decision Letter for (RSOS-190814.R4)

See Appendix E.

RSOS-190814.R5

Review form: Reviewer 1

Is the manuscript scientifically sound in its present form?

Yes

Are the interpretations and conclusions justified by the results?

No

Is the language acceptable?

No

Do you have any ethical concerns with this paper?

No

Have you any concerns about statistical analyses in this paper?

No

Recommendation?

Major revision

Comments to the Author(s)

The authors replicate findings by Korn et al (2014) of an optimistic update bias in healthy individuals but not depressed individuals. Overall, this is a valuable addition to the literature.

There are some inaccuracies that need to be corrected before the paper merits publication. These are listed in detail below.

Comments

1. When using the update bias task it is critical to control for estimation errors (EEs) (Sharot & Garrett 2021; Garrett & Sharot 2017). Failure to do so will produce false results, as been shown with simulations (Garrett & Sharot 2017). The authors follow Korn et al's method of controlling for EEs. This is good. However, they seem to dismiss the results when controlling for EE, and instead highlight the ones that are known to produce false findings.

In particular, despite the analysis which EE is controlled for showing a clear belief update bias for positive events (Figure , pg 90), throughout the manuscript they wrongly claim that no such effect was found: "optimistic belief updating did not occur for positive life events" (abstract). This claim needs to be corrected to say "optimistic belief updating occurred for positive life events". Similar statement should be corrected throughout the manuscript including multiple times in the discussion and results section.

2. The authors cite reference 14 and 15 as dealing with optimistic belief updating, including in the introduction (pg 5 line 25) and in the discussion (pg 5 line 40). This is incorrect. Those references are not about updating and should be deleted from these places and elsewhere where the authors suggest they are.

References 18 and 19 are not about the belief update task the authors are using. Thus, it is misleading to cite those in a way that makes the reader assume they are.

As for reference 16, it is odd to cite an abstract published almost a decade ago, which has never since been reviewed nor appeared in a journal. I suggest deleting this, for fear of mudding the waters with statements in an abstract, which is not accompanied by details of methods, statistics or results. The editor can indicate if this is encouraged or nor.

3. Pg 6, line 18-22 is missing a reference.

4. In the pre-registration did the authors calculate the N needed to have enough power to find a correlation between update bias and BDI? If not, then perhaps they are underpowered? Please calculate the N needed based on Korn et al., 2014, and if underpowered state so clearly.

5. Korn et al, 2014 reports all paired t-test including comparing undesirable update between control and depression groups , and comparing desirable update between control and depression groups- please do the same and provide p values.

6. The authors make some claims (e.g., pg. 19 line 36-42) regarding the presence or absence of differences in updating for desirable and undesirable trials within each group but provide no statistics (e.g., no p nor t). Please provide statistics each time claims are made about the presence or lack thereof of differences.

7. In pg 19 line 54 the word "some" could be misleading. Suggest deleting

8. Pg 21 – first paragraph: The authors say they find no three way interaction, yet the figure shows there may be one, and thus they conduct exploratory analysis they did not pre-register. This is fine as long as they first conduct a Bayes factor (BF10: Rouder et al., 2009) analysis that will assert whether the null finding is reliable or not. Only if the null finding is statistically reliable then the exploratory analysis is justified. The critical number is the Bayes Factor – and the critical question is whether it is greater or smaller than 1. This can be done using JASP (Love et al., 2015). A BF10 greater than 1 provides evidence for rejecting the null hypothesis, a BF10 smaller than 1 provides evidence for the null hypothesis. This is common when reporting null findings.

9. Pg 24 line 8 – the authors only cite Sharot and Sharot & Garrett as providing evidence for the “good news” “bad news” effect. This may seem to do injustice to many others who have provided evidence of this effect. I suggest including:

Kuzmanovic B, Jefferson A, Vogeley K. 2016. The role of the neural reward circuitry in self-referential optimistic belief updates. *NeuroImage* 133:151–162. doi:10.1016/j.neuroimage.2016.02.014

Kuzmanovic B, Rigoux L. 2017. Valence-Dependent Belief Updating: Computational Validation. *Frontiers in Psychology* 8:1087. doi:10.3389/fpsyg.2017.01087

Ma Y, Li S, Wang C, Liu Y, Li W, Yan X, Chen Q, Han S. 2016. Distinct oxytocin effects on belief updating in response to desirable and undesirable feedback. *Proceedings of the National Academy of Sciences of the United States of America* 113:9256–61. doi:10.1073/pnas.1604285113

Kappes A, Faber NS, Kahane G, Savulescu J, Crockett MJ. 2018. Concern for Others Leads to Vicarious Optimism. *Psychol Sci* 29:379–389. doi:10.1177/0956797617737129

Oganian, Y., Heekeren, H. R., & Korn, C. W. (2019). Low foreign language proficiency reduces optimism about the personal future. *Quarterly Journal of Experimental Psychology*, 72(1), 60-75.

Krieger JL, Murray F, Roberts JS, Green RC. 2016. The impact of personal genomics on risk perceptions and medical decision-making. *Nat Biotechnol* 34:912–918. doi:10.1038/nbt.3661

Korn, C. W., Prehn, K., Park, S. Q., Walter, H., & Heekeren, H. R. (2012). Positively biased processing of self-relevant social feedback. *Journal of Neuroscience*, 32(47), 16832-16844.

Korn, C. W., La Rosée, L., Heekeren, H. R., & Roepke, S. (2016). Social feedback processing in borderline personality disorder. *Psychological Medicine*, 46(3), 575-587.

Eil, D., & Rao, J. M. (2011). The good news-bad news effect: asymmetric processing of objective information about yourself. *American Economic Journal: Microeconomics*, 3(2), 114-38.

10. Pg 24 line 8 – I believe reference 15 is irrelevant here. For reference 16 see my comment above. When citing 17 I suggest stating that 17 has itself been critiqued by Garrett & Sharot, 2017. Similarly, when citing 55 (line 40 pg 28) it should be stated that an attempt to replicate the study failed and that the study was criticized (Garrett & Sharot 2021) for not following the appropriate guidelines for running and analyzing the update bias task as detailed in Sharot & Garrett, 2021.

11. Discussion, pg 25 line 16-22, may need to be changed depending on BF10 results. Pg 23 line 37 may need to be changed based on power calculation results.

12. Discussion pg 28 line 35: “That we did not find evidence of optimistic belief updating for positive events in either people with depression or healthy controls, speaks against the existence of a universal optimistic updating bias. Alternatively, our pattern of results are a potential sign of an underlying problem with the validity of the optimistic updating paradigm, which is currently being debated [14–17,55]. As such, reduced optimistic belief updating effects associated with depression may require further validation after a more robust measure of optimistic updating is developed.” – But the authors do find optimistic updating bias in healthy individuals and in depressed individuals (figure pg 90). They only fail to find one when they fail to control for EEs. All past studies of the update task control for EEs. This is the appropriate thing to do. These sentences are thus not supported by the authors data and need to be deleted.

13. Discussion: “It has been shown that a more appropriate way of administering the task necessitates asking participants about both their personal as well as the average person’s likelihood of experiencing life events, that is, ‘the base rate’ [17].” – Why is this more appropriate? If the authors claim so, they must explain. It has been shown that the results do not change whether following this or not (Garrett & Sharot, 2017). If the authors believe it is much more appropriate, why did the authors not do so? And does this mean that the authors’ results in this paper are of no clinical value?

14. Discussion “There are also limitations in asking participants to accurately compare their chances of experiencing life events relative to the average person including, but not limited to, scale attention and base rate regression [14,15].” I do not see how this sentence is relevant, as in the task the subjects are not asked any such thing.

15. Discussion – last sentence – the authors suggest mechanism of the effect are unknown. This is inaccurate. There are studies detailing the neural mechanism of the effect in depression (Garrett et al., 2014) including studies detailing the role of dopamine in producing it (Sharot et al., 2012)

Review form: Reviewer 2

Is the manuscript scientifically sound in its present form?

Yes

Are the interpretations and conclusions justified by the results?

Yes

Is the language acceptable?

Yes

Do you have any ethical concerns with this paper?

No

Have you any concerns about statistical analyses in this paper?

No

Recommendation?

Accept with minor revision

Comments to the Author(s)

In our view, the following five criteria are fulfilled.

1. Whether the data are able to test the authors’ proposed hypotheses by passing the approved outcome-neutral criteria (such as absence of floor and ceiling effects or success of positive controls)
2. Whether the Introduction, rationale and stated hypotheses are the same as the approved Stage 1 submission
3. Whether the authors adhered precisely to the registered experimental procedures
4. Where applicable, whether any unregistered exploratory statistical analyses are justified, methodologically sound, and informative
5. Whether the authors’ conclusions are justified given the data

We have a few comments for improvement:

- Even though not fully conclusive, the differences between negative and positive items are relevant and important: One main point here that the authors do not mention or discuss is that positive events are not positive for everybody. In contrast, most negative items are negative

for everybody. Such individual differences in the “positivity” of positive events could explain some of the potential differences.

- Relatedly, one interesting thing that we haven't yet seen them discuss is that for the depression group the positive items are less positive and more negative than they are for the healthy group. For negative items, both groups seem to be similar, but for positive items, there seems to be more of a difference. Not sure, whether this difference is meaningful though and what it's driven by, but might be worth discussing this or looking into, to see whether there's anything interesting going on here.

- Again related: The estimation errors for desirable and undesirable information seem to differ largely for positive events, with undesirable estimation errors being larger than desirable estimation errors. This can also be seen by comparing the right parts of Fig 5 and Fig 7, i.e., between unscaled and scaled updates. This should be explicitly mentioned and potentially be discussed.

- It might also be nice to have a large table in which they list the positivity/negativity etc. for each item, so that future studies can exclude those items that didn't work as well as the others. Or they could just provide a list of items that seemed to work very well to facilitate better future studies. If there aren't many studies that use positive items, then this study might become the basis for future studies.

- In the 'Results' section, in the 'Sample' subsection, the authors mention the differences between their two groups but don't mention that the depression group is almost twice as likely (44% vs 23%) to be single. That seems like quite a difference and worth mention whilst they're already listing differences.

Review form: Reviewer 3

Is the manuscript scientifically sound in its present form?

Yes

Are the interpretations and conclusions justified by the results?

Yes

Is the language acceptable?

Yes

Do you have any ethical concerns with this paper?

No

Have you any concerns about statistical analyses in this paper?

No

Recommendation?

Accept as is

Comments to the Author(s)

I enjoyed reading this at the pre-registered phase and am delighted to see the study completed. The analyses are appropriate and in line with the pre-registered report. Additional exploratory analyses are justified and appropriate. I have only a couple of minor comments.

I don't remember if I picked this up at pre-registered stage but on reading this submission, I was unclear about whether the PHQ was used as a pre-screen on Prolific, i.e. were participants invited to complete the PHQ and then only moved on to the rest of the study if they fell within the defined ranges (over 15 or under 4)? It may be worth making this explicitly clear.

Looking at the list of life events, it seems that there could be an effect of age when it comes to making an estimate. For example, an older person who is already married would be likely to make a more accurate judgement regarding the question about likelihood of being married in their lifetime. Since age was different between the two groups and not included in the analyses, is it possible that this could have affected any of the outcomes?

The conclusions are well thought-out and offer a range of possible mechanisms. Another contributing factor may be the cognitive inflexibility that is common in depression, although if it was a simple neuropsychological feature it might be expected to affect positive and negative life events equally.

Overall, a very interesting study with real-world relevance. Thank you for the opportunity of reviewing this.

Decision letter (RSOS-190814.R5)

Dear Ms Hobbs:

On behalf of the Editor, I am pleased to inform you that your Stage 2 Registered Report RSOS-190814.R5 entitled "Is depression associated with reduced optimistic belief updating?" has been deemed suitable for publication in Royal Society Open Science subject to minor revision in accordance with the referee suggestions. Please find the referees' comments at the end of this email.

The reviewers and Subject Editor have recommended publication, but also suggest some minor revisions to your manuscript. We invite you to respond to the comments and revise your manuscript. Below the referees' and Editors' comments (where applicable) we provide additional requirements. Final acceptance of your manuscript is dependent on these requirements being met. We provide guidance below to help you prepare your revision.

In addition, we ask that all authors have an active email address able to receive correspondence from the journal. As your colleagues' email addresses (pv343@bath.ac.uk; as3603@bath.ac.uk) are not currently receiving messages from the ScholarOne website, please can you do either or both of the following:

- Ensure that these authors' email addresses are 'whitelisting' emails from ScholarOne (these may be in the form 'onbehalf@manuscriptcentral.com');
- Confirm an alternative, active email address for these co-authors with the journal's editorial office.

Please submit your revised manuscript and required files (see below) no later than 21 days from today's (ie 06-Dec-2021) date. Note: the ScholarOne system will 'lock' if submission of the revision is attempted 21 or more days after the deadline. If you do not think you will be able to meet this deadline please contact the editorial office immediately.

Please note article processing charges apply to papers accepted for publication in Royal Society Open Science (<https://royalsocietypublishing.org/rsos/charges>). Charges will also apply to papers transferred to the journal from other Royal Society Publishing journals, as well as papers submitted as part of our collaboration with the Royal Society of Chemistry

(<https://royalsocietypublishing.org/rsos/chemistry>). Fee waivers are available but must be requested when you submit your revision (<https://royalsocietypublishing.org/rsos/waivers>).

on behalf of Professor Chris Chambers
(Registered Reports Editor, Royal Society Open Science)
openscience@royalsociety.org

Associate Editor Comments to Author (Professor Chris Chambers):

Associate Editor: 1

Comments to the Author:

The three reviewers from Stage 1 kindly returned to evaluate the Stage 2 manuscript. As you will see, the comments are broadly very positive and the manuscript should be suitable for full acceptance following a round of minor revision. Key issues to address including concerns with the factual accuracy of certain claims, clarification (and possible optimisation) of the presentation and implementation of the exploratory analyses, and resolving interpretative concerns in the Discussion. In revising, please avoid making any changes to the Introduction and Method sections unless doing so is necessary to correct a factual error or resolve a major point of confusion. Provided you are able to address all issues in a thorough revision, final Stage 2 acceptance should be forthcoming without requiring further in-depth review.

Comments to Author:

Reviewer: 1

Comments to the Author(s)

The authors replicate findings by Korn et al (2014) of an optimistic update bias in healthy individuals but not depressed individuals. Overall, this is a valuable addition to the literature. There are some inaccuracies that need to be corrected before the paper merits publication. These are listed in detail below.

Comments

1. When using the update bias task it is critical to control for estimation errors (EEs) (Sharot & Garrett 2021; Garrett & Sharot 2017). Failure to do so will produce false results, as been shown with simulations (Garrett & Sharot 2017). The authors follow Korn et al's method of controlling for EEs. This is good. However, they seem to dismiss the results when controlling for EE, and instead highlight the ones that are known to produce false findings.

In particular, despite the analysis which EE is controlled for showing a clear belief update bias for positive events (Figure , pg 90), throughout the manuscript they wrongly claim that no such effect was found: "optimistic belief updating did not occur for positive life events" (abstract). This claim needs to be corrected to say "optimistic belief updating occurred for positive life events". Similar statement should be corrected throughout the manuscript including multiple times in the discussion and results section.

2. The authors cite reference 14 and 15 as dealing with optimistic belief updating, including in the introduction (pg 5 line 25) and in the discussion (pg 5 line 40). This is incorrect. Those references are not about updating and should be deleted from these places and elsewhere where the authors suggest they are.

References 18 and 19 are not about the belief update task the authors are using. Thus, it is misleading to cite those in a way that makes the reader assume they are.

As for reference 16, it is odd to cite an abstract published almost a decade ago, which has never since been reviewed nor appeared in a journal. I suggest deleting this, for fear of mudding the waters with statements in an abstract, which is not accompanied by details of methods, statistics or results. The editor can indicate if this is encouraged or not.

3. Pg 6, line 18-22 is missing a reference.

4. In the pre-registration did the authors calculate the N needed to have enough power to find a correlation between update bias and BDI? If not, then perhaps they are underpowered? Please calculate the N needed based on Korn et al., 2014, and if underpowered state so clearly.

5. Korn et al, 2014 reports all paired t-test including comparing undesirable update between control and depression groups, and comparing desirable update between control and depression groups- please do the same and provide p values.

6. The authors make some claims (e.g., pg. 19 line 36-42) regarding the presence or absence of differences in updating for desirable and undesirable trials within each group but provide no statistics (e.g., no p nor t). Please provide statistics each time claims are made about the presence or lack thereof of differences.

7. In pg 19 line 54 the word "some" could be misleading. Suggest deleting

8. Pg 21 - first paragraph: The authors say they find no three way interaction, yet the figure shows there may be one, and thus they conduct exploratory analysis they did not pre-register. This is fine as long as they first conduct a Bayes factor (BF10: Rouder et al., 2009) analysis that will assert whether the null finding is reliable or not. Only if the null finding is statistically reliable then the exploratory analysis is justified. The critical number is the Bayes Factor - and the critical question is whether it is greater or smaller than 1. This can be done using JASP (Love et al., 2015). A BF10 greater than 1 provides evidence for rejecting the null hypothesis, a BF10 smaller than 1 provides evidence for the null hypothesis. This is common when reporting null findings.

9. Pg 24 line 8 - the authors only cite Sharot and Sharot & Garrett as providing evidence for the "good news" "bad news" effect. This may seem to do injustice to many others who have provided evidence of this effect. I suggest including:

Kuzmanovic B, Jefferson A, Vogeley K. 2016. The role of the neural reward circuitry in self-referential optimistic belief updates. *NeuroImage* 133:151-162. doi:10.1016/j.neuroimage.2016.02.014

Kuzmanovic B, Rigoux L. 2017. Valence-Dependent Belief Updating: Computational Validation. *Frontiers in Psychology* 8:1087. doi:10.3389/fpsyg.2017.01087

Ma Y, Li S, Wang C, Liu Y, Li W, Yan X, Chen Q, Han S. 2016. Distinct oxytocin effects on belief updating in response to desirable and undesirable feedback. *Proceedings of the National Academy of Sciences of the United States of America* 113:9256-61. doi:10.1073/pnas.1604285113

Kappes A, Faber NS, Kahane G, Savulescu J, Crockett MJ. 2018. Concern for Others Leads to Vicarious Optimism. *Psychol Sci* 29:379-389. doi:10.1177/0956797617737129

Oganian, Y., Heekeren, H. R., & Korn, C. W. (2019). Low foreign language proficiency reduces optimism about the personal future. *Quarterly Journal of Experimental Psychology*, 72(1), 60-75.

Krieger JL, Murray F, Roberts JS, Green RC. 2016. The impact of personal genomics on risk perceptions and medical decision-making. *Nat Biotechnol* 34:912-918. doi:10.1038/nbt.3661

Korn, C. W., Prehn, K., Park, S. Q., Walter, H., & Heekeren, H. R. (2012). Positively biased processing of self-relevant social feedback. *Journal of Neuroscience*, 32(47), 16832-16844.

Korn, C. W., La Rosée, L., Heekeren, H. R., & Roepke, S. (2016). Social feedback processing in borderline personality disorder. *Psychological Medicine*, 46(3), 575-587.

Eil, D., & Rao, J. M. (2011). The good news-bad news effect: asymmetric processing of objective information about yourself. *American Economic Journal: Microeconomics*, 3(2), 114-38.

10. Pg 24 line 8 – I believe reference 15 is irrelevant here. For reference 16 see my comment above. When citing 17 I suggest stating that 17 has itself been critiqued by Garrett & Sharot, 2017.

Similarly, when citing 55 (line 40 pg 28) it should be stated that an attempt to replicate the study failed and that the study was criticized (Garrett & Sharot 2021) for not following the appropriate guidelines for running and analyzing the update bias task as detailed in Sharot & Garrett, 2021.

11. Discussion, pg 25 line 16-22, may need to be changed depending on BF10 results. Pg 23 line 37 may need to be changed based on power calculation results.

12. Discussion pg 28 line 35: “That we did not find evidence of optimistic belief updating for positive events in either people with depression or healthy controls, speaks against the existence of a universal optimistic updating bias. Alternatively, our pattern of results are a potential sign of an underlying problem with the validity of the optimistic updating paradigm, which is currently being debated [14–17,55]. As such, reduced optimistic belief updating effects associated with depression may require further validation after a more robust measure of optimistic updating is developed.” – But the authors do find optimistic updating bias in healthy individuals and in depressed individuals (figure pg 90). They only fail to find one when they fail to control for EEs. All past studies of the update task control for EEs. This is the appropriate thing to do. These sentences are thus not supported by the authors data and need to be deleted.

13. Discussion: “It has been shown that a more appropriate way of administering the task necessitates asking participants about both their personal as well as the average person’s likelihood of experiencing life events, that is, ‘the base rate’ [17].” – Why is this more appropriate? If the authors claim so, they must explain. It has been shown that the results do not change whether following this or not (Garrett & Sharot, 2017). If the authors believe it is much more appropriate, why did the authors not do so? And does this mean that the authors results in this paper are of no clinical value?

14. Discussion “There are also limitations in asking participants to accurately compare their chances of experiencing life events relative to the average person including, but not limited to, scale attention and base rate regression [14,15].” I do not see how this sentence is relevant, as in the task the subjects are not asked any such thing.

15. Discussion – last sentence – the authors suggest mechanism of the effect are unknown. This is inaccurate. There are studies detailing the neural mechanism of the effect in depression (Garrett et al., 2014) including studies detailing the role of dopamine in producing it (Sharot et al., 2012)

Reviewer: 2

Comments to the Author(s)

In our view, the following five criteria are fulfilled.

1. Whether the data are able to test the authors’ proposed hypotheses by passing the approved outcome-neutral criteria (such as absence of floor and ceiling effects or success of positive controls)

2. Whether the Introduction, rationale and stated hypotheses are the same as the approved Stage 1 submission
3. Whether the authors adhered precisely to the registered experimental procedures
4. Where applicable, whether any unregistered exploratory statistical analyses are justified, methodologically sound, and informative
5. Whether the authors' conclusions are justified given the data

We have a few comments for improvement:

- Even though not fully conclusive, the differences between negative and positive items are relevant and important: One main point here that the authors do not mention or discuss is that positive events are not positive for everybody. In contrast, most negative items are negative for everybody. Such individual differences in the "positivity" of positive events could explain some of the potential differences.
- Relatedly, one interesting thing that we haven't yet seen them discuss is that for the depression group the positive items are less positive and more negative than they are for the healthy group. For negative items, both groups seem to be similar, but for positive items, there seems to be more of a difference. Not sure, whether this difference is meaningful though and what it's driven by, but might be worth discussing this or looking into, to see whether there's anything interesting going on here.
- Again related: The estimation errors for desirable and undesirable information seem to differ largely for positive events, with undesirable estimation errors being larger than desirable estimation errors. This can also be seen by comparing the right parts of Fig 5 and Fig 7, i.e., between unscaled and scaled updates. This should be explicitly mentioned and potentially be discussed.
- It might also be nice to have a large table in which they list the positivity/negativity etc. for each item, so that future studies can exclude those items that didn't work as well as the others. Or they could just provide a list of items that seemed to work very well to facilitate better future studies. If there aren't many studies that use positive items, then this study might become the basis for future studies.
- In the 'Results' section, in the 'Sample' subsection, the authors mention the differences between their two groups but don't mention that the depression group is almost twice as likely (44% vs 23%) to be single. That seems like quite a difference and worth mention whilst they're already listing differences.

Reviewer: 3

Comments to the Author(s)

I enjoyed reading this at the pre-registered phase and am delighted to see the study completed. The analyses are appropriate and in line with the pre-registered report. Additional exploratory analyses are justified and appropriate. I have only a couple of minor comments. I don't remember if I picked this up at pre-registered stage but on reading this submission, I was unclear about whether the PHQ was used as a pre-screen on Prolific, i.e. were participants invited to complete the PHQ and then only moved on to the rest of the study if they fell within the defined ranges (over 15 or under 4)? It may be worth making this explicitly clear. Looking at the list of life events, it seems that there could be an effect of age when it comes to making an estimate. For example, an older person who is already married would be likely to make a more accurate judgement regarding the question about likelihood of being married in their lifetime. Since age was different between the two groups and not included in the analyses, is it possible that this could have affected any of the outcomes? The conclusions are well thought-out and offer a range of possible mechanisms. Another contributing factor may be the cognitive inflexibility that is common in depression, although if it was a simple neuropsychological feature it might be expected to affect positive and negative life events equally. Overall, a very interesting study with real-world relevance. Thank you for the opportunity of reviewing this.

===PREPARING YOUR MANUSCRIPT===

one version should clearly identify all the changes that have been made (for instance, in coloured highlight, in bold text, or tracked changes);

===PREPARING YOUR REVISION IN SCHOLARONE===

-- If you are requesting an article processing charge waiver, you must select the relevant waiver option (if requesting a discretionary waiver, the form should have been uploaded, see 'File upload' above).

-- If you have uploaded any electronic supplementary (ESM) files, please ensure you follow the guidance at <https://royalsociety.org/journals/authors/author-guidelines/#supplementary-material> to include a suitable title and informative caption. An example of appropriate titling and captioning may be found at https://figshare.com/articles/Table_S2_from_Is_there_a_trade-off_between_peak_performance_and_performance_breadth_across_temperatures_for_aerobic_scope_in_teleost_fishes_/3843624.

Author's Response to Decision Letter for (RSOS-190814.R5)

See Appendix F.

Decision letter (RSOS-190814.R6)

Dear Katie

It is a pleasure to accept your Stage 2 Registered Report entitled "Is depression associated with reduced optimistic belief updating?" in its current form for publication in Royal Society Open Science.

Thank you for your fine contribution. On behalf of the Editors of Royal Society Open Science, we look forward to your continued contributions to the journal.

on behalf of Professor Chris Chambers (Subject Editor)
openscience@royalsociety.org

Appendix A

Editor Comments to Author (Professor Chris Chambers):

Three expert reviewers have now assessed the manuscript. Two of the reviews (Rev 2 and 3) are mostly positive, noting that one or both of the Stage 1 primary criteria are already met, while one review is more negative, noting that they neither of the primary criteria are met. Reviewer 1 offers a valuable critical appraisal of the entire project, among other points noting lack of clarity in the conceptual framing of the study, and potential interpretative confusion between optimism bias and update bias. Most crucially for the Replication article type, however, the reviewer also notes areas that lack sufficient detail and which deviate substantially from the original report. These comments in particular suggest that the manuscript may fall short of primary criteria #1 and #2 (see section 3.2 in the author guidelines for details: <https://royalsocietypublishing.org/rsos/replication-studies>). Please note that only the primary criteria need be met to achieve in principle acceptance as a Replication article type.

Reviewer 2 is more positive but also notes areas of deviation. An exact replication of a study is rarely possible, and not all deviations from the original study will necessarily violate primary criterion #2. However, please take care in responding to minimise and fully justify any deviations so that a final editorial judgment can be made. In the event that the deviations are too substantial to qualify for the Replications track, it may be possible for your submission to be diverted as a regular Registered Report while preserving the current review process. As a Registered Report, a broader range of criteria must be met to achieving in principle acceptance (bringing to the fore many of the comments of Reviewer 1 that fall outside the Replications criteria) but there is also greater flexibility to deviate from the original target study. Finally Reviewer 3 offers a mostly positive assessment but also notes areas of the methodology that require additional clarification and justification.

Overall, these assessments prompt a Major Revision recommendation, focusing especially on satisfying the primary criteria for a Replication article type, while also responding as thoroughly as possible to other issues raised by the reviewers. Where recommendations from the reviewers conflict with the primary criteria (e.g. suggestions to change the design away from the target study) note that the primary criteria take precedence, unless as indicated above, the authors decide to pursue the Registered Reports track instead.

Comments to Author:

Reviewer: 1

The authors plan to test whether the 'update bias' for negative life events is abolished in depression, replicating Korn et al. They further plan to extend the task to study update bias for positive life events.

Thank you to the reviewer for raising these potential issues with the use of positive life events in this paradigm. To date a number of papers have published findings with 'positive events' that use a less stringent criteria (for example, derived statistics)¹⁻³. It is important to understand whether previous mixed findings regarding replication of update bias to positive events are due to design limitations, or a lack of effect for positively valenced stimuli. We have responded to each of the reviewer's points in detail below.

(i) My main concern is the extension plan to positive life events. There are serious limitations in the design stemming from the problematic choice of "positive" life events.

1. "Positive events" are not positive. While all the negative life events are clearly negative many of the 'positive life events' are not in fact positive. Some events may be viewed as

positive by some individuals but quite terrible by others. For example, I consider the following 'positive events' very negative: Have 3 or more children, Owning a cat, Owning a dog, Owning a pet, Retire early (before 65), Having a baby before the age of 34, Go to an Opera, Having musical instrument lessons, Go on a caravan trip or camping trip. I will pay good money to avoid all (especially the first and last). In fact, I found most of the "positive events" mildly negative or of no interest to me. I would not like to "play a musical instrument", "go to a national park", "rent out 2 or more properties", neither am I interested in "owning 2 cars" and on and on. Now, while I like skiing and running I know many people that dislike these activities. Most of the positive stimuli will raise objection.

We acknowledge that there is likely to be some variation in the extent to which events are viewed positively or negatively by participants, particularly for positive life events. We therefore ask participants to rate each event on seven scales, including negativity ('How negative would this event be for you?') and positivity ('How positive would this event be for you?'). It is therefore possible for us to determine whether any findings in relation to positive life events are influenced by variation in perceptions of negativity or positivity. We have amended our analysis plan to include a linear regression model that would allow us to examine the potential confounding role of perceptions of positivity and negativity.

2. No relevant base rates for positive events. It seems unlikely that the authors would have base rates for the positive events. How do the authors know what the base rate is of "going on a spontaneous holiday" or "taking part in a running race"? The authors should include a table with the base rates of all events and the source where they got the base rate. Moreover, the probability will surely alter for each subject according to their preference. That is – I like the ballet so I know I am 100% likely to go in the future. My friends hates the ballet so they know they are 0% likely to go. The base rates the authors will provide will be irrelevant to specific subjects. This is partially because of point 3 below.

We apologise for not including the base rates and sources in the initial article. Please find the rates and sources attached with this resubmission.

As in Korn et al (2014), these rates are based on the average probability for each individual. Whilst there is likely to be some variation between individuals in terms of probability, participants are told that the presented values are the average probability for someone from a similar environment to themselves. This issue is also applicable to negative life events used in this paper, and in the original paper that we are replicating. For example, the probability of Cancer is likely to be higher for participants that smoke. Previous research has demonstrated update biases despite potential individual variation.

Furthermore, to address issues over the extent to which life events are controllable we ask participants to provide a rating of controllability ('How much control do you have over this event?'), prior experience ('Has this event happened to you before?'), and familiarity ('Regardless if this event has happened to you before, how familiar do you feel it is to you from TV, friends, movies and so on?'). We would therefore be able to ascertain whether differences in these factors influence update scores. We have amended our analysis plan to include linear regression models that would allow us to examine this. Evidence of confounding would provide relevant findings to this field, contributing towards an account of previous mixed findings, and specifying in which contexts these biases are apparent.

3. Positive events under subject's control, not so negative events. Almost all the positive events are very much under the subjects' control. A person can mostly decide if they want to go visit a national park or take a holiday. Many of the negative events are much less under the subject's control. It has recently been shown that control is critical to asymmetric learning from positive and negative outcomes (Dorfman, H.M., Bhui, R., Hughes, B.L., & Gershman, S.J– Psychological Science 2019). The authors may well be contrasting events that are under one's control with events that are not, rather than contrasting valence. This is a serious confound.

We acknowledge that there is likely to be some variation in the extent of control that an individual has over life events. As stated above we ask participants to provide a rating of controllability ('How much control do you have over this event?'). We have included a linear regression model in our analysis plan that would allow us to examine whether update scores may be influenced by controllability, and whether this differs according to valence. Evidence of confounding by controllability would provide relevant findings to this field through specifying the context in which update biases may occur. Furthermore, this may illuminate previous mixed findings regarding the role of positive life events in update biases.

There is a reason why the "update task" for life events has not been extended to positive events despite the original task being replicated numerous times by labs around the world. It just does not lend itself to such extension – there are not enough positive events that are positive to all and that have objective statistics that are associated with them. All the above critique has already been articulated before (18).

The reviewer states that this design has twice failed to replicate. However reference 18² reported significant optimism biases in relation to both positive and negative life events. There has therefore been one study suggesting the presence of optimism bias for positive life events in healthy individuals. However, this has not yet been tested in individuals experiencing depression as this study hopes to contribute to the field. Whilst some cognitive theories of depression suggest that depression is characterised by emotional blunting to both positive and negative stimuli, others have suggested that depression is better characterised by the loss of positive biases seen in healthy individuals. Understanding the situations in which updating biases are present in individuals with depression would contribute to a greater understanding of the nature of cognitive biases in depression, and provide specific areas for therapeutic intervention.

I would recommend the authors consider using the design of (Lefebvre et al. 2017- Nature Human Behaviour). They look at valence-dependent asymmetric learning using a basic reinforcement learning task which allows them to look at updating in response to both positive and negative outcomes about rewards (positive events) and losses (negative events). They also relate the results to optimism and have replicated the results a few times. There are other well designed tasks that allow looking at update about positive and negative stimuli (e.g., Eil & Rao, 2011) why not use one of those rather than a design (15) that has twice failed to replicate (18,19)?

We thank the reviewer for the suggestion of using a reinforcement learning task to examine updating in response to positive and negative outcomes about reward and losses. However a number of studies have examined impaired reward and punishment processing in depression in relation to a range of stimuli with fairly consistent findings^{4, 5}. We are specifically interested in whether these biases would emerge specifically in relation to updating beliefs regarding the probability of experiencing positive and negative life events. This may offer a specific target for therapeutic

intervention (rather than more general information processing biases), and address previous disparities regarding the use of this task with positive life events.

(ii) I have some comments regarding the replication portion.

1. This study should be considered a “conceptual replication” rather than a pure replication as there are many differences between the proposed study and that of Korn et al. First, almost half the stimuli are different than those used originally. The authors add 35 new stimuli to the 40 Korn et al used. Half of the new stimuli are about the likelihood of dying and use the word “death” or “mortality”. There is a huge literature about how priming people with death alters cognition and this new manipulation could have unknown effects. I think it is fine to have new stimuli, but under these conditions it can simply not be considered a pure replication and this should be clear in the abstract, intro and discussion. Moreover, the population used by Korn et al., are different to that used here. Korn et al., had a large number of hospitalized patients, many on medication. It seems the authors plan to use a non-hospitalized convenient sample. Also take note that Korn et al., tested German subjects in German. All these differences should be noted from the beginning as they may influence the results. This is not a “pure replication” as the authors suggest in page 11 line 32 (also see pg 12 line 45).

In our submission we stated ‘This procedure is a replication of the procedure outlined by Korn et al (2014) but with the additional test blocks of positive life events’, ‘The subsequent task outcomes follow the procedure outlined by Korn et al (2014)’, and ‘This ensured that a pure replication of the procedure used by Korn et al (2014) was available for the subgroup of participants who completed the negative block first’. We believe that these statements are accurate as the outlined procedures and calculated task outcomes follow the procedures reported in Korn et al (2014), with the addition of positive life events (as we have stated).

We acknowledge that our sample differs in terms of the source of our sample. However, the clinical characteristics of the samples are likely to be similar as we recruit participants based on displaying clinical symptoms of depression (PHQ-9 scores greater than 9). Participants in Korn et al (2013) were recruited from patients at the local hospital, but were not restricted to inpatients (as ‘hospitalised’ may suggest). Furthermore, whilst Korn et al (2014) includes German subjects with life events translated to German, event probabilities from the UK were used. Similar research conducted in the UK has reported similar findings to Korn et al (2014) suggesting minimal cultural variation².

Furthermore, we acknowledge that we have used 32 different life events to those implemented by Korn (in addition to the 40 original life events). Unfortunately, it was not possible for us to use the baseline probabilities used in the original paper as the probabilities and their sources were not reported. Furthermore, in their paper Korn et al specify that life event probabilities must fall within 10% to 70% to allow for sufficient movement in re-estimates. This prevented us from including a number of life events used in the original paper, as they were rarer or more common than these specified probabilities. For example, the prevalence of dementia in the UK population is 1.3% across all age ranges, and 7.1% in those aged 65 and over⁶. This therefore falls below the specified probability of 10%. To account for previous criticisms of attempts to replicate this paradigm, relating to validity of statistics, and to reduce possible floor and ceiling effects we chose to identify novel negative life events. However, we appreciate the reviewers note that the life events we have chosen to include primarily relate to Death. We therefore propose to amend our application to use this task with a reduced number of life events, only including those used by Korn et al but that also meet the criteria of (1) being a validated statistic, and (2) falling within the range of 10-70%. This would

involve using 40 negative life events, and two training events (rather than the 70 specified in our submission).

2. The authors should provide all base rates of the stimuli in the supplementary Table. A major critique of the work by Shah et al., (15) is that the base rates used induced a statistical artefact (18). Also the authors say they calculated new base rates, which are different from Korn et al. “to account for possible changes over time” - please clarify what is meant by this.

Again we apologise for not including the base rates and sources. This has now been amended in the resubmitted paper. “to account for possible changes over time” refers to the fact that Korn et al’s work was published in 2014. There may therefore be changes in the probability of life events occurring since the publication of Korn et al’s (2014) work. As the authors did not report the baseline probabilities in the original paper, we determined new baseline probabilities, including statistical sources after 2014. There is therefore likely to be some slight variation to the probabilities presented in the original. As the authors of the original paper did not report the baseline probabilities, we are not able to ascertain the extent to which our probabilities varied from those used originally. However, both the original paper and this study use accurate probabilities from validated sources.

3. Power. The authors calculate that 56 subjects are needed in each group. That seems very reasonable. However, they then say that only half (23 in each group) can be considered as a replication of Korn et al because the other half will be given positive event blocks before negative event blocks and thus cannot be considered in the analysis of the replication data. This is problematic as it means the authors would be underpowered. My suggestion is to not include the positive events, which are problematic in any case. Moreover, the authors say they will re-do the analysis on subjects that meet the strict criteria used in Korn et al (pg 17), but they do not indicate which N they will use. Could it be that as a result only about 11 subjects will be left in each group? If this is a replication attempt than only subjects that meet the criteria used by Korn et al (with regards to depression or its absence) should be recruited to begin with, and the N should be 56 of those per group. (side note - Can the authors include a diagram of all blocks?).

Thank you to the reviewer for raising these points regarding the sample size calculation. We would like to clarify that 54 subjects are required per group according to our sample size calculation as stated in our submitted work (not 56 as suggested by the reviewer), half of which would be 27 participants per group (not 23 as suggested by the reviewer).

In regards to the strict replication of only individuals that completed the negative life events first, this would provide 86% power to detect an effect of $\eta_p^2 = 0.113$ at an alpha level of 0.05. Whilst this is less power than provided by the entirety of our sample, this would be adequate to detect the previously reported effect. Furthermore, this would primarily be an issue if order effects influenced our findings. As stated in our submission, we plan to determine whether the order in which participants completed the life events using an adjusted linear regression model. The subgroup analysis using a smaller sample would only be conducted if order effects were present.

Regarding the re-analysis of subjects that meet the stricter criteria used by Korn et al (2014), we are unable to specify an exact N as participants are recruited on the basis of PHQ-9 scores. Therefore, whilst this is indicative of participants experiencing depression it does not confirm a clinical diagnosis. Previous work in our lab that has recruited participants meeting the same criteria of PHQ-9 scores (greater than or equal to 10), found that 75% of participants met diagnostic criteria for a

depressive disorder. Additionally, 92% of participants in a healthy control group, recruited on the basis of PHQ-9 scores less than 5, met Korn et al's (2014) criteria for the healthy control group. We therefore estimate that approximately 40 participants would meet the inclusion criteria outlined by Korn et al for the depressed group, and approximately 49 participants would meet the inclusion criteria outlined by Korn et al for the healthy control group. A sample size of 89 participants would provide 89% power to detect an effect of $\eta_p^2 = 0.113$ at an alpha level of 0.005.

4. Update should be calculated as movement towards the information given rather than absolute (pg 13 line 54). That is, if the initial estimate is 20 and the information is 10 and the second estimate is 30 then update is -10 not 10, because the subject is updating away from the direction of the information.

We agree that the current procedure confuses situations where participants consistently move away from the average probability with situations where participants consistently move towards the average probability. We have therefore amended this to follow the procedure used in a number of papers examining biased updating, where update scores are calculated so that positive scores indicate a move towards the average probability, whereas negative scores indicate a move away from the average probability (irrespective of valence and desirability). In specific, the absolute difference between first and second estimates will be calculated, and then coded as positive if the update is in the direction of the base rate and negative when the update is in the direction away from the base rate. We have amended this in our submission.

5. Different analysis plans. The authors plan to use a different analysis to that used by Korn et al., (pg 15) – linear regression rather than ANOVA. I believe that a replication should follow the exact analysis. The authors can add additional analysis if they wish. Why are there no plans to look at, and control for, the additional ratings (familiarity, arousal etc)?

We acknowledge that we plan to use a linear regression rather than an ANOVA, as used by Korn et al. However, ANOVA and linear regression are equivalent statistical models and both follow general linear modelling, it is the information presented that differs⁷. Linear regression has the advantage of providing more interpretable regression coefficients, aiding reader comprehension. We would be happy to run an ANOVA in addition to the linear regression model, but this would not provide additional information beyond that provided by the linear regression model.

We have amended our data analysis plan to include analyses examining the influence of each of the rating scales. Thank you to the reviewer for highlighting this oversight.

General comments

The authors confuse “optimism bias” with “update bias”. The distinction between the two is important. Optimism bias is the tendency to overestimate the likelihood of positive events and underestimate the likelihood of negative events. This has already been shown to be absent in depression with both positive and negative life events (Strunk et al., 2006 – please cite). The update bias is the tendency to update beliefs more in response to information that is better than expected than information that is worse. The latter is theorized to be a mechanism generating the former.

I have noted a few sentences that need to be amended to correct for the above confusion, but I suspect there are other incidents I missed. Please correct third sentence of abstract; pg 4 line 44; pg

5 line 22 – the authors cite papers putting in question the optimism bias and the update bias but this study is about the update bias. In theory one bias may exist while the other does not – the authors should clarify which is of interest in this paper and which reference is related to which phenomena; pg 7 line 27.

We apologise for our lack of clarity regarding the difference between optimism and update biases, and thank the reviewer for providing a clear explanation. We have amended this in our submission.

Reviewer: 2

Comments to the Author(s)

Hobbs et al. suggest a study to thoroughly and independently replicate an article published in 2014 that assessed biased updating of beliefs about negative future events in healthy and depressed individuals. The authors plan to extend this replication in several ways, most importantly by including positive future events. A PhD student in the lab, who helped me write this review, and I agree that this extended replication would considerably advance the literatures on belief updating, optimism, and depression. Therefore, we strongly favor this registered report. Below, we provide several points for improvements on the current plan. We realize that a few of these points would equally apply to the to-be-replicated study; nevertheless we think the authors should try as much as possible to improve on this earlier study.

Thank you to the reviewers for providing a very clear discussion of the limitations of the positive life events included in our submission. We agree with many of the points raised by the reviewers. We therefore propose to reduce the number of positive life events to 40 (matching the number of negative life events included in the original paper that we have been able to obtain validated statistics for within Korn et al's (2014) specified probability ranges of 10-70%). Reducing the number of positive life events has enabled us to use a higher quality of events, addressing many of these concerns. We have addressed each of their points in detail below.

Major points:

1) Stimuli: The authors now include positive stimuli and replaced some of the negative stimuli from the earlier study (because some original stimuli did not comply with their criteria, which is fine). However, in our opinion there were several issues with the stimuli; some conceptual and some related to insufficient information. We realize that it is very difficult to compile a sufficient number of good stimuli but the points below should ideally be addressed—or at least the caveats should be mentioned.

o Many of the stimuli are overlapping, both for negative and positive events. That is, these events are statistically not independent. To name but a few:

♣ “Being a victim of fraud, computer virus or being hacked” & “Fraud when buying something on the internet” & “Victim of computer fraud leading to loss of money”

♣ “Getting married before the age of 40” & “Getting married in the summer” & “Getting married or having a civil partner”

♣ “Living in a house that you own” & “Owning a house” & “Buying a home with help of friends”

♣ “Earn more than £30,000 or more a year (before tax)” & “Earning £570 or more per week” are almost exact equivalents.

Stimuli that were overlapping were removed, for example: getting married before the age of 40, buy a home with help from family or friends, fully own your home before turning 65, rent out 2 or more properties, rent out your property privately, owning a dog, owning a cat.

o Many of the positive events are likely to happen at least once (and some are even likely to happen multiple times). Such events should clearly be avoided since this was a major issue in earlier attempts to test biased belief updating in healthy individuals as the authors noted themselves in their introduction. Alternatively, the time horizon could be limited as has been done previously.

Events were for example:

- ♣ “Attend a ballet performance or a dance event
- ♣ “Attend a music concert”
- ♣ “Go on a beach holiday”

Events that were likely to happen multiple times have been removed, for example: visit a museum or gallery, attend a music concert, go to an opera, go on a caravan trip or camping trip, go to a national park, attend a ballet performance or a dance event.

- Relatedly, items falling in the previous category are likely to have happened before; plus some others; e.g., items related to vacation. This could affect participants’ probability estimates.

In addition, the events that were removed as outlined in the previous section the following items have been removed: having some savings, visit France. Furthermore, as in the procedure outlined by Korn et al (2014), in the task participants are asked to estimate their chances of experiencing each event in the future irrespective of prior experience. We also ask participants to rate their previous experience and familiarity of each of the life events, and have amended our analysis plan to examine the influence of these ratings on our findings of update biases.

- o Some stimuli are quite vague, e.g.,
- ♣ “Death due to reasons considered avoidable”

As outlined in response to reviewer 1, we are happy to include only the life events used by Korn et al (2014) that validated probabilities were available for (40 negative life events and two training events in total). This would therefore remove vague stimuli such as the example given.

o Many positive events problematic because of what we would like to dub the “inverse Anna-Karenina principle,” following Tolstoi’s statement that all happy families are alike but that unhappy families are unhappy in different ways. Here, negative events are alike for everybody but positive events are differently positive for many people. That is, almost all people want to avoid negative events equally (at least most of them; e.g., divorce may be a positive event in a deleterious relationship). But the positivity of many positive events depends on whether people actually want to experience them. Again a few examples:

- ♣ “Being admitted into Oxford University ...”
- ♣ All items related to marriage, engagement, wedding, honeymoon, children, etc.
- ♣ All items related to owning a house, a pet, etc.
- ♣ Many items related to vacations

We are not sure how to exactly deal with this issue; maybe an individual selection of events for each person (a priori or a posteriori) based on preference ratings.

As outlined in response to reviewer 1, we agree that there is likely to be some variation in the extent to which events are viewed as positive. Participants are asked to provide a rating of positivity and negativity for each life event. We have incorporated an analysis that would allow us to ascertain whether variations in positivity influenced our findings regarding update biases.

Additionally the following life events have been removed: being admitted into Oxford University, owning a cat, owning a dog

o Some positive items do not seem to be positive—or only for altruistic people. Pre-ratings could help here. E.g.,

♣ Lend money to a friend or family member

♣ Doing some unpaid volunteer work

In order to address this, the following items have been removed: set up a regular donation to a charity, lend money to a friend or family member, doing some unpaid volunteer work.

Furthermore, as we have discussed above there is likely to be some variation in the extent to which events are viewed as positive. We therefore ask participants to rate the positivity of each event, and have included this as an additional predictor in the linear regression models to ascertain whether this may influence our findings.

o Some positive items seem to imply a bad starting point, such that the item would only be considered good if you started off in a bad position. E.g.:

♣ “Earn more than £13 an hour” is not a high salary (~50% more than national minimum wage), so this would only be good if you’re current salary is below that

♣ The same applies to “Earn more than £30,000 or more a year (before tax).” We are not sure what the average wage is but why not choose a salary that is obviously much better than most people’s, e.g., £90,000?

All positive events fall within the same probability range used in the study of Korn et al. (2014). That is, the average chances of experiencing each event range from 10% to 70%. We aimed to select the events that are achievable for at least 10% of the population. An event such as “Earn more than £90,000 a year” would fall outside of this probability range. Unfortunately available statistics are not available for wages at different percentiles, but only at median and interquartile points. In previous research conducted with a similar sample within our lab the mean age of participants was 22 (SD = 7) and 89% were students. Therefore, for the majority of participants we expect that earning £30,000 greater than £13 per hour would be viewed positively. Additionally, ratings are obtained for the extent to which events are viewed positively. We are therefore able to test the assumption that positive events are indeed positive.

o The probabilities to be shown should be included in the Table.

o Ideally the sources of the used probabilities should be included as well. The authors write that these sources were of “sufficient quality.” This should be made a bit more explicit.

We apologise for omitting this from our original submission. This is now included in the amended paper.

- Related to the previous point, participants may hold (second-order) beliefs about the reliability or credibility with which the presented probabilities can be acquired from statistical data. This is likely to influence belief updating. Such differences in estimated reliability may also lead to important difference between negative and positive events since statistical bureaus and medical and criminal databases tend to track negative events more reliably (probably because of the “inverse Anna-Karenina principle”). One way to address this would be to ask participants or some independent group to rate perceived reliability. Alternatively, the authors could rank the reliability of their sources and the precision of the provided probabilities.

Participants are informed that presented probabilities have been derived from reputable statistical sources.

o A few items are a bit problematic because they are contingent on other events, which are also included, e.g., divorce is contingent on being married

Whilst we agree that some life events are contingent on other events (e.g. divorce is contingent on being married) we believe that as these are life events this would still apply to participants that would not meet the initial criteria (e.g. both participants that are married and not married would be able to provide a likelihood of being divorced). We also ask participants to report their experience of the event, and control for this in the analysis, and thereby address the issue.

2) Group assignment:

o PHQ-9 was not in the to-be-replicated study. This should be made clear. Can the authors provide literature on the correlation between PHQ-9 and BDI, which was used in the previous study?

We have highlighted that the PHQ-9 was not included in the original paper in our amended submission. We wish to clarify that both the PHQ-9 and the BDI-II will be included in this study. The PHQ-9 was included in this study as it is a more appropriate screening measure for Depression, and is used in UK clinical care to a greater extent than the BDI-II⁸.

Within the general population a correlation of $r = 0.73$ has been reported between the PHQ-9 and BDI-II⁹. Within clinical populations a correlation of $r = 0.77$ has been reported¹⁰.

- Relatedly, the to-be-replicated study recruited partly in clinical settings and assessed diagnosis in a personal interview. This is different here. Again, this should be clarified and motivated.

We aim to recruit participants from the general population that experience clinical levels of Depression, as determined by PHQ-9 scores greater than or equal to 10. These participants would therefore meet criteria to be treated within clinical settings, as recruited by Korn et al (2014). Furthermore, diagnosis will be assessed using the Clinical Interview Interview Schedule Revised. This provides ICD-10 diagnoses for affective disorders. It is a fully structured self-administered computerised assessment that has been validated for use by lay researchers¹¹. This is therefore comparable to the Structured Clinical Interview used in the original paper, but allows for administration by non-clinicians.

- Relatedly, the authors plan to conduct “sensitivity analyses” by only including participants with clinical depression in some follow up analyses. How many individuals exceeding the threshold on the PHQ-9 are expected to also fulfill the diagnostic criteria? That is, what would be the power of these analyses?

As discussed in response to reviewer 1, we are unable to specify an exact N as participants are recruited on the basis of PHQ-9 scores. Therefore, whilst this is indicative of participants experiencing depression it does not confirm a clinical diagnosis. Previous work in our lab that has recruited participants meeting the same criteria of PHQ-9 scores (greater than or equal to 10), found that 75% of participants met diagnostic criteria for a depressive disorder. Additionally, 92% of participants in a healthy control group recruited on the basis of PHQ-9 scores less than 5 did not meet diagnostic criteria for a psychiatric disorder. We therefore estimate that approximately 40 participants would meet the inclusion criteria outlined by Korn et al for the depressed group, and approximately 49

participants would meet the inclusion criteria outlined by Korn et al for the healthy control group. A sample size of 89 participants would provide 89% power to detect an effect of $\eta_p^2 = 0.113$ at an alpha level of 0.005.

3) Power calculations:

o Why was the alpha-level set at 0.005? This is of course desirable. But it is also different to the earlier study.

Researchers have recommended using a more stringent alpha level, such as 0.005, to lower the Type 1 error rate¹². However, we do acknowledge that in the context of replications an alpha level of 0.05 has been suggested. We chose the more stringent level to be conservative and to allow for >80% power at 0.05 should we need to split the sample in the case of evidence of order effects (see next comment).

- As we understood it, the authors calculate the required power, and for the strict replication of the earlier study they use a subset of all participants – doesn't that to some degree defeat the purpose of the power analysis?

As discussed in response to reviewer 1, in regards to the strict replication of only individuals that completed the negative life events first (n = 27 per group), this would provide 86% power to detect an effect of $\eta_p^2 = 0.113$ at an alpha level of 0.05. Whilst this is less power than provided by the entirety of our sample, this would be adequate to detect the previously reported effect. Furthermore, this would primarily be an issue if order effects influenced our findings. As stated in our submission, we plan to determine whether the order in which participants completed the life events using an adjusted linear regression model. The subgroup analysis using a smaller sample would only be conducted if order effects were present.

4) Overall length of experiment: This seems to be quite long (at least double the time of the earlier study).

o Especially the depressed group may suffer from concentration difficulties, fatigue, etc. Could the number of trials or ratings be reduced (which may mitigate some of the concerns mentioned above)? Could the experiment be split into two sessions on different days? Alternatively, would there be breaks?

Following further piloting of this task with 70 positive and 70 negative life events included we agree that this task is quite long (approximately 1 – 1.5 hours for completion of the task alone). In combination with reviewer's criticism of the content of positive life events and use of new negative life events (beyond those used by Korn et al), we therefore propose to reduce this to 40 positive life events and 40 negative life events. Previous work¹ has used this number of positive and negative life events. We believe that this is a more appropriate option than splitting the experiment into two sessions, as this may produce time confounds.

- Relatedly, the authors plan to assess only participants who performed the blocks with the negative events first (to be more similar to the earlier study). What would be the power of these analyses (see before)?

As discussed previously, a sample of 27 per group would provide 86% power to detect an effect of $\eta_p^2 = 0.113$ at an alpha level of 0.05. Whilst this is less power than provided by the entirety of our sample, this would be adequate to detect the previously reported effect. Furthermore, this would primarily be an issue if order effects influenced our findings. As stated in our submission, we plan to determine whether the order in which participants completed the life events using an adjusted linear

regression model. The subgroup analysis using a smaller sample would only be conducted if order effects were present.

- The repetition of blocks with 1st and 2nd estimates might alter the updating bias because participants learn over the course of the experiment that they will have to re-estimate the event probabilities. The authors could assess this.

Participants are instructed at the beginning of the task that they will be asked to provide a second estimate of experiencing life event, after being told the average estimate. Learning effects should therefore be minimal. Furthermore, our design follows that of Korn et al (2014), where participants were counterbalanced to estimate the likelihood of the event happening or not happening. There was therefore a similar repetition of blocks. Additionally, as the completion order of positive and negative life events, as well as of happening versus not happening is counterbalanced this should reduce the influence of potential learning effects.

- 5) Analyses: Overall, it is good that the authors plan to conduct LMEs and ANOVAs.
- o We are not completely sure why comparing updates between first estimates with desirable or undesirable feedback would be circular. This approach is definitely different from selecting non-independent ROIs in neuroimaging. The calculations are just giving an updating metric scaled by the estimation error.

In Korn et al's (2014) analysis negative life events are categorised as desirable or undesirable on the basis of initial estimates and average probabilities. Specifically, an average probability higher than the initial estimate would be undesirable, whereas an average probability lower than the initial estimate would be desirable. Update biases are calculated by the difference between initial and second estimates. Therefore, initial estimates are included in the model in two separate ways, (1) as the predictor in the calculation of event desirability and (2) as the outcome in the calculation of update bias. Korn et al (2014) note that initial estimates of negative life events were significantly higher for depressed participants. The depressed group therefore have a higher number of desirable trials, and subsequently a greater opportunity to demonstrate optimistic updates, potentially biasing results.

The linear regression model we propose with participant's mean re-estimates as the outcome, and initial estimates, average probabilities and group as predictors uses all available data to assess whether depression status predicts change in initial estimates whilst taking into account the influence of average probabilities.

- We are not quite sure about the presented LME: Shouldn't there be a binary regressor for desirable and undesirable information (and also an interaction term for information and group)? Alternatively, a regressor with (signed) estimation errors could be included. Or two regressors: One with desirable estimation errors and zeros otherwise and another with undesirable estimation errors and zeros otherwise. This would roughly correspond to an analysis in the earlier study, in which correlations between estimation errors and updates were calculated separately for desirable and undesirable events on an individual participant level.

We apologise for any confusion but we have included a mixed-effects linear regression model in our analysis plan that includes desirability as a binary predictor, group, and an interaction term between group and desirability as the reviewers suggest (page 14, lines 3-10 in our original submission). This analysis replicates the ANOVA analysis conducted by Korn et al (2014).

- Are there any exclusion criteria of participants with insufficient trial numbers? That is, what is the minimum number of trials per bin?

As in Korn et al (2014) there is not a specified number of trials that participants are required to complete. Korn et al (2014) report that on average healthy controls did not respond on 1.79 trials (SD = 1.18), and depressed patients did not respond on 1.5 trials (SD = 1.29). Likewise, there were a minimal number of trials where estimation errors were 0 (healthy controls mean = 0.68, sd = 1.20; depressed patients mean = 1.28, SD = 1.78). This did not differ between groups ($p > 0.1$). We therefore expect that there will only be a small proportion of trials that will meet exclusion criteria.

- What happens if updates “overshoot” (i.e., updates exceed the feedback) or go into the wrong direction? This could happen for single items or on average for some participants? In some events, this could be a result of simple typing mistakes, which could unduly influence the updating measure (or the estimates of the LMEs), especially if single trial updates are divided by single trial estimation errors.

Thank you for highlighting this oversight. As we have outlined in response to reviewer 1, we agree that the current procedure confuses situations where participants consistently move away from the average probability with situations where participants consistently move towards the average probability. We have therefore amended this to follow the procedure used in a number of papers examining biased updating, where update scores are calculated so that positive scores indicate a move towards the average probability, whereas negative scores indicate a move away from the average probability (irrespective of valence and desirability). In specific, the absolute difference between first and second estimates will be calculated, and then coded as positive if the update is in the direction of the base rate and negative when the update is in the direction away from the base rate. We have amended this in our submission.

Trials where estimates fall outside the specified probability range of the event happening of 3% to 77% will be removed (as stated in our submission). This will address possible typing errors that may unduly influence updating measures.

- Do the authors also conduct ANOVAs with scaled absolute mean update scores?

The primary analysis reported by Korn et al (2014) used absolute mean updates scores as the outcome (an analysis using scaled absolute mean update scores was included as an additional analysis). Our primary replication analysis therefore uses mean update scores. We would be happy to include an analysis using scaled mean update scores, however the linear regression model outlined on page 14 lines 18-50 in our original submission allows us to look at re-estimates adjusted for initial and average estimates. This therefore accounts for differences in mean estimation errors, as was the purpose of Korn et al's (2014) additional analysis using scaled absolute mean update scores.

- Did the authors consider including random slopes in the LMEs?

Subject was entered as a random intercept in our model to control for clustering within repeated measures. This therefore controls for non-independent between data points. While the use of random slopes can slightly reduce the Type 1 error rate, it also reduces the power of the model¹³.

- Are random effects for items included in the LMEs? This could mitigate some of the concerns above.

As discussed above, subject was entered as a random intercept in our model to control for non-independence of repeated measures. Data is analysed at an aggregate level according to valence and desirability, rather than trial level. It would therefore not be possible to include event as a random effect within our current analysis plan.

o The authors could also check interactions of negative and positive events for initial estimates.

We include two separate linear regression models according to valence of events to examine whether initial estimates differ according to group (hypothesis 3 and hypothesis 4, page 15, lines 30-50).

Minor points:

6) How will it the data be made publicly available?

We plan to publish an anonymised dataset on Open Science Framework.

7) The terms 'probability' and likelihood' are often used synonymously in everyday language, but in a research paper that deals with probabilities it might be useful to use the correct statistical term. So all cases of 'likelihood' should be changed to 'probability' (at least in the technical write-up; for the text given to participants this does not necessarily apply, given that the participants probably aren't aware of the distinction)"

Thank you to the reviewers for pointing out this inconsistency, we have amended this in our submission.

8) In the hypotheses section the authors expect a "reduced optimism bias" in depressed sample, which is completely fine and corresponds to the earlier study. But in the introduction they write mostly about an "absent optimism bias." This could be adjusted in the introduction.

We apologised for theses inconsistencies, this has been amended this in our submission. Korn et al (2014) used the term an 'absence of optimism bias', however we believe it is more appropriate to phrase this as a 'reduced optimism bias', as evidence that a optimism bias is reduced in those with depression compared to those without would be taken as sufficient support of the earlier findings.

9) The authors could also briefly introduce the hypotheses about the correlations between the initial estimates and depressive symptoms. Furthermore, the wording could be a bit briefer and more precise (e.g., correlation instead of association).

We have amended our wording in the hypothesis section to be more precise. As a linear regression models for this hypothesis including a binary group variable the hypothesis has been amended to specify a difference between groups (rather than correlation as suggested by the reviewers).

10) Post-ratings: How would they be used? Are they all necessary?

We ask participants to rate the events for vividness, familiarity, prior experience, emotional arousal, negativity, positivity and controllability. This allows us to control for variations in these factors. We have added an additional analysis with these factors as additional predictors. This will allow us to test whether the relationship between group and update bias persists when adjusted for these

factors. These ratings replicate those used by Korn et al (2014) and were thus deemed as necessary for inclusion.

11) What is the targeted age range? This should be roughly matched to the used event stimuli.

Participants are aged 18 and over. In previous research with a similar recruitment criteria conducted within our lab participants on average were 22 years old (SD = 7, minimum = 18, maximum = 65), which we anticipate in the current study. Calculated probabilities fall across the life span.

12) Rewriting the section 'task procedure'. This section seems is somewhat confusing and could profit from a restructuring.

- It might be useful to take the last sentence of that section ("This procedure is a replication...") and put it at the beginning of the section. The authors already do this in the next section ("Task outcomes" starts with "The subsequent task outcomes follow the procedure outlined by Korn et al (2014)"), so it would make the paper's writing style more consistent.

This has been reformatted according to the reviewer's suggestions.

o It might also help to create a figure outlining the block order. First the authors talk about blocks (1,3,6,8), then about blocks (2,4,7,9), then again about (1,3,6,8) and then about the missing blocks (5,10). Visualizing this might clarify everything and not have the reader trying to piece together what happened in which order

We have rewritten the task procedure section and amended figure 1 to clarify this section. We have also included two supplementary tables to clarify the block order and counterbalancing of the task.

13) Add labels to figure 3. Adding 'negative-desirable', 'negative-undesirable', 'positive-desirable', and 'positive-undesirable' should be a small change that makes the figure a little more obvious

We apologise for any confusion but figure 3 already contains these labels (see below).

Negative Life Event - Desirable

Negative Life Event - Undesirable

Positive Life Event - Desirable

Positive Life Event - Undesirable

Reviewer: 3

Comments to the Author(s)

I think this is a strong piece of research with a solid theoretical basis and important implications. I think the description of methodology would benefit from some clarification around the trials and the definition of desirability. In both cases, I found I had to draw some diagrams and flow charts to be able to follow the process. I didn't find Fig 3 particularly useful in terms of explaining what is described as desirability. If I have understood correctly, "desirability" refers to the extent to which the provided estimates are perceived as desirable or undesirable to the participants? If so, this isn't immediately clear and it would be useful to have this explicitly stated. I also wonder whether a diagram with the blocks might be useful - there are so many dichotomies (happen/not happen; desirable/not desirable; positive/negative) that with the counterbalancing it gets very hard to conceptualise. While I appreciate and applaud the rigorous approach, I think it would be helpful to the reader to present some of this in diagrammatic form.

We have reformatted the method section and figure 3 on the basis of the reviewer's comments. We have added captions on figure 3 to clarify that events are categorised as desirable when (1) a negative life event is less likely to occur than first estimated, (2) a positive life event is more likely to occur than first estimated, whereas events are categorised as undesirable when (1) a negative life

event is more likely to occur than first estimated, (2) a positive life event is less likely to occur than first estimated. We have included a supplementary table to illustrate the precise breakdown of blocks, and have amended the text in the task procedure section to make this clearer. We have also provided a supplementary table of the possible counterbalancing permutations for each of the blocks.

I have two queries around the analysis. (1) I may have misunderstood but you state that trials are removed where the estimation error is 0 and yet the description of analysis on pg 15 (line 22-26) seems to contradict this?

We apologise for the lack of clarity regarding this. Analyses which include the binary desirability variable exclude these trials as it is not possible to categorise trials as desirable or undesirable when estimation errors are 0. However, the analysis on pg 15 (lines 22-26) includes these trials as desirability is not calculated. This was stated on lines 19-20, page 13 of our submission, but we have further clarified this in our submission.

(2) I am unclear what the statement about removing those trials where estimates are <3% or >77% means or where in the procedure this was specified to participants (line 22, pg 14). Otherwise, I think this is an interesting and robust piece of research.

In keeping with Korn et al (2014), participants are told that the probability of the events happening lies between 3% and 77% (see lines 7-8, page 10 of our submission)

References

1. Shah P, Harris AJ, Bird G, Catmur C, Hahn U. A pessimistic view of optimistic belief updating. *Cognitive Psychology*. 2016;90:71-127.
2. Garrett N, Sharot T. Optimistic update bias holds firm: Three tests of robustness following Shah et al. *Consciousness and Cognition*. 2017;50:12-22.
3. Marks J, Baines S. Optimistic belief updating despite inclusion of positive events. *Learning and Motivation*. 2017;58:88-101.
4. Chen C, Takahashi T, Nakagawa S, Inoue T, Kusumi I. Reinforcement learning in depression: a review of computational research. *Neuroscience & Biobehavioral Reviews*. 2015;55:247-67.
5. Eshel N, Roiser JP. Reward and punishment processing in depression. *Biological psychiatry*. 2010;68(2):118-24.
6. Prince M, Knapp M, Guerchet M, McCrone P, Prina M, Comas-Herrera A, et al. *Dementia UK: -overview*. 2014.
7. Rutherford A. *Introducing ANOVA and ANCOVA: a GLM approach*: Sage; 2001.
8. Kendrick T, Dowrick C, McBride A, Howe A, Clarke P, Maisey S, et al. Management of depression in UK general practice in relation to scores on depression severity questionnaires: analysis of medical record data. *Bmj*. 2009;338:b750.
9. Martin A, Rief W, Klaiberg A, Braehler E. Validity of the brief patient health questionnaire mood scale (PHQ-9) in the general population. *General hospital psychiatry*. 2006;28(1):71-7.
10. Kung S, Alarcon RD, Williams MD, Poppe KA, Moore MJ, Frye MA. Comparing the Beck Depression Inventory-II (BDI-II) and Patient Health Questionnaire (PHQ-9) depression measures in an integrated mood disorders practice. *Journal of affective disorders*. 2013;145(3):341-3.
11. Brugha TS, Bebbington PE, Jenkins R, Meltzer H, Taub NA, Janas M, et al. Cross validation of a general population survey diagnostic interview: a comparison of CIS-R with SCAN ICD-10 diagnostic categories. *Psychological Medicine*. 1999;29(5):1029-42.
12. Benjamin DJ, Berger JO, Johannesson M, Nosek BA, Wagenmakers E-J, Berk R, et al. Redefine statistical significance. *Nature Human Behaviour*. 2018;2(1):6.
13. Matuschek H, Kliegl R, Vasishth S, Baayen H, Bates D. Balancing Type I error and power in linear mixed models. *Journal of Memory and Language*. 2017;94:305-15.

Appendix B

Associate Editor Comments to Author (Professor Chris Chambers):

Associate Editor: 1

Comments to the Author:

The revised manuscript was returned to two of the original reviewers. As you will see the reviews are polarised, with Reviewer 1 recommending rejection and Reviewer 2 recommended acceptance following minor revision. Some of Reviewer 1's concerns arise from deviations from the original study protocol, and as noted in my previous decision letter, transferring the article type to the Registered Reports track (as the authors have done), means that the authors are no longer subject to the Replications criteria. However, the reviewer also raises concerns about many of the extended elements of the design, which are pertinent will need to be addressed through comprehensive revision or rebuttal. Reviewer 2 is more positive and notes some areas requiring further attention, primarily in the selection of positive life events.

This submission has presented a number of challenges, both in terms of polarised reviews and also the transfer between article types. On balance, given the positive appraisals of Reviewer 2 (and Reviewer 3 from the previous round), I am convinced that the study has potential merit, provided the concerns that relate specifically to the study as a Registered Report (as opposed to a direct Replication) can be addressed. Therefore I invite the authors to submit a final revision. To avoid unnecessarily prolonging the review process, I will make a final Stage 1 editorial decision to accept or reject following the next submission.

Response: We thank the editor for their balanced approach to this article, and for the opportunity to submit a final revision. Please see below for our response to the reviewer's comments.

Comments to Author:

Reviewer: 1

Comments to the Author(s)

The revised manuscript did not address most of my original concerns. The major concern is that the manuscript does not meet Primary Criterion #2 - The manuscript does not describe a sufficiently valid (i.e. close) and robust (e.g. statistically powerful) replication of the original study methods. The proposed study is not a replication attempt. It is a proposal for a new study, asking novel questions with new experimental conditions, new stimuli and examining a non-clinical population. It simply cannot be considered a replication attempt. Below I list the main deviations from the original study:

Major Deviations from original study:

1. **New Experimental Condition:** The original study examined how people with depression update their beliefs about negative life events. The new study suggests testing a new experimental variable - comparing update in response to negative life events with updating in response to positive life events. This additional variable was not a factor in the original study. Including this new variable means that the question asked is different, the design is different, the stimuli are different. The new question is indeed interesting, but it is quite simply not a replication attempt and should not be submitted via the replication path.
2. **New Stimuli (new items and new associated stats – which are miscalculated and misestimated):** The authors suggest using 40 new positive stimuli which were not used in the original study as well as 40 old negative stimuli which were used in the original study.

However, even for the “old stimuli” the authors plan to present new probabilities to the subjects associated with those stimuli. They claim that the reason for this is that “Unfortunately, it was not possible for us to use the baseline probabilities used in the original paper as the probabilities and their sources were not reported.” Yet, in a previous study by one of the current authors (Shah) the authors “thank Tali Sharot and Christoph Korn for providing a list of the negative life events and their associated probabilities” (acknowledgements). This indicates that (i) the current authors should have the materials they need, and (ii) the original authors are willing to provide any necessary materials to the current authors. If the current authors no longer have the materials previously provided to them by the original authors, have they attempted to request these materials again from the original authors?

The journal clearly states the following:

“In some cases, the published methods of the original study may not provide enough detail to permit a close replication (Stage 1 Primary Criterion #1), and the original authors of the study may also be unwilling or unable to provide the necessary missing information. In such cases, estimates of the original procedures will be accepted provided the authors can establish that the chosen method is a reasonable estimation of the original approach, and provided (where applicable) they can supply documentary evidence to the editor, such as emails, proving any attempts to obtain such information from the original authors.”

The authors then provide an additional reason for using new statistics “Furthermore, in their paper Korn et al specify that life event probabilities must fall within 10% to 70% to allow for sufficient movement in re-estimates. This prevented us from including a number of life events used in the original paper, as they were rarer or more common than these specified probabilities.” Are the authors suggesting that the original authors falsely claim that they are using events with

probabilities between 10-70 when in fact they were not? Or perhaps different sources provide different estimates and the current authors are using different sources than the original authors?

While I strongly suggest the authors simply obtain the original statistics from the original authors, as I explain below that the current authors have miscalculated and misestimated the probabilities associated with the stimuli in Supplementary Table 2 and elsewhere.

The question the subject is asked is "How likely is this event to happen to you in the future". This means that the associated probability needs to reflect the likelihood of a condition (lets say Alzheimer's) happening to a person in their lifetime (i.e. anytime in the next 50 years or so) and not the current % of people suffering from a condition.

Lets take Alzheimer's – the current authors claims that the likelihood of a person suffering from Alzheimer's anytime in the future is 0.68%. Yet "According to the Alzheimer's Association, 10% of all people over the age of 65 have Alzheimer's disease, and as many as 50% of people over 85 have it." <https://www.alz.org/alzheimers-dementia/facts-figures>

The authors employ the same miscalculation for almost every stimulus. Here is another example – they claim the likelihood of ever being fired from your job is 1.5%. However, the source they attach to this number suggests this is the number of currently unemployed individuals. This is not the same as the probability of being fired over your life-time (most subjects have about 40 more years of work years). Moreover, being unemployed is not the same as being fired, as one can be fired but still be employed elsewhere.

Moreover, many of the sources they provide have nothing to do with the associated stimulus they claim. For example, they claim the likelihood of fraud when buying something on the internet is 6% (see Supp Table 2) - as their source they cite articles about Hepatitis B. Online fraud b.t.w is about 70% over your life time according to some sources. For domestic burglary – they cite an epilepsy paper. They claim it is 2.3% when in fact it is 30% in London (where the very first update study was conducted). For epilepsy they cite a heart failure paper. For heart failure they suggest the likelihood is 1.39% when in fact it is 20% according to www.heart.org ("Of all adults 40 and older, one in five Americans will develop heart failure in their lifetime") and as their source they cite a paper about hospital injuries. And so on and so forth.

3. Non-Clinical Population. The original study examines individuals that have been clinically diagnosed with depression "Participants were assessed for psychiatric disorders using a structured clinical interview (SCID-I; Wittchen et al. 1997) by a cognitive neuroscientist (C.W.K.), who had been trained by a psychotherapist in conducting the SCID-I." (Korn et al., 2014). The subjects had severe, clinically diagnosed, depression - an average BDI of 32.6 with an SD 7.69 (for those not familiar with the BDI - these numbers reflect severe depression). The controls had a BDI of 4.3 (SD 3.56). In contrast, the current authors have no plans to conduct a clinical diagnosis. Moreover they suggest also recruiting subjects with only moderate depression (PHQ-9 > 10). This is a critical deviation. The original study examined a clinical population that were carefully diagnosed as having clinical depression. The "replication" suggests examining a convenient sample that are not clinically diagnosed with depression. Whether the subjects have clinical depression is not a minor deviation in the characteristics of the population - it is the major point of the whole study.

Insufficient Power

1. 57% of number of original trials. The authors suggest using 57% of the number of trials used in the original study (the original study included 70 negative events and the current study only 40). This means power is greatly reduced.

2. Incorrect power calculation in justifying N, resulting in insufficient power. The authors start by claiming their power is 95% with 108 participants, but that in fact only 89 are likely to meet criteria so power will be 89% for 89 subjects. Then they say that in fact only half of 108 will undergo the same procedure as in Korn et al thus 54 subjects (27 in each group) can be considered a replication which will give 86% power. However, they do not take into account that out of those 54 not all will reach criteria. They say in fact that only 75% will reach criteria. That means only 40 subjects (20 in each group) will be considered in the replication. I do not think that will provide enough power. The original study had 19 subjects in one group and 18 in the other. Given the publication bias I assume they will need significantly more than 20 in each group. My suggestion is - first provide the calculation of how many is needed in each group for a set power (lets say 85%). Then double that number (as only half the subjects are planned for replication)- that number will reflect $\frac{3}{4}$ of the number you actually need due to those not meeting criteria.

Response: As outlined in our previous response and noted by the editor, we have transferred to the registered reports route, so these deviations are no longer relevant.

We have contacted Dr Korn to request the probabilities used for negative life events in his previous study. Dr Korn kindly provided these probabilities, but agreed that we should calculate new probabilities to reflect potential changes over time.

Regarding the reviewer's more specific notes on the calculations of probability of life events, we have repeated the process of identifying probabilities of negative life events, focusing on lifetime risk as advised by the reviewer. We have therefore obtained the best available plausible *lifetime* estimates of risk for 42 negative life events (supplementary table 1). For the remaining 30 life events, the probabilities did not fall into the specified category of 10-70% (which allows for sufficient movement in participant estimates), or lifetime risks were not available (supplementary table 2). If researchers wish to use different sources (as it may be possible to obtain a range of estimates for a life event, as noted by the reviewer), it will be possible to replicate our work using the study materials we plan to make available on Open Science Framework.

Thank you to the reviewer for noting the errors in supplementary table 2 which were the result of a formatting error. These have now been amended.

Other problems

The rest of my concerns are related to the new experimental design and stimuli. This relates to the portion of the study that is not meant to be a replication but meant to be a new design (which as mentioned above makes this submission inappropriate in the first place).

1. The authors claim that inserting a confound of "controllability" will provide new information. The confound will not provide new information as it covaries with valence, thus one will not be able to conclude anything about valence nor about controllability

Response: At present it is unclear whether perceived controllability does indeed vary as a function of valence as the reviewer suggests. Descriptive data from piloting (n = 16) indicates this is a possibility. However, as outlined in our submission we plan to repeat our primary analyses adjusted for participant's ratings of life events (including controllability). If the results of our primary analysis persist after adjustment, this suggests that participant's perceptions do not influence belief updates. If results differ following adjustment we will be able to conduct exploratory follow-up analyses to explore how differences in ratings influence belief updates. Furthermore, we will also publish the materials and data for this study as open access allowing

other researchers to explore the influence of participant’s perceptions of life events of belief updating. We have clarified this in the manuscript.

Table 1

Pilot Participants (n = 16) Ratings of Life Events according to Valence

	Valence	
	Negative	Positive
Positivity		
Mean	1.4	4.8
SD	0.4	1.1
Negativity		
Mean	5.0	1.6
SD	0.6	0.4
Arousal		
Mean	3.3	4.1
SD	1.2	1.2
Familiarity		
Mean	4.0	4.4
SD	1.0	1.2
Vividness		
Mean	3.9	4.5
SD	1.2	1.1
Controllability		
Mean	3.2	4.2
SD	0.6	1.0

2. The authors claim they will account for controllability and the fact that “positive events” are not necessarily positive in their analysis. However, no details about the new analysis is provided. They only offer a vague sentence on page 44, with no sufficient details about statistical analysis plan.

Response: We have specified in our analysis plan that the regression models for our primary analyses will be repeated with participant ratings added as additional predictors in separate models. We have added further details to clarify this in the manuscript.

Descriptive data from piloting (table 1) suggests that positive events are regarded as more positive than negative events.

3. The authors claim they will include the original statistical approach - using an ANOVA – yet no such plan has been added to the manuscript.

Response: As stated in our previous rebuttal, ANOVA and linear regression are equivalent statistical models and both follow general linear modelling, it is the information presented that differ¹. Linear regression has the advantage of providing more interpretable regression

coefficients, aiding reader comprehension. We have further clarified this in the manuscript. However, we would be willing to include an ANOVA as the editor sees fit.

Reviewer: 2

Comments to the Author(s)

We thank Hobbs et al. for responding to our comments, especially for providing additional details on stimuli and analyses. Most of our points have been addressed. Below you will find a list of some remaining issues, which are almost all about the selection of positive events.

1) Positive Stimuli (previously major point #1): The sub-selection of 40 (out of 70) items improved the stimuli.

a. Some items still overlap (or are contingent on each other): E.g., many events relate to marriage or vacation. Could this be reduced further? Alternatively, could these events be clustered in LME analyses?

Response: Due to the relatively high number of trials participant complete, it is inevitable that there will be some overlap in categories of life events. There are similar levels of overlap between life events for both negative and positive life events (e.g. there are a number of physical health conditions within negative events) and also events that are contingent on each other (e.g. for negative life events, osteoporosis leads to bone fractures). Current overlap between positive life events therefore allows greater comparability between valence conditions.

b. Positive events that are rated as negative: Ratings are a good way of addressing the issue that positive words may not be regarded as positive by all participants. We are just not sure whether the authors would treat these events as “negative” or whether they would exclude these events altogether. It sounds as if the authors think that including these ratings as a covariate would mitigate all individual differences, which may not be the case if most of the events are rated as negative by some participants. That is, depressed participants might be more likely to rate “positive” events as negative.

Response: Thank you to the reviewer for highlighting this possibility. Descriptive data from piloting suggests that participants do regard positive events as more positive and negative events as less negative (and vice versa for negativity; table 1). However, as stated in our previous submission and mentioned above, we will repeat the primary analyses with participant’s ratings included as an additional predictor. This has the effect of controlling for possible variation. If effects are similar after positivity ratings are taken into account, this suggests that variations in perceptions of positivity have not influenced our findings. However, if effects differ after adjustment we will be able to follow this up in exploratory analyses to explore the influence of participant’s ratings of belief updates. We have also added an additional sensitivity analyses, which will repeat the primary analyses but with positive and negative events classified according to each participants negativity and positivity ratings, to see how robust our findings are to any discrepancies in classification of positive/negative events. Furthermore, we will publish the materials and data for this study as open access, allowing other researchers to explore the influence of variations in participant’s perceptions of life events. We have clarified this in the manuscript.

c. Problematic statistics: The following issues might reduce the credibility of the provided statistics. Maybe part of the solution will be to rate the credibility of the statistics (per item or at least overall)

i. Overlapping events and problematic extrapolation: It seems weird that “celebrating your diamond wedding anniversary” has a probability of 60% and “getting married” has a probability of 66%. Will only 6% get divorced or die? Probably, data for the diamond anniversary are based on people who got married > 60 years ago. These were very

different times. So a simple extrapolation is not possible. (Admittedly, one might say that probabilities for many age-related might get lower within the next 20-40 years). Also, it seems weird that 23% will live abroad but only 13% will go on a holiday abroad for > 2 weeks.

Response: The statistic for ‘celebrating your diamond wedding anniversary’ originates from the Office of National Statistics 2013. Unfortunately, as 60 years has to have elapsed since the start of records in order to measure this statistic it is not possible to obtain more recent data (e.g. data for 2017 only provides probabilities of divorce up to the 53rd year of marriage). Whilst we agree that this statistic may vary in future generations it is not possible for us to calculate this. However, we have adjusted the probability to reflect the proportion of people that are married (66% of people are married, and 16% of people celebrate their diamond wedding anniversary; $0.16 \times 0.66 = 0.1056$).

Furthermore, the outcome of interest in this study is whether individuals with depression differ in their belief updates compared to healthy controls. Minor variations in the presented likelihood of life events (dependent on the source chosen) should have a minimal influence if this is a true effect, rather than contingent on study stimuli.

Based on the reviewer’s comments we have amended our procedure to include a funnel debriefing to examine participant’s perceptions of the credibility of statistic. We will therefore be able to conduct sensitivity analyses to examine whether removing participants who question the credibility of provided statistics influences our findings. As the study data will be published as open access, other researchers will also be able to explore the influence of perceptions of credibility.

ii. Specific example 1: The authors say that the probability of ‘celebrating your 20th wedding anniversary’ is 60%. We think they calculated it the following way: according to (<https://webarchive.nationalarchives.gov.uk/20160106011951/http://www.ons.gov.uk/ons/rel/vsob/1/divorces-in-england-and-wales/2011/sty-what-percentage-of-marriages-end-in-divorce.html>), which we got from the document they cited, the probability of marriage ending because one spouse has died is around 5%, and the probability of marriage ending in divorce is around 35% at the 20-year mark. It seems as if they added those two numbers together and then subtracted that from 100%, meaning that the 60% probability of celebrating one’s 20th wedding anniversary is only true if everyone (or close to everyone) got married. Is that true though? If not, then maybe it should be made clear that this figure of 60% only applies to people who have gotten married in the first place.

Response: The reviewers are correct regarding the source of this calculation. We agree that this therefore only applies to individual’s that are married and have adjusted our calculation to reflect this (66% of people are married, and 60% of people celebrate their 20th wedding anniversary; $0.60 \times 0.66 = 0.396$)

ii. Specific example 2: The probability of celebrating one’s 70th birthday is given as 14%, but the probability of celebrating one’s diamond (60 years) wedding anniversary is given at 16% - so, for the sake of the argument, if we assume that people get married on average at age 10, then at age 70 you’re more likely to be married than alive.)

Response: We have adjusted our calculation to reflect the proportion of people who are married (66% of people are married, and 16% of people celebrate their diamond wedding anniversary; $0.16 \times 0.66 = 0.1056$).

d. Problem with context-dependence of positive life events: it's still not clear what the context is for the positive items, and this can have grave effects on how many items people will rate as positive. For example: 'Be mortgage free before turning 55' really depends on the context: if you're 54 and mortgage free because you've bought a house and paid it off, then that's great; but if you're 54 and mortgage free because you've never been able to afford a house in the first place, then that's not so great. So the question is: what exactly is the scenario under which I'm rating these items? Similarly, 'live abroad' can be great if you've moved aboard on your own wish, but might be terrible if e.g., you're a refugee or you've moved abroad because your spouse accepted a job offer abroad and you feel forced to follow (even though you want to live close to your family). Owning 2 cars or more is great if you can afford it, but the (largely) student sample the authors plan to recruit probably can't afford 2 cars, so they'd have to sell them. So, depending on the context, the same person will rate the same items differently and depending on the context under which participants choose to make their ratings, a lot of the items might be rated as not-positive, meaning that the authors would be left with few data points to use for the positive items (on a personal glance, I'd probably rate most items as neutral, simply because I don't know the context in which they'll take place)

Response: We have amended the phrasing of 'Be mortgage free before turning 55' to 'Fully paying off your mortgage before turning 55'. We agree that there may be some variability in perceptions of positivity but as outlined above we have addressed these by incorporating participant's ratings of positive and negativity for each life event, and plan to adjust our primary analyses with these ratings to assess their influence on belief updates. As mentioned above, we have also added an additional sensitivity analyses, which will repeat the primary analyses but with positive and negative events classified according to each participants negativity and positivity ratings, to see how robust the main findings are to any discrepancies in classification of positive/negative events. It is worth reiterating, however, that descriptive data from piloting suggests that positive events are, on average, regarded as positive relative to negative events (table 1).

e. Vague items:

i. 'Be mortgage free before turning 55': anyone who is younger than 55 and doesn't have a mortgage (i.e., most students) is currently mortgage free

Response: We have amended the phrasing of 'Be mortgage free before turning 55' to 'Fully paying off your mortgage before turning 55'.

iii. 'Fall in love with someone you met on a dating site': the emphasis of this item seems to be the dating site, rather than falling in love (otherwise, why mention the dating site?). To me, the whereabouts of falling in love are somewhat secondary to the falling in love itself and I wouldn't care much whether I fell in love on or off a dating site, so I'd probably rate this items as neutral

Response: Unfortunately it is not possible to obtain a validated statistic for individuals falling in love (although we agree that this would be a useful item to include). As stated previously, we have included participant's ratings of positivity and negativity for each life event into adjusted analyses to examined whether this influences our findings regarding belief updates.

iii. 'Hold an investment': an investment doesn't have to entail a lot of money, so this item is somewhat empty without specifying 1) that that investment is (at least for the person) large, and 2) not depreciating rapidly

Response: Validated statistics are only available for the proportion of individuals subscribed to a Stocks & Shares ISA Account. As there is no minimum investment for some of these, it is not possible to estimate the amount of the investment. As mentioned previously, we have accounted for variations in perceptions of positivity by asking participant's to rate each life event on positivity and negativity. We plan to include participant's ratings of positivity and negativity for each life event into adjusted analyses to examined whether this influences our findings regarding belief updates, and perform additional sensitivity analyses.

f. Reference population: E.g., a probability of 25% for getting engaged on Valentine's Day seems rather high. Is this with respect to all people, i.e., also those who never get engaged, or only with respect to those who get engaged. This has to be clear for several events. Copying mistake in Supplementary Table 2: Something is wrong on page 55. Sources do not fit the events (unless e.g., the British Heart Foundation really cares about epilepsy, while Epilepsy Action really cares about domestic burglary).

Response: We have adjusted our calculation to reflect the proportion of people who are married (66% of people are married and 25% of people get engaged on Valentine's Day; $0.25 \times 0.66 = 0.165$).

Thank you to the reviewer for noting the errors in supplementary table 2 which were the result of a formatting error. These have now been amended.

2) Replication versus new data: Maybe it would be helpful to include a Table listing the differences (and communalities) with the previous work by Korn et al (including recruitment, analyses, inference criteria, etc.). This could also help future readers to appreciate the new aspects of the current work and to assess which effects could be replicated and which not

Response: We have altered our submission to a registered report rather than a replication paper. We agree that there are some deviations from the original paper, but have implemented controls into our design that will allow us to examine the influence of these deviations. We will consider our findings in light of these deviations.

References

1. Rutherford A. *Introducing ANOVA and ANCOVA: a GLM approach*: Sage; 2001.

Dear Professor Chambers,

RE: Stage 1 Registered Report RSOS-190814.R2 entitled "Is depression associated with reduced optimistic belief updating?"

We are pleased to resubmit our stage 1 manuscript to Royal Society Open Science. This manuscript was previously accepted on the 1st November 2019, with a submission date of 6th November 2020 for the stage 2 manuscript. This deadline was extended indefinitely on the 31st March 2020 due to the COVID-19 pandemic, with the hope of beginning data collection once lockdown restrictions were lifted. Unfortunately due to the continuing situation with COVID and subsequent restrictions on face to face testing at the University of Bath we feel it is not feasible to collect data face to face for this project within the timeframe of available funding. We therefore wish to convert data collection for this study to take place online to ensure the safety of participants and researchers. Data collection has not taken place for this project to date.

To summarise, the following amendments have been made to our registered protocol:

- We now wish to recruit participants using the online recruitment website 'Prolific' (<https://www.prolific.co/>), rather than through the University of Bath research participation scheme or advertisement on and around the University of Bath campus
 - We plan to conduct initial surveys on Prolific to identify participants meeting inclusion criteria for our study, using the PHQ-9 to screen participants for depression as in our original protocol. To identify patients with high levels of depression we would restrict participants in this initial survey to those that reported experiencing an ongoing mental health condition on Prolific. A separate survey with the same procedure would be used to recruit healthy controls, without this restriction.
 - To ensure high quality data we would restrict participants across both surveys to those that had previously completed ≥ 5 studies with a 98% acceptance rate. We would also include 8 attention checks in the form of requiring participants to provide specified responses in the self-report questionnaires and belief updating task (e.g. 'Please select Disagree'). We will apply the most stringent of data quality checks by excluding participants that fail any of these attention checks.
 - To ensure a similar geographical context for face to face testing, participants recruited online would be restricted to current residents of the United Kingdom.
- Testing would also occur online rather than in Psychology laboratories at the University of Bath.
 - The online survey software 'Qualtrics' will be used to collect questionnaire self-report measures, and 'Inquisit Web' will be used to collect data for the belief updating task.
 - To ensure compliance with task instructions in an online setting, responses on the task will be restricted to within the probabilities of the event happening / not happening specified to participants at the start of the task. Specifically, participants are informed in the task introduction that presented life events have a 3%-77% chance of occurring. Attempts to enter responses outside of this range will produce an error message and participants will be asked to provide another response within the specified values. We previously planned to allow participants to provide

responses outside of this range, and exclude these trials. However, we feel that this change allows us to make optimum use of data.

- To reduce participant burden for online conditions, we wish to shorten our procedure by removing the Clinical Interview Schedule-Revised (CIS-R), which can take up to half an hour to complete. We previously included this measure to identify participants meeting ICD-10 diagnostic criteria for Major Depressive Disorder according to the CIS-R in order to conduct an additional analysis directly replicating the sample used for a previous study within this field (Korn et al, 2014), which used a case-control design comparing patients with MDD to healthy controls.
 - To mitigate this we are now recruiting from Prolific participants reporting a current mental health condition and we have increased the PHQ-9 threshold from ≥ 10 , reflecting moderate levels of depression, to ≥ 15 , reflecting moderately severe levels of depression. Based on previous research within this group we expect 93% of participants with PHQ-9 scores ≥ 15 to meet criteria for a primary diagnosis of a major depressive episode, versus 65% with PHQ-9 scores ≥ 10 . We believe this change will therefore allow more similarity between our sample and Korn et al's, compared to our previous protocol.
- We previously included a funnelled debriefing procedure to ascertain participants' perceptions of the credibility of presented probabilities during the task. To account for online testing we have replaced this with a self-report measure following completion of the belief updating task. Participants will be asked to indicate how strongly they agree with the item 'The average probabilities presented in the task were accurate', on a five point scale ranging from 'Strongly Agree' to 'Strongly Disagree'.

As previously agreed, we will archive all data, code and digital materials in a freely accessible repository at the point of Stage 2 Acceptance. We also agree to the publication of a Withdrawn Registration if we voluntarily withdraw the submission after Stage 1 in principle acceptance.

Sincerely,

Catherine Hobbs

Appendix D

Thank you to the reviewer and their PhD student for taking the time to re-review our paper. We have responded to each of their points below.

1. The authors could include parts of their cover letter into the manuscript to state up-front why and how they adjust their original plan (e.g., they could mention in the abstract that this is an online study). This would be slightly more transparent. Everybody will understand these changes.

Response: As the reviewers have suggested, we have further clarified in the manuscript the changes made in the abstract (line 29) and study design sections (lines 169-172).

2. The authors apply high quality standards to online recruitment. Two comments here:
 - a. We are not sure if this is easily possible; if they want to the authors could conduct verbal interviews to strengthen and refine the diagnosis of depressive symptomatology.

Response: Due to the sample size and online recruitment method we feel that it is not possible to include a verbal interview with participants within the timeframe available to conduct this study. Additionally, conducting diagnostic interviews would require supervision by a qualified medical professional to mitigate possible ethical issues, falling outside the remit of the current research team. However, to ensure participants are experiencing depressive symptomatology we have increased our PHQ-9 criteria for recruitment into the depressed group to ≥ 15 , reflecting moderately severe to severe levels of depression. Based on a previous study conducted within our lab, we expect 88-93% of participants with PHQ-9 scores ≥ 15 to meet criteria for a primary diagnosis of MDD. We will also target initial screening of participants to those that have reported experiencing an ongoing mental health condition on Prolific.

- b. Can the authors make sure that participants do not search online for the statistics? Probably, the time limit precludes that on a trial-by-trial base. But participants might do that in-between blocks.

Response: The reviewer is correct that the time limit precludes this. In addition, we will use Inquisit Web to run the belief updating task online. The software runs using a full screen display, and it is not possible to exit the window without closing the task and ending participation. We have clarified this in our manuscript (lines 266-267). Participants are therefore not able to search online for statistics using the same computer running the task. It is possible that participants could use an alternative computer or their phone, however as the reviewer noted a 10 second limit is applied per trial reducing this possibility. Twenty events are presented per block, limiting the possibility of participants recalling every event and identifying statistics between blocks. Additionally, a maximum completion time limit is applied on Prolific, removing participants that are irregularly slow at completing the task, which may be indicative of this issue.

3. Have the authors already pre-registered the study with OSF? If yes, this could be mentioned and amended.

Response: We have clarified this further in our manuscript (lines 523-527). In accordance with Royal Society Open Science's requirements for registered reports the previously accepted stage 1 manuscript has been registered on Open Science Framework. Dependent on

acceptance of the current revised manuscript, we will update the Open Science Framework page highlighting changes, prior to collection of data.

4. Stimuli: We would like to stress that—in our view—some of our previous suggestions regarding the stimuli and the probabilities have not been taken into account properly. See details below. To the very least, we deem it necessary to read about these potential limitations in the final manuscript.

a. The positive events (i) overlap considerably (e.g., various items about holidays and marriage), (ii) differ in their positivity (e.g., participants not wanting to get married), and/or (iii) happen with an extremely high probability across the lifetime (e.g., holidays). This makes the comparison between negative and positive events very difficult. If the authors really want to use this list of stimuli, they should make this clear in their descriptions of the methods and results.

Response:

- (i) **Overlapping positive life events:** As we stated in our previous response, due to the relatively high number of trials participants complete, it is inevitable that there will be some overlap in categories of life events. There are similar levels of overlap between life events for both negative and positive life events (e.g. there are a number of physical health conditions within negative events) and also events that are contingent on each other (e.g. for negative life events, osteoporosis leads to bone fractures). Current overlap between positive life events therefore allows greater comparability between valence conditions.
- (ii) **Differences in perceptions of positivity:** As we stated in our previous response, descriptive data from piloting suggests that participants do regard positive events as more positive and negative events as less negative (and vice versa for negativity; table 1). However, as stated in our previous submission, we will repeat the primary analyses with participant's ratings included as an additional predictor. This has the effect of controlling for possible variation. If effects are similar after positivity ratings are taken into account, this suggests that variations in perceptions of positivity have not influenced our findings. However, if effects differ after adjustment we will be able to follow this up in exploratory analyses to explore the influence of participant's ratings of belief updates. Furthermore, we will publish the materials and data for this study as open access, allowing other researchers to explore the influence of variations in participant's perceptions of life events.
- (iii) **Events with high levels of probability:** The statistics we have presented for holidays are based on validated statistics, reflecting the *average* probability across the population. We have excluded events that are very common ($\geq 70\%$) for both positive and negative blocks.

As stated in our previous responses, the outcome of interest in this study is whether individuals with depression differ in their belief updates compared to healthy controls. Minor variations in the presented likelihood of life events should have a minimal influence if this is a true effect, rather than contingent on study stimuli. As suggested, we will discuss these potential limitations in our discussion.

Table 1

Pilot Participants (n = 16) Ratings of Life Events according to Valence

	Valence	
	Negative	Positive
Positivity		
Mean	1.4	4.8
SD	0.4	1.1
Negativity		
Mean	5.0	1.6
SD	0.6	0.4
Arousal		
Mean	3.3	4.1
SD	1.2	1.2
Familiarity		
Mean	4.0	4.4
SD	1.0	1.2
Vividness		
Mean	3.9	4.5
SD	1.2	1.1
Controllability		
Mean	3.2	4.2
SD	0.6	1.0

b. The reputability/applicability of the statistics varies for both negative and positive events. We rather randomly selected three items (obesity, skin burn, create a successful start-up). We did not find numbers for lifetime prevalence in the cited papers. Also, it seems rather odd to select data from Northern Saudi-Arabia for a UK sample for “ski burn.” More importantly, for the positive event “successful start-up for 5 years,” the number in the literature refers to the number of people who have actually launched a start-up—not the number of people overall. So, the number is not what the item in their Registered Report is about. The question is not about how many business survive the 5-year mark; the question is about what is “the average probability that someone in the same environment as the participant experiencing these events in their lifetime.” Now let’s think this through just a minimal amount: how many people even start a business? Probably only few people.

Response: Regarding skin burns, the statistic is stated in the abstract ‘Our study reported that there were 66.4% of the participants had history of burns’ (1). We have corrected errors in the spelling of authors’ names within the reference for this statistic. Validated statistics specifically for skin burns within the UK general population rather than specialised clinics for severe injuries were not available. We have corrected the reference for the obesity statistic, which referred to 2016

data (26% obesity prevalence) rather than 2019 (29% obesity prevalence as we stated). Finally, for creating a successful start up we have adjusted for the proportion of the general population that have reported starting a business (22%) (2), of which 44% are successful, leaving 10% (when rounded to a whole number). However, as we stated above, the outcome of interest in this study is whether individuals with depression differ in their belief updates compared to healthy controls. The accuracy of presented likelihood of life events should have a minimal influence if this is a true effect, rather than contingent on study stimuli.

c. The above point also motivates us to suggest asking participants for the confidence in the statistics presented (even if this was not included in the to-be-replicated study).

Response: As outlined in the 'Debriefing' section of our manuscript (lines 333-339) we plan to ask participants how strongly they agree with the item 'The average probabilities presented in the task were accurate', on a five-point scale ranging from 'Strongly Agree' to 'Strongly Disagree'.

References

1. Alanazi A, Alanazi A, Alanazi M, Alenezi N, Qaisy F, Asiri A, et al. Burn Injuries Prevalence, Causes, Complications and Improvement in Northern Saudi Arabia. *EC Emerg Med Crit Care*. 2019;3(6):383–93.
2. Ipsos. Entrepreneurialism. The Emergence of Social Entrepreneurialism to Compete with Business Entrepreneurialism. 2018.

Friday, 01 October 2021

Dear Professor Chambers,

RE: Stage 2 Registered Report "Is depression associated with reduced optimistic belief updating?"

We are pleased to submit our stage 2 manuscript to Royal Society Open Science. Following the procedures outlined in our stage 1 report, we replicated previously observed effects of reduced optimistic belief updating in depression for negative life events. However, our findings suggest that these effects do not extend to positive life events. Our results add confidence to previous findings that depression is characterised by a reduced ability to update beliefs in response to good news, and suggest further research is required to understand the specificity of this to negative life events.

We believe this paper is a valuable contribution to the readership of *Royal Open Science* in clarifying the role of optimistic belief updating in depression in relation to positive and negative life events. Additionally, this will be the first study using the belief updating paradigm to publish all study materials and data as open access.

The URL for the stage 1 protocol on the Open Science Framework, and the study data, code and materials are reported on page 2 of our manuscript. We can confirm that no data for the pre-registered study was collected prior to the date of the in-principle acceptance of our stage 1 report. We can also confirm that the study has been executed and analysed in the manner originally approved with any unforeseen changes in those approved methods and analyses clearly noted.

Sincerely,

Catherine Hobbs

Appendix F

Reviewer: 1

1. When using the update bias task it is critical to control for estimation errors (EEs) (Sharot & Garrett 2021; Garrett & Sharot 2017). Failure to do so will produce false results, as been shown with simulations (Garrett & Sharot 2017). The authors follow Korn et al's method of controlling for EEs. This is good. However, they seem to dismiss the results when controlling for EE, and instead highlight the ones that are known to produce false findings.

In particular, despite the analysis which EE is controlled for showing a clear belief update bias for positive events (Figure , pg 90), throughout the manuscript they wrongly claim that no such effect was found: "optimistic belief updating did not occur for positive life events" (abstract). This claim needs to be corrected to say "optimistic belief updating occurred for positive life events". Similar statement should be corrected throughout the manuscript including multiple times in the discussion and results section.

Response: Our primary analyses as set out in stage 1 of our registered report replicated Korn et al's (2014) primary analyses, which were conducted on update scores unadjusted for estimation errors. We therefore focus on these results in our discussion as these were pre-registered. It should also be noted our main focus is on the role of depression in optimistic updating, not the presence or absence of optimistic updating in the general population (although this is clearly relevant). However, we have now further highlight results controlling for estimation errors in our discussion and link to the wider debate around optimistic updating in healthy samples. Optimistic updating to positive life events is clearly a controversial issue and our mixed results for positive life events do little to resolve the issue. We have tried to present these as logically and transparently as possible. We have made all data and resources available as open access to allow other research groups to conduct further exploratory analyses. The ongoing controversies in this field might be better resolved if other groups did the same. We now mention this in the discussion.

2. The authors cite reference 14 and 15 as dealing with optimistic belief updating, including in the introduction (pg 5 line 25) and in the discussion (pg 5 line 40). This is incorrect. Those references are not about updating and should be deleted from these places and elsewhere where the authors suggest they are.

References 18 and 19 are not about the belief update task the authors are using. Thus, it is misleading to cite those in a way that makes the reader assume they are.

As for reference 16, it is odd to cite an abstract published almost a decade ago, which has never since been reviewed nor appeared in a journal. I suggest deleting this, for fear of mudding the waters with statements in an abstract, which is not accompanied by details of methods, statistics or results. The editor can indicate if this is encouraged or not.

3. Pg 6, line 18-22 is missing a reference.

Response: In line with Royal Society Open Science's guidelines for stage 2 registered reports we have not amended our introduction as the stage 2 submission should reflect the stage 1 manuscript which was previously approved following extensive peer review.

4. In the pre-registration did the authors calculate the N needed to have enough power to find a correlation between update bias and BDI? If not, then perhaps they are underpowered? Please calculate the N needed based on Korn et al., 2014, and if underpowered state so clearly.

Response: In our pre-registration we powered our study based on the interaction between desirability and group for update scores in Korn et al (2014). An a priori power calculation in G*Power indicated that 47 participants would be required to detect effects of the same magnitude to those observed by Korn et al (2014) for the association between updating and BDI-II scores in the depression group at an alpha of 0.005 and 80% power. Our depression group consisted of 54 participants; we were therefore sufficiently powered to detect the effect observed by Korn et al.

5. Korn et al, 2014 reports all paired t-test including comparing undesirable update between control and depression groups , and comparing desirable update between control and depression groups— please do the same and provide p values.

6. The authors make some claims (e.g., pg. 19 line 36-42) regarding the presence or absence of differences in updating for desirable and undesirable trials within each group but provide no statistics (e.g., no p nor t). Please provide statistics each time claims are made about the presence or lack thereof of differences.

Response: We have reported tukey adjusted follow-up contrasts examining differences in desirability by group based on our models for our primary analyses and those adjusted for expectation errors relating to differences between undesirable and desirable trials. We refer to these in our discussion.

7. In pg 19 line 54 the word “some” could be misleading. Suggest deleting

Response: We have removed the word “some”.

8. Pg 21 – first paragraph: The authors say they find no three way interaction, yet the figure shows there may be one, and thus they conduct exploratory analysis they did not pre-register. This is fine as long as they first conduct a Bayes factor (BF10: Rouder et al., 2009) analysis that will assert whether the null finding is reliable or not. Only if the null finding is statistically reliable then the exploratory analysis is justified. The critical number is the Bayes Factor – and the critical question is whether it is greater or smaller than 1. This can be done using JASP (Love et al., 2015). A BF10 greater than 1 provides evidence for rejecting the null hypothesis, a BF10 smaller than 1 provides evidence for the null hypothesis. This is common when reporting null findings.

Response: As the follow-up contrast t-tests based on our primary analyses now address the extent of differences in updating in each group for positive life events we have now removed the exploratory analysis that we did not pre-register.

9. Pg 24 line 8 – the authors only cite Sharot and Sharot & Garrett as providing evidence for the “good news” “bad news” effect. This may seem to do injustice to many others who have provided evidence of this effect. I suggest including:

Kuzmanovic B, Jefferson A, Vogeley K. 2016. The role of the neural reward circuitry in self-referential optimistic belief updates. *NeuroImage* 133:151–162. doi:10.1016/j.neuroimage.2016.02.014

Kuzmanovic B, Rigoux L. 2017. Valence-Dependent Belief Updating: Computational Validation. *Frontiers in Psychology* 8:1087. doi:10.3389/fpsyg.2017.01087

Ma Y, Li S, Wang C, Liu Y, Li W, Yan X, Chen Q, Han S. 2016. Distinct oxytocin effects on belief updating in response to desirable and undesirable feedback. *Proceedings of the National Academy of Sciences of the United States of America* 113:9256–61. doi:10.1073/pnas.1604285113

Kappes A, Faber NS, Kahane G, Savulescu J, Crockett MJ. 2018. Concern for Others Leads to Vicarious Optimism. *Psychol Sci* 29:379–389. doi:10.1177/0956797617737129

Oganian, Y., Heekeren, H. R., & Korn, C. W. (2019). Low foreign language proficiency reduces optimism about the personal future. *Quarterly Journal of Experimental Psychology*, 72(1), 60-75.

Krieger JL, Murray F, Roberts JS, Green RC. 2016. The impact of personal genomics on risk perceptions and medical decision-making. *Nat Biotechnol* 34:912–918. doi:10.1038/nbt.3661

Korn, C. W., Prehn, K., Park, S. Q., Walter, H., & Heekeren, H. R. (2012). Positively biased processing of self-relevant social feedback. *Journal of Neuroscience*, 32(47), 16832-16844.

Korn, C. W., La Rosée, L., Heekeren, H. R., & Roepke, S. (2016). Social feedback processing in borderline personality disorder. *Psychological Medicine*, 46(3), 575-587.

Eil, D., & Rao, J. M. (2011). The good news-bad news effect: asymmetric processing of objective information about yourself. *American Economic Journal: Microeconomics*, 3(2), 114-38.

Response: In this sentence, we are referring to evidence relating to the processing of negative *life events*. We have now included the suggested references assessing belief updating for negative life events.

10. Pg 24 line 8 – I believe reference 15 is irrelevant here. For reference 16 see my comment above. When citing 17 I suggest stating that 17 has itself been critiqued by Garrett & Sharot, 2017. Similarly, when citing 55 (line 40 pg 28) it should be stated that an attempt to replicate the study failed and that the study was criticized (Garrett & Sharot 2021) for not following the appropriate guidelines for running and analyzing the update bias task as detailed in Sharot & Garrett, 2021.

Response: We now reference both papers critiquing the task and rebuttals to provide a balanced overview of current opinions.

11. Discussion, pg 25 line 16-22, may need to be changed depending on BF10 results. Pg 23 line 37 may need to be changed based on power calculation results.

Response: As outlined above we were sufficiently powered and have removed the exploratory analysis so we have not amended this part of the discussion.

12. Discussion pg 28 line 35: “That we did not find evidence of optimistic belief updating for positive events in either people with depression or healthy controls, speaks against the existence of a universal optimistic updating bias. Alternatively, our pattern of results are a potential sign of an underlying problem with the validity of the optimistic updating paradigm, which is currently being debated [14–17,55]. As such, reduced optimistic belief updating effects associated with depression may require further validation after a more robust measure of optimistic updating is developed.” – But the authors do find optimistic updating bias in healthy individuals and in depressed individuals (figure pg 90). They only fail to find one when they fail to control for EEs. All past studies of the update task control for EEs. This is the appropriate thing to do. These sentences are thus not supported by the authors data and need to be deleted.

Response: As mentioned above, our pre-registered primary analyses are unadjusted for estimation errors and are thus given the appropriate weight in our discussion. We have amended our discussion to further comment on our results relating to estimation errors.

13. Discussion: “It has been shown that a more appropriate way of administering the task necessitates asking participants about both their personal as well as the average person’s likelihood of experiencing life events, that is, ‘the base rate’ [17].” – Why is this more appropriate? If the authors claim so, they must explain. It has been shown that the results do not change whether following this or not (Garrett & Sharot, 2017). If the authors believe it is much more appropriate, why did the authors not do so? And does this mean that the authors results in this paper are of no clinical value?

Response: We have amended this sentence to reflect that it has been suggested that this is the most appropriate way of administering the task. We considered incorporating this element into the current study. However, asking participants about both their own and the average likelihood of experiencing life events for both positive and negative life events would have doubled the testing time for participants on an already lengthy task. To reduce fatigue effects, we focused only on adding positive life events within this study. However, we have made all study materials openly available to allow other researchers to amend our procedure to investigate this possibility. We have outlined the potential clinical value of our findings extensively in our discussion.

14. Discussion “There are also limitations in asking participants to accurately compare their chances of experiencing life events relative to the average person including, but not limited to, scale attention and base rate regression [14,15].” I do not see how this sentence is relevant, as in the task the subjects are not asked any such thing.

Response: We have amended this section to reflect that we are referring to previously published critiques of the belief updating task.

15. Discussion – last sentence – the authors suggest mechanism of the effect are unknown. This is inaccurate. There are studies detailing the neural mechanism of the effect in depression (Garrett et al., 2014) including studies detailing the role of dopamine in producing it (Sharot et al., 2012)

Response: We have amended this to clarify that we are referring to cognitive mechanisms.

Reviewer: 2

- Even though not fully conclusive, the differences between negative and positive items are relevant and important: One main point here that the authors do not mention or discuss is that positive events are not positive for everybody. In contrast, most negative items are negative for everybody. Such individual differences in the “positivity” of positive events could explain some of the potential differences.

- Relatedly, one interesting thing that we haven't yet seen them discuss is that for the depression group the positive items are less positive and more negative than they are for the healthy group. For negative items, both groups seem to be similar, but for positive items, there seems to be more of a difference. Not sure, whether this difference is meaningful though and what it's driven by, but might be worth discussing this or looking into, to see whether there's anything interesting going on here.

Response: The reviewer raises some interesting points relating to participants' perceptions of the life events. We have now further expanded upon differences in life event ratings between groups in our results section and have visualised these differences to aid the reader's interpretations (Supplementary Figure 2). Whilst there were some differences in ratings of positivity and negativity between groups, these differences appear to be smaller than for other ratings (emotional arousal, familiarity, vividness, prior experience). Additionally, we did not find evidence that perceptions of life events, including positivity and negativity, influenced our findings (Supplementary Table 12). Furthermore, when we recategorized the life events according to participants' perceptions of positivity/negativity our findings were unchanged (Supplementary Table 13). We therefore do not believe that our data suggests that variations in perceptions of positivity and negativity influenced differences in belief updating by groups. We have further clarified this in our discussion.

- Again related: The estimation errors for desirable and undesirable information seem to differ largely for positive events, with undesirable estimation errors being larger than desirable estimation errors. This can also be seen by comparing the right parts of Fig 5 and Fig 7, i.e., between unscaled and scaled updates. This should be explicitly mentioned and potentially be discussed.

Response: We have now further commented on these differences in our discussion section.

- It might also be nice to have a large table in which they list the positivity/negativity etc. for each item, so that future studies can exclude those items that didn't work as well as the others. Or they could just provide a list of items that seemed to work very well to facilitate better future studies. If there aren't many studies that use positive items, then this study might become the basis for future studies.

Response: Thank you for the suggestion, we have now included this as a supplementary table (Supplementary Table 14).

- In the 'Results' section, in the 'Sample' subsection, the authors mention the differences between their two groups but don't mention that the depression group is almost twice as likely (44% vs 23%) to be single. That seems like quite a difference and worth mention whilst they're already listing differences.

Response: We have now included this when outlining the differences between groups.

Reviewer: 3

I don't remember if I picked this up at pre-registered stage but on reading this submission, I was unclear about whether the PHQ was used as a pre-screen on Prolific, i.e. were participants invited to complete the PHQ and then only moved on to the rest of the study if they fell within the defined ranges (over 15 or under 4)? It may be worth making this explicitly clear.

Response: The reviewer is correct that the PHQ was used to pre-screen participants. Amendments to methods are not generally permitted in stage 2 registered reports however we have included a sentence clarifying that participants were only invited to participate in the test phase of the study if their PHQ-9 scores were within the defined ranges per group to reduce potential confusion. We leave it to the editor's discretion as to whether to include this amendment in the final manuscript.

Looking at the list of life events, it seems that there could be an effect of age when it comes to making an estimate. For example, an older person who is already married would be likely to make a more accurate judgement regarding the question about likelihood of being married in their lifetime. Since age was different between the two groups and not included in the analyses, is it possible that this could have affected any of the outcomes?

Response: We agree that this is an interesting possibility. We conducted further exploratory analyses adjusting our primary analyses for age to investigate this. In line with the reviewer's suggestion older participants tended to give lower initial estimates for experiencing positive life events ($\beta = -0.37$, 95% CI: -0.54, -0.20, $p < .001$). However, age was not associated with update scores or initial estimates for negative life events, and results relating to our outcomes were unchanged. We have reported this in Supplementary Table 15.

The conclusions are well thought-out and offer a range of possible mechanisms. Another contributing factor may be the cognitive inflexibility that is common in depression, although if it was a simple neuropsychological feature it might be expected to affect positive and negative life events equally.

Response: The reviewer raises an interesting point which we have now mentioned in our discussion.